# Sensory perception relies on fitness-maximizing codes

Jonathan Schaffner[1,2], Sherry Dongqi Bao [1,2], Philippe N. Tobler [1,2], Todd A. Hare [1,2,4] ✉ & Rafael Polania [2,3,4] ✉

Sensory information encoded by humans and other organisms is generally presumed to be as accurate as their biological limitations allow. However, perhaps counterintuitively, accurate sensory representations may not necessarily maximize the organism's chances of survival. To test this hypothesis, we developed a unified normative framework for fitness-maximizing encoding by combining theoretical insights from neuroscience, computer science, and economics. Behavioural experiments in humans revealed that sensory encoding strategies are flexibly adapted to promote fitness maximization, a result confirmed by deep neural networks with information capacity constraints trained to solve the same task as humans. Moreover, human functional MRI data revealed that novel behavioural goals that rely on object perception induce efficient stimulus representations in early sensory structures. These results suggest that fitness-maximizing rules imposed by the environment are applied at early stages of sensory processing in humans and machines.

One of the main goals of the neural and behavioural sciences is to understand what general principles explain the solutions evolution has selected to extract and process information from the environment to guide behaviour. Half a century ago, it was postulated that neural systems should represent the sensory world as accurately and efficiently as possible by exploiting information about the statistical regularities of the environment, an idea known as efficient coding[1,2].

Efficient coding in sensory perception is typically assumed to be based on an information maximization criterion—that is, the sensory world must be represented as accurately as possible. One may think that this criterion makes sense for early sensory systems, as this is precisely the role of a sensor: a good measurement instrument must reliably measure the environmental variable that it was built for. However, the information maximization criterion does not necessarily consider the behavioural goals of the organism[3–7].

Is it reasonable that our sensory systems invest their limited resources to represent the world as accurately as possible irrespective of the organism's goals? This question has kept scientists and philosophers busy for centuries and led to heated debates across various fields and domains including neuroscience, psychology, economics and evolutionary biology[8]. Some views support the idea that organisms should represent objects as they exist in the world, as closely as biological limitations allow[9,10]. Others posit that perceptual representations should be in general different from the actual physical world, and these representations should directly map onto the utility they offer to the agents[11–13].

In partial support of the latter idea, recent neurophysiological evidence shows that early sensory systems represent not only information about physical sensory inputs but also non-sensory information according to the requirements of a specific task and the behavioural relevance of the stimuli[14–18]. This does not necessarily imply that sensory systems should give up representing the 'veridical' world, as it has been demonstrated that neural systems can develop computational strategies that allow representing multiple behaviourally relevant features alongside objective sensory information[19,20]. However, this line of research provides no indication of the actual benefit of having such mixed neural representations at the earliest stages of sensory processing, or how this information could be used to efficiently

[1]Zurich Center for Neuroeconomics, Department of Economics, University of Zurich, Zurich, Switzerland. [2]Neuroscience Center Zurich, Zurich, Switzerland. [3]Decision Neuroscience Lab, Department of Health Sciences and Technology, ETH Zurich, Zurich, Switzerland. [4]These authors jointly supervised this work: Todd A. Hare, Rafael Polania. ✉e-mail: todd.hare@econ.uzh.ch; rafael.polania@hest.ethz.ch

guide behaviour, given that we are limited in our capacity to process information.

The study of how systems should trade off the maximization of some utility function relevant to the goals of the organism against information-processing constraints is part of a growing body of research inspired by the work of Shannon, who put forward the idea that when optimizing a distortion function that characterizes the cost of particular errors, not all such errors are equally important. This implies that the unreliability of signal transmission is not necessarily uniform across the space of possible messages that can be transmitted[21]. Similar concepts have been borrowed from the field of statistical mechanics, where information processing in capacity-limited systems can be modelled as the energy required to move away from default states in thermodynamic systems, which can be quantified by differences in free energy[22,23]. These principled approaches have played a fundamental role in neural process theories of early sensory systems[24–28] as well as higher cognitive functions[29–35].

Our work builds on these normative theoretical principles to determine how a (neural) system should allocate information-processing resources to maximize fitness in different situations. We focus on two of the most common problems studied in decision-making: accuracy maximization in perceptual discrimination tasks and reward maximization for situations in which a particular attribute is related to a given currency value. This allowed us to test the following hypothesis: given that noisy communication channels always lose information during transmission, the brain will adapt to the fitness-maximizing rules of a particular environment at the earliest stages of sensory processing. We demonstrate that early visual structures in humans and artificial agents with sensory information-processing bottlenecks follow fitness-maximizing encoding schemes.

## Results

### Neural codes in an insect's retina

Before moving on to humans, we introduce an illustrative example, as anecdotal evidence, to motivate the theoretical framework applied in our experiments. Specifically, we studied the responses of retinal neurons in the blowfly—the large monopolar cells (LMCs)—which encode sensory information about visual contrast levels. These neural codes are considered the first demonstration of efficient coding in biological organisms[36].

Visual features such as shape, colour and texture are important sensory signals that insects use to discriminate between competing flowers and fruit species, with visual contrast playing a key role[37,38]. In our example, we assume that blowflies use knowledge of the different levels of contrast displayed by flowers and fruits to select food sources that promise more beneficial nutrients (that is, reward). In other words, we assume that there is a monotonic association between contrast and reward that makes some contrast discrimination mistakes more costly than others. Please note that this corresponds to the standard and most studied class of economic problems, where choices with a particular attribute are monotonically related to a given currency value. The hypothesis that we test here is that a neural code that simply maximizes information accuracy in the LMCs would not maximize the fitness of the organism.

Concretely, we studied the following problem. Suppose that the distribution of contrasts encountered by the blowfly in its natural environment is given by $f(s)$. We define the function that transforms the contrast stimulus input $s$ to neural responses $r$ in the blowfly retina as $r = h(s)$. Then, what is the optimal neural response shape $h(s)$ if, given biological limitations, such a function can only generate a limited set of neural responses? Under this formulation, the following problem can be studied[39]: find the optimal neural response function under two evolutionary optimization criteria, (1) the probability of mistakes minimization criterion and (2) the expected reward loss minimization criterion.

To solve this problem, we assume that the organism must make choices between alternatives drawn from the stimulus distribution, $f(s)$, which describes the relative availability of the different alternatives in its environment (for example, how often a blowfly encounters a particular flower). The goal is to select the alternative that promises more reward to the organism, as this should lead the organism to maximize its fitness[3,39]. In the case of the blowfly LMCs, one may suppose that the blowfly must often make fine discriminations and that different contrast levels are monotonically related to different reward values.

On the one hand, if the goal of the organism is to minimize the number of erroneous responses (that is, maximize discrimination accuracy between two stimuli $s_1$ and $s_2$)

$$\min_h \iint f(s_1, s_2) P(\text{error}|h(s_1), h(s_2)) \mathrm{d}s_1 \mathrm{d}s_2, \tag{1}$$

it can be shown that the optimal neural response $h(s)$ matches the cumulative distribution function (CDF) of the stimulus distribution (Fig. 1b; see also equation (11) in the Methods). However, this accuracy maximization strategy does not provide a precise account of the distribution of neural responses in the blowfly (Fig. 1).

On the other hand, if the goal of the organism is to minimize expected reward loss (that is, maximize the amount of reward received after many decisions; see Supplementary Note 1 for derivation)

$$\min_h \iint f(s_1, s_2) P(\text{error}|h(s_1), h(s_2))|s_1 - s_2| \mathrm{d}s_1 \mathrm{d}s_2, \tag{2}$$

the optimal fitness-maximizing neural response $h(s)$ provides a nearly perfect account of the neural responses of the blowfly retina (Fig. 1b,c and Methods). We emphasize that the remarkable overlap between the fitness-maximizing predictions and the empirical neural responses presented in Fig. 1 are not the product of curve fitting; instead, these predictions emerge from the normative decision model, which has no degrees of freedom (Methods).

A common approach adopted in computer science and neuroscience to study the way in which a system penalizes estimation mistakes to optimize performance is via the $L_p$ loss function defined as $|\hat{s}(r) - s|^p$, where $\hat{s}$ is the sensory estimation, $s$ is the true signal and $p$ determines how errors are penalized. A recent parameter estimation study asked what type of error penalty best explained the LMC data[40]. However, in this study, the authors did not have explicit hypotheses for the potential evolutionary and behavioural meaning of different values for error penalties, and they instead relied on numerical estimations of $p$ that best explained the data. We demonstrate that the error penalty that provides a nearly perfect fit to the LMC response function corresponds to the blowfly LMC encoding function that guarantees maximal reward expectation to the organism under our sensory-reward mapping assumptions (Fig. 1c and Methods).

While the predictions of reward-maximizing sensory codes and the data from blowfly retinal neurons (that is, the earliest level of encoding) show a striking similarity (Fig. 1), this result does not directly address all aspects of our hypothesis that neural codes in early sensory areas adapt to the organism's behavioural goals. This is because we do not know the specific function linking contrast to fitness for the blowfly, and we cannot show that the code used in their retinas adapts between contexts because we have data from only one context. While we emphasize that this result should be treated at this stage as anecdotal, it inspired us to test the fitness-maximizing hypothesis more directly in humans (see below) and may inspire others to do the same in other animals.

### Adaptive fitness-maximizing sensory codes in humans

To more directly test the hypothesis that sensory perception relies on adaptive fitness-maximizing codes, we implemented an experiment with more than one context in human sensory encoding. To date, it

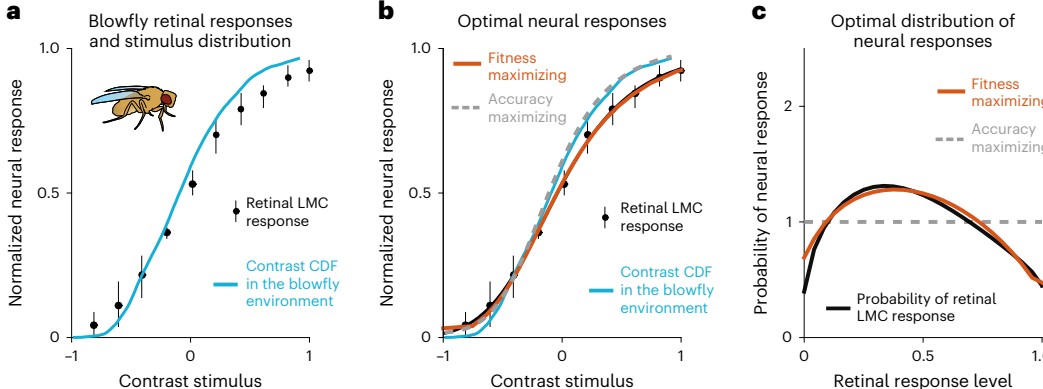

**Fig. 1 | Blowfly LMC responses are better explained by fitness than information maximization coding schemes. a**, Responses measured from the LMCs (black dots) and the CDF (blue line) of contrasts in the natural environment of the blowfly. If accurate perception of the environment is maximized by the LMCs, then the line indicating the CDF should lie on top of the dots reflecting the empirical data. The data points were averaged from $n$ = 6 cells; the range bars show the total scatter (data of the LMC responses was reproduced from ref. 36). **b**, The black dots represent the same empirical data and the blue line the same contrast stimulus CDF as in **a**. The grey dashed line represents the predicted response function from an accuracy maximization code. The orange line indicates a coding rule that maximizes fitness. It matches the data better than

the grey line. The data points were averaged from $n$ = 6 cells; the range bars show the total scatter (data of the LMC responses was reproduced from ref. 36, work distributed with license CC BY-NC-ND 3.0). **c**, Neural response probability density distributions predicted by a fitness maximization rule (orange) also align better with the empirical data (black) than those predicted by infomax coding (dashed grey). This suggests that the fitness maximization model describes the empirical data more accurately than the accuracy maximization model. The same fitness-maximizing solution emerges when studying the $L_p$ reconstruction error penalty, with optimal solution $p$ = 0.5, which is the error penalty that best describes the LCM neural response data[40].

has been widely accepted that the default neural code for orientation perception in humans is information maximization (infomax) coding[41], as it can be shown that this code will minimize the probability of mistakes in perceptual discrimination tasks (Supplementary Note 1). One reason that infomax coding may typically explain human perception well in regard to orientation is that, for humans, orientation information does not typically signify reward and is instead used for navigation purposes. The fitness-maximizing code for orientation perception may thus be equivalent to infomax under standard conditions. Our experiments deviate from these standard conditions to test whether sensory encoding strategies adapt in a manner predicted by the theory of fitness maximization.

We designed behavioural tasks in which, on any given trial, human participants had to choose which of two simultaneously presented orientation stimuli, $s_1$ or $s_2$, was more diagonal (that is, closer to a 45-degree angle; Fig. 2). In experiment 1, the participants were trained in two different contexts but were always tested with stimuli in the same retinotopic locations, while in experiment 2, the participants were trained in only one context or the other but were tested with stimuli in trained and untrained retinotopic locations. The key aspect of both experiments is that decisions were made in two different stimulus–reward association contexts. In one context, the participants were paid a fixed reward for correct discrimination of the more diagonal stimulus in each trial and received no reward for incorrect decisions (henceforth the accuracy context, $K_{acc}$). In the second context, the participants were rewarded depending on the stimulus $s$ that they selected in each trial, and the amount of reward was linearly mapped to the degree of diagonality of the input stimulus (henceforth the reward context, $K_{rew}$). Crucially, the prior distribution of sensory signals $f(s)$ was exactly the same in both contexts. Stimuli close to cardinal orientations were presented the most often to match the statistics of natural scenes that humans typically encounter[42] (Fig. 3a). This experimental design allowed us to test the competing hypotheses that neural codes in early sensory areas (1) maximize accurate representations of the environment and are thus constant in both reward contexts, or (2) adapt between contexts to instantiate efficient coding strategies that maximize fitness. The location-specific training in experiment 2 allowed us to test whether

any adaptation occurs in early sensory regions that maintain retinotopic mappings or only later in downstream circuits that generalize across locations.

We employed a general method for defining efficient codes by investigating optimal allocation of limited neural resources[43]. On the basis of this framework, sensory precision, measured as Fisher information $J(s)$, should be proportional to the amount of resources available $k$ and the prior distribution $f(s)$ raised to a power $q$

$$J(s) \propto k \times f(s)^q, \tag{3}$$

hence known as the power-law efficient code. We show that an advantage of employing this framework is that there is a direct link between the power-law efficient codes and the fitness maximization solutions for the contexts that we consider here (Methods and Supplementary Note 1). In brief, the connection of the power-law efficient codes with accuracy versus reward maximization objectives is the following: if the power-law parameter $q$ is relatively low, it shifts some neural resources away from where $f(s)$ is high and relocates them where it is low. The reason for this spreading of coding resources is that, even though observing a rare stimulus is unlikely (and thus the probability of error will be low), if there is an error, it could be a very costly mistake. Therefore, when stimuli are directly associated to rewards relative to situations in which all mistakes are equally costly, it pays to allocate more neural resources to the segments of the stimulus space where $f(s)$ is low (Fig. 3e and Supplementary Fig. 1). This theoretical link allowed us to derive various qualitative predictions that we used to test whether humans indeed adopt fitness-maximizing, as opposed to information-maximizing, neural codes of sensory perception.

A first prediction of fitness maximization theory is related to sensory discrimination differences in the two reward association contexts considered here. If participants maximize fitness under limited resources, discrimination accuracy for diagonal (that is, oblique) relative to cardinal orientations should improve more in $K_{rew}$ than in $K_{acc}$ over the course of the decision task (Fig. 3e). In line with this prediction, we found an interaction between reward association context, orientation and task phase (early or late) on discrimination accuracy

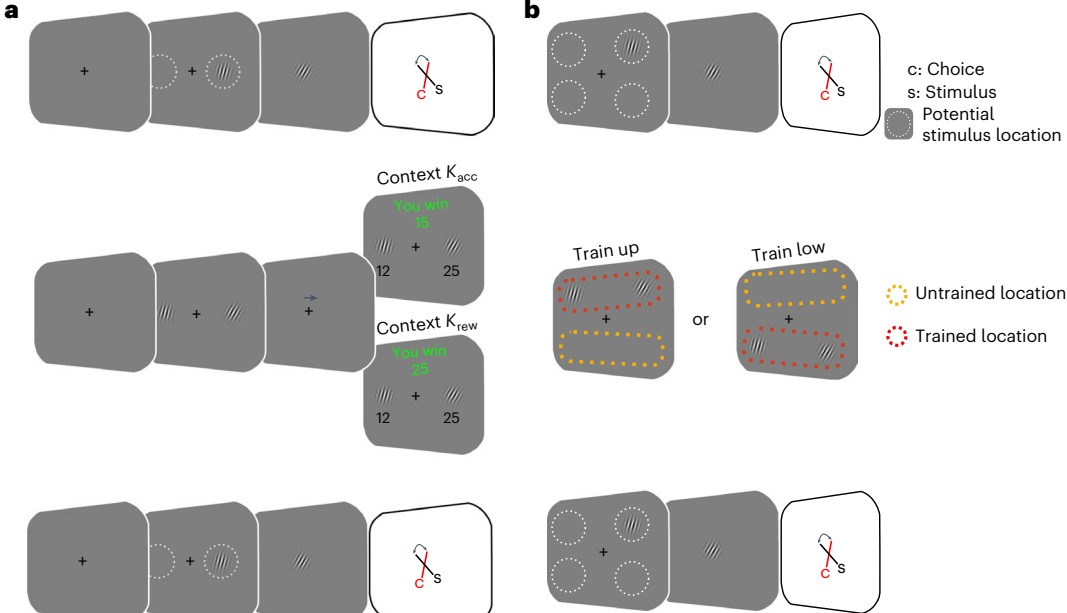

**Fig. 2 | Behavioural paradigms used with human participants. a,b,** Separate groups of participants completed experiments 1 and 2. In both experiments, they performed the 'estimation task' before and after training in the 'decision task'. In the estimation task, after perceiving a Gabor patch stimulus on the left or right side of the screen, the participants had to rotate a Gabor cue in the middle of the screen until its orientation matched the orientation of the perceived stimulus. In the decision task, the participants decided which of two Gabor patch stimuli was more diagonal. In context $K_{acc}$, the participants received a constant reward for a correct decision, whereas in context $K_{rew}$, the reward magnitude was linearly related to the degree of diagonality of the stimuli. In experiment 1 (**a**), the participants ($n = 25$) completed multiple sessions of the estimation–decision–estimation sequence using either $K_{rew}$ or $K_{acc}$ for the decision task. In experiment 2 (**b**), there were four locations for stimulus presentation in the estimation task. However, during the decision task, each participant ($n = 61$) trained in only two of these locations (up or down) and only one stimulus–reward association context (either $K_{rew}$ or $K_{acc}$).

($\beta = 1.46 \pm 0.65$; $P_{MCMC} < 0.001$; Supplementary Tables 1 and 2). Note that this interaction is not driven by a simple increase in sensitivity in $K_{rew}$—that is, a general improvement across the whole orientation space (Fig. 3b). Despite the differences between contexts, fitness maximization theory predicts that for both optimization objectives, discriminability should be higher in regions of the stimulus distribution prior with higher density (that is, greater in cardinal than oblique orientations) because these stimuli occur more frequently in all contexts (see the thick blue and thick red lines in Fig. 3e). The data are consistent with this prediction as well (main effect of obliqueness $s$ in $K_{acc}$: $\beta = -4.50 \pm 0.45$; $P_{MCMC} < 0.001$; in $K_{rew}$: $\beta = -5.50 \pm 0.47$; $P_{MCMC} < 0.001$; Fig. 4a,b). These results thus support our hypothesis that perceptual coding of sensory information uses a fitness-maximizing code.

To further substantiate the conclusion that changes in behaviour were driven by fitness-maximizing codes rather than experience-driven increases in sensitivity, we fit the encoding model to the choice data to estimate parameters $q$ and $k$. In line with the fitness maximization predictions, we found that in context $K_{rew}$ the value of $k$ did not change ($\Delta k = -0.0004 \pm 0.0014$; $P_{MCMC} = 0.61$), while $q$ decreased between the first and last part of the decision experiment by $\Delta q = -0.29 \pm 0.15$ ($P_{MCMC} = 0.03$). The final value of $q$ was significantly smaller in $K_{rew}$ than in $K_{acc}$ ($\Delta q = -0.38 \pm 0.14$; $P_{MCMC} = 0.004$; Fig. 4c). Taken together, our results clearly indicate that the empirically observed behavioural changes in $K_{rew}$ versus $K_{acc}$ were not caused by simple practice-related sensitivity enhancements or differences in monetary payoffs (Methods).

**Fitness-maximizing adaptation at sensory estimation stages**
We next sought to determine whether the form of efficient adaptation observed in the decision task takes place only in downstream decision circuits or whether it is already implemented at earlier processing stages—for instance, in the circuits that generate estimations of sensory stimuli. To answer this question, we had participants perform an edge orientation estimation task before and after the contextual decision-making task (Fig. 2 and Methods). After training in either context, there was a significant decrease in the estimation bias ($P_{MCMC} = 0.02$; Supplementary Fig. 2). A decrease in the estimation bias is predicted by either a fitness-maximizing code or increased sensitivity (Fig. 3c,f). However, similar to the decision task, experience-dependent changes in sensitivity versus a fitness-maximizing code make distinct predictions for the estimation task in terms of the estimation variance. Greater sensitivity would lead to lower estimation variability for all orientations (Fig. 3d). In contrast, the fitness maximization hypothesis predicts that after participants adapt to context $K_{rew}$ in the decision task, estimation variability for more oblique orientations will decrease, while estimation variability for stimuli near cardinal orientations will be slightly higher (Fig. 3g). This is because the theory predicts a shift in coding resources from high-probability cardinal orientations to low-probability diagonal orientations. Crucially, fitness maximization predicts that after exposure to context $K_{acc}$, there will be no change in the relative estimation variability for cardinal versus diagonal orientations. In line with these predictions, we found a significant interaction ($\beta = -0.16 \pm 0.09$; $P_{MCMC} = 0.04$) between the change in estimation variability for oblique relative to cardinal stimuli across contexts (Fig. 5a and Supplementary Table 3).

**Fitness-maximizing codes are retinotopically specific**
A key conclusion that we draw from our results is that fitness-maximizing adaptation in both the decision task and the estimation task appears to have a common origin that does not depend on comparisons between decoded stimuli in downstream decision circuits. To explicitly test whether fitness-maximizing neural codes are indeed present at the earliest stages of sensory processing, we modified the decision and estimation tasks to train and test behavioural performance in retinotopically specific locations. In the modified estimation task, the participants were

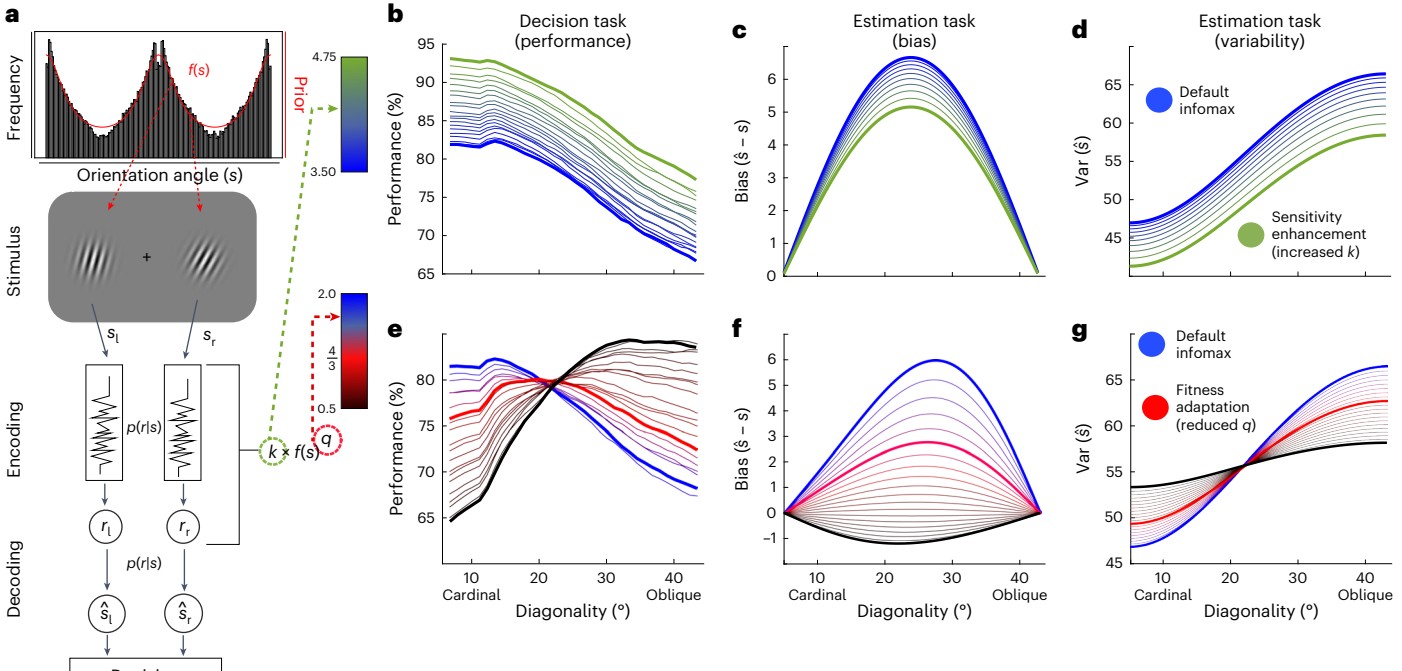

**Fig. 3 | Inference model theoretical predictions. a**, Simplified schema of the decision task. The orientation of the Gabor patches was drawn from the distribution of edges in natural scenes. The orientation $s$ of the perceived Gabor patches is encoded with the internal response $r$. The corresponding likelihood function $p(r|s)$ is constrained by the encoding rule. The prior $f(s)$ is combined with the likelihood to generate an estimation $\hat{s}$. The encoding rule depends on the model parameters $q$ and $k$. **b–d**, Model predictions for the decision and estimation tasks assuming enhanced sensitivity $k$. The blue line represents a common infomax encoding model. As sensory precision increases (blue to green gradient), performance increases consistently at all orientations (**b**). In the estimation task, increased sensitivity $k$ leads to a decrease in estimation bias

$(\hat{s} - s)$ (**c**) and reduced variance over the whole range of diagonality (**d**). **e–g**, Performance predictions as a function of power-law encoding $q$ in both tasks. As in **b–d**, the blue line represents the infomax model. The thick red line shows the prediction of the fitness-maximizing model. As $q$ decreases, performance in the decision task drops for the more cardinal trials and increases for the more oblique trials (**e**). If $q$ decreases for a constant capacity level, estimation biases should decrease (**f**). As $q$ decreases, estimation variability decreases for oblique angles and increases for cardinal angles (**g**). The predictions for behaviour following an increase in sensitivity $k$ or a decrease in the power-law encoding parameter $q$ are thus distinctly different in terms of changes in encoding accuracy and variability.

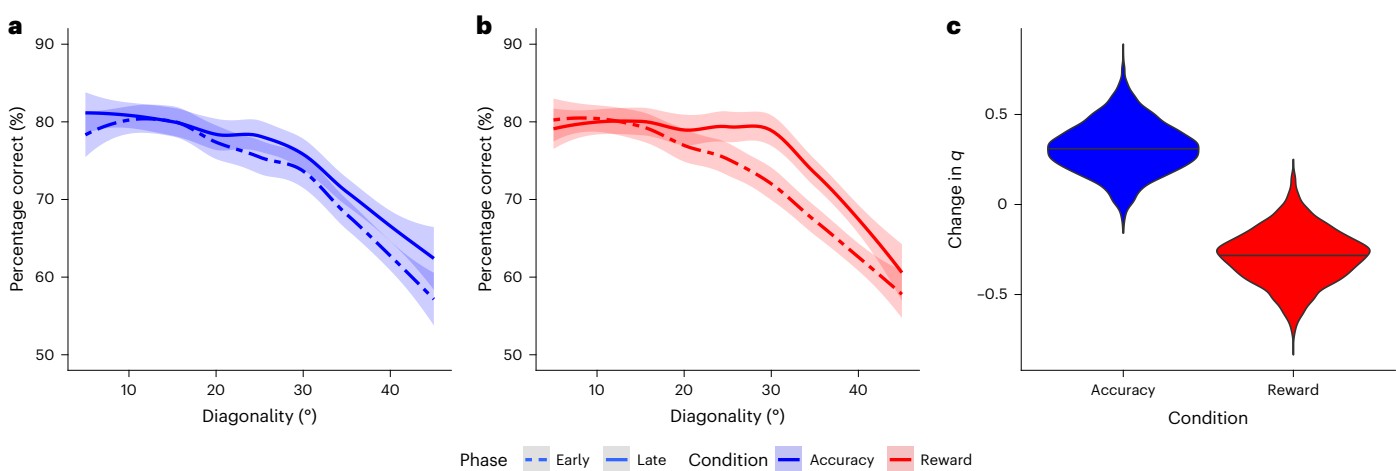

**Fig. 4 | Human performance in the binary decision task. a**, Accuracy does not differ between early and late training trials for any level of diagonality in $K_{acc}$. The lines represent the means, and the shaded intervals denote 90% confidence intervals. **b**, However, in $K_{rew}$ there is a significant interaction such that accuracy increases more for oblique than for cardinal orientations as predicted by a decrease in the $q$ parameter to implement a fitness-maximizing code. **c**, Fitting the early and late training decisions separately in $K_{rew}$ and $K_{acc}$ showed that

decreases in $q$ for $K_{rew}$ training were greater than for $K_{acc}$ training. In each panel, the data from $K_{acc}$ are shown in blue, while those from $K_{rew}$ are shown in red. The violin plots in **c** show the posterior distributions for the group-level estimates of $q$ in late minus early trials for $K_{acc}$ and $K_{rew}$ training sessions. The horizontal black lines indicate the medians of the posterior distributions, and the width of the violin plot represents their density.

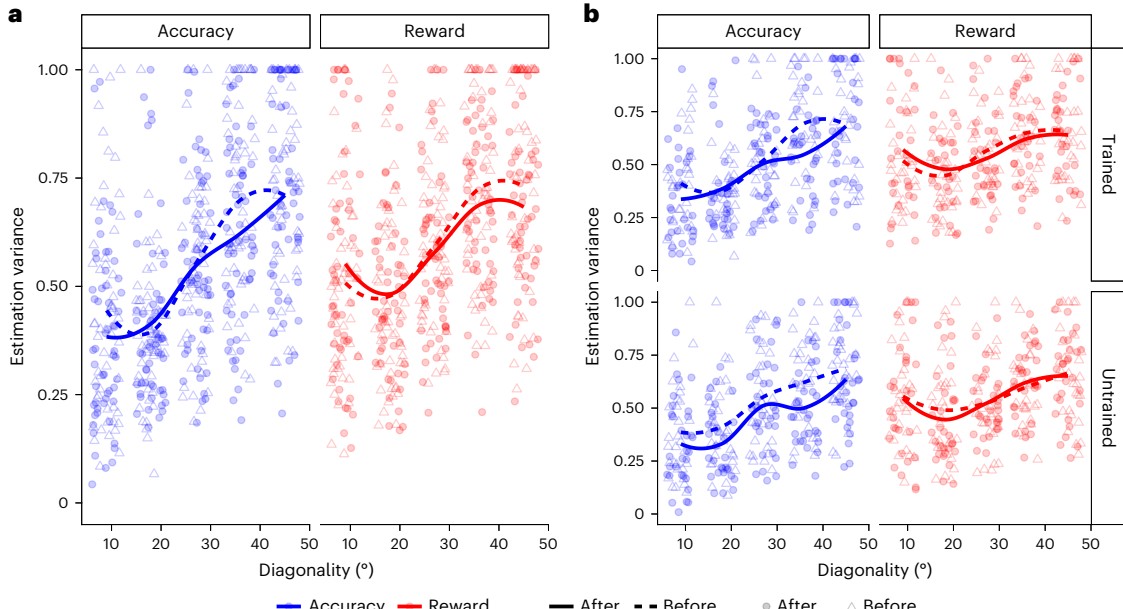

**Fig. 5 | Estimation task performance. a**, Changes in estimation variability for $K_{acc}$ (blue) and $K_{rew}$ (red) stimulus–reward association contexts. Estimation variability before training is shown in the dotted lines and open triangles, while post-training variability is shown in the solid lines and filled circles. The results from $K_{rew}$ training show the interaction between oblique (decreased variance) and cardinal angles (increased variance) predicted by the implementation of a fitness-maximizing coding scheme for linear stimulus–reward mappings. In contrast, $K_{acc}$ training produces no such interaction. **b**, Changes in estimation variability for the $K_{acc}$ and $K_{rew}$ contexts in trained (top row) and untrained locations (bottom row) in experiment 2 indicate retinotopically specific training effects. The $K_{rew}$ training leads to an interaction between oblique and cardinal variance changes in trained locations only. Training $K_{acc}$ does not lead to this interaction in either location.

presented with an orientation stimulus in one of four spatial locations (Fig. 2b). Crucially, the participants in these experiments were trained in only two of these locations during the decision task and completed the decision task in only one context (either $K_{rew}$ or $K_{acc}$). If adaptation is retinotopically specific, then changes in estimation task performance should be specific to the retinotopic locations trained during the decision task. In line with the fitness-maximizing predictions, for those trained in $K_{rew}$, we found the pattern predicted by a fitness-maximizing code in the location-specific changes in estimation variability (location × time(after − before) × oblique: $\beta = -0.12 \pm 0.07$; $P_{MCMC} = 0.04$; Fig. 5b and Supplementary Table 4). For those trained in $K_{acc}$, this interaction was not significant (Fig. 5c and Supplementary Table 5). A comparison across groups showed that the effect was greater in the $K_{rew}$ than the $K_{acc}$ group (context × location × time(after − before) × oblique: $\beta = -0.17 \pm 0.10$; $P_{MCMC} = 0.04$). Together, these results confirm the retinotopic specificity of fitness-maximizing coding rules in humans.

### Artificial neural networks with sensory processing bottlenecks use fitness-maximizing codes

The existence of fitness-maximizing codes in early sensory systems—where they will literally change the way an organism sees the world—must be for very good reasons. We conducted machine learning analyses with artificial neural networks (ANNs[44]) to investigate whether agents with informational bottlenecks must recode their sensory representations to fitness-maximizing schemes to achieve the best performance in decision-making tasks. Alternatively, the downstream decision circuits (which are not involved in estimating orientation as such) may not care about the accuracy with which orientation can be estimated and may have enough flexibility to maximize fitness even if the encoding scheme at early sensory stages is fixed to an infomax strategy.

The precision of neural representations at different sensory processing stages can be studied using recently developed neural network techniques in machine learning. We constructed an ANN

implementation to test how (that is, at what layer of processing) it incorporates the behavioural goals of the agent when encoding sensory stimuli. More specifically, our premise is that the way in which internal representations of retinal sensory information are formed and used in the nervous system can be studied with a variational information bottleneck (VIB)-like objective[45–48], where in general the goal is to minimize the following loss function:

$$\min_{\phi,\theta} \quad E[\text{reward loss}] + \beta \times I, \qquad (4)$$

where $\phi$ and $\theta$ are the parameters of the encoder and downstream decision circuit, respectively. In our ANN, the VIB-like objective trades (an approximation of) the amount of 'visual' information $I$ that the encoder can process with the expected reward loss, via the regularization parameter $\beta$. Note that the analytical solutions developed in our work 'drop' costs on $I$ by assuming that the noise in the encoder is small compared with the dynamic range of the signal (that is, the small-noise approximation, which is commonly adopted in early sensory systems to study neural coding efficiency, often leading to satisfactory predictions[49]). The reason for using the VIB-like objective in our ANNs is that it provides a parsimonious way to induce pressures in the encoder to disentangle information up to a certain bound in a systematic manner (Supplementary Fig. 3).

We implemented an ANN that solved the same task as in our human experiments (Fig. 6a; see Methods for the details). The ANN received two retinal images corresponding to screen locations where the two Gabor patches were presented in our task. Just as in the human experiment, the decision rule that the ANN had to learn was to indicate which of the two input stimuli (left or right) was more diagonal, while maximizing the reward received across many trials. We trained networks in two contexts, corresponding to the human experiments: $K_{acc}$ and $K_{rew}$ (Methods). After the information had been encoded, it was fed to downstream neural circuits that used the encoded information

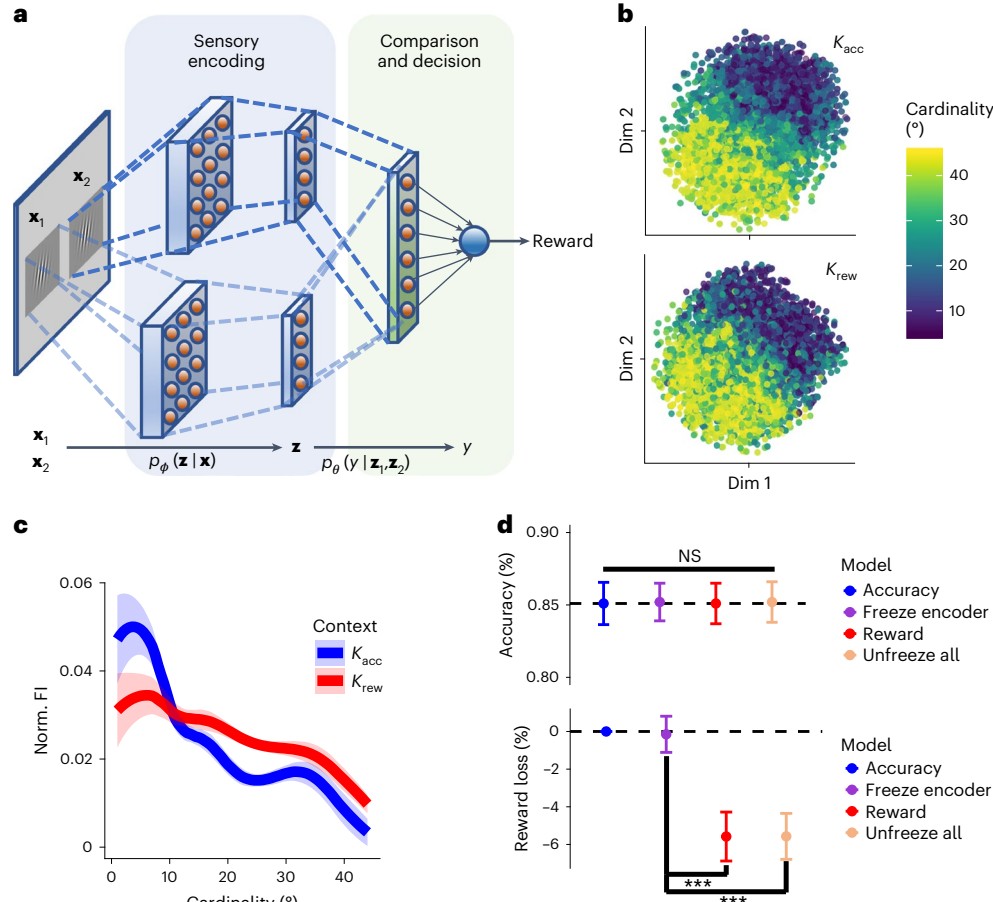

**Fig. 6 | ANN implementation and analyses. a**, Simple schematic of the deep neural network implementation, where $\mathbf{x}_{1,2}$ are the retinal inputs, $\mathbf{z}_{1,2}$ are the latent codes, and $y$ is the network response. **b**, A $t$-SNE analysis revealed that the information bottleneck encoder layer appears to learn a useful representation of the objective levels of cardinality. This is evident from the fact that different angles (blue-to-yellow colour scale) are generally represented in different portions of the multi-dimensional encoding space. The smooth gradient of transition between angle representations might be beneficial for downstream circuits to implement decision rules. **c**, Estimation of Fisher information (FI) in the encoder layer for the networks trained in each context ($K_{acc}$ and $K_{rew}$) shows that the relationship between angle cardinality and FI in the neural network depends on the training context, consistent with the normative solution and behaviour exhibited by human participants. In $K_{acc}$ (blue), FI is higher for cardinal orientations but lower for diagonal orientations than in $K_{rew}$ (red). This suggests that the sensory encoder learns useful representations according to the behavioural goals of the agent. The lines represent the means, and the shaded intervals denote ±1 s.d. across repeated neural network simulations. **d**, We trained networks in contexts $K_{acc}$ (blue) and $K_{rew}$ (red) to have the same levels of discrimination accuracy (top, Bayesian paired $t$-test $P_{MCMC} > 0.51$

for all pairwise combinations) and investigated the amount of reward loss according to the reward contingencies in $K_{rew}$ (bottom). As predicted by the normative model, we found that reward loss in $K_{rew}$ is greater when the network was trained in $K_{acc}$ relative to $K_{rew}$. Next, we investigated whether freezing the $K_{acc}$ network information bottleneck layers after training (purple) would allow this network to reach optimal reward loss when retrained using the reward contingencies in $K_{rew}$. We found that irrespective of the degree of complexity of the downstream network, it was not possible to reduce the levels of reward loss if the information bottleneck encoding was fixed to maximize accuracy, reaching levels matching the network trained from scratch in $K_{rew}$ (Bayesian paired $t$-test $P_{MCMC} = 0.86$). However, when the originally trained $K_{acc}$ network was allowed to learn to minimize reward loss according to the $K_{rew}$ reward contingencies without freezing any network weights, it could reach optimal levels of reward loss reduction (light orange) (Bayesian paired $t$-test $P_{MCMC} < 0.001$). Critically, in this case the encoding scheme changed from infomax to fitness-maximizing as predicted by the normative model. The points represent the means and the error bars represent ±1 s.d. across neural network simulations. ***$P_{MCMC} < 0.001$ (Bayesian paired $t$-tests); NS, not significant.

to solve the task at hand (in our case, select the Gabor patch that was more diagonal), while considering the goals of the agent within the environmental context (for example, maximize decision accuracy or maximize reward consumption; Methods).

After the networks were trained in each context, we first investigated whether the network could disentangle the hidden structure in the retinal image statistics to solve the downstream task. By applying a $t$-distributed stochastic neighbour embedding ($t$-SNE) algorithm to the neural responses of the bottleneck structure, we found that the network indeed learned a useful representation of the scalar angular orientations from the retinal images (Fig. 6b). However, this $t$-SNE

solution provided no direct insight into how the encoder allocated its limited resources (that is, using an infomax or fitness-maximizing code). We therefore analysed the amount of information contained in the encoder layer, quantified as Fisher information, as a function of the angular orientation. This analysis more directly illuminated the pressures shaping the learning of the latent representation in the encoder.

Mirroring the predictions of fitness-maximizing theory and the human behavioural results, we found that, in general, the amount of information in the encoder layer was larger for cardinal orientations than for diagonal orientations. The network thus allocated the limited processing resources in the bottleneck encoder to the more recurrent

portions of the angular stimulus space. Crucially, we found that for ANNs trained in context $K_{acc}$ relative to ANNs trained in context $K_{rew}$, the amount of information was larger for more cardinal angles but smaller for diagonal orientations (Fig. 6c). Moreover, we found that these results were insensitive to the information-processing costs imposed in the encoder (Supplementary Fig. 3). Interestingly, we found that the retinal layer in our ANN architecture in which we did not explicitly incorporate information-processing regularization (Fig. 6a) also revealed signatures of information-processing allocation similar to the ones encountered in the second retinotopic layer (although less pronounced; Supplementary Fig. 4). We also studied how the ANN allocates information-processing resources in the first and second retinotopic layers when informational bottleneck pressures are imposed at the decision-making layer. We found that the fitness-maximizing patterns were also present in this scenario in the second retinotopic layer (Supplementary Fig. 5). We found that the fitness-maximizing patterns in layer 1 were present for high levels of network performance (that is, generally low $\beta$ in the decision layer; Supplementary Fig. 6). Thus, even when information-processing pressures are very small at early sensory stages (and relatively large in downstream decision layers), neural networks still try to develop fitness-maximizing codes at the early sensory stages to compensate for reward loss due to processing limitations in downstream circuits. However, these effects are more pronounced if information-processing constraints are present at early stages. This set of results indicates that the network learns to allocate its neural resources in a fitness-maximizing manner following the predictions of the algorithmic normative theory and the behaviour exhibited by human participants.

The first network analysis additionally addressed the following concern: as downstream circuits are not involved in estimating orientation as such, they may not care about the accuracy with which orientation can be estimated. The ANN architecture we implemented here addressed exactly this question because the objective function the networks sought to optimize did not explicitly incorporate "reconstruction error minimization" (as classically implemented in variational autoencoder architectures[47,48]). Instead, in our ANN architecture, the network had to find encoding solutions that benefited downstream operations supporting decision behaviour. That is, all that mattered to the network was what information about the latent (angular) space was most relevant to solve the decision task at hand and maximize reward. Nevertheless, we found that ANNs learned to implement efficient coding schemes in their encoding layers that maximized reward in each context.

Having demonstrated the efficient, fitness-maximizing nature of the encoder in an ANN, we investigated whether an ANN could achieve solutions similar to the ones obtained in the fitness-maximizing context $K_{rew}$, if we forced the encoder layer to maximize information transmission (that is, to use an infomax code). To test this, we first trained an ANN to maximize decision accuracy and then froze all network weights up to the encoder, but we left the downstream network weights free to change. Theoretically, ANNs with sufficient complexity can interpolate any objective function. One could therefore hypothesize that even if the encoder is restricted to maximize information transmission, downstream circuits could still figure out solutions that maximize reward gain on the basis of the representations coming from an infomax encoder. However, a competing hypothesis comes from an information-theoretic point of view, which holds that, in sensory discrimination tasks that face a bottleneck due to limited resources (like the one we study), once information is lost or processed in a suboptimal manner in one step of a noisy transmission channel, it cannot be recovered, irrespective of how complex the downstream circuits are. In line with the predictions from information theory, we found that freezing the encoder layers after training them in context $K_{acc}$ resulted in a significant reward loss when downstream layers were retrained in context $K_{rew}$ (Fig. 6d). Once we unfroze the encoder layer (that is, allowed it to

depart from the $K_{acc}$ constraint and use fitness-maximizing codes), we found that reward loss was significantly lower than in the $K_{acc}$ trained network (Bayesian paired $t$-test $P_{MCMC} < 0.001$; Fig. 6d, bottom), reaching levels matching the network trained from scratch in context $K_{rew}$ (Bayesian paired $t$-test $P_{MCMC} = 0.86$; Fig. 6d, bottom). Crucially, just as in the human experiments, the degree of discrimination accuracy was calibrated to be identical in all cases, and thus our results do not depend on different levels of accuracy across the ANNs (Fig. 6d, top; Bayesian paired $t$-test for all pairwise comparisons $P_{MCMC} > 0.51$). Moreover, we found that this result held independent of the degree of complexity (that is, size) of the downstream network, indicating that downstream circuits cannot compensate for the lack of fitness-maximizing codes at the encoding stage.

The findings from our ANN analysis clarify our human behaviour results. Our ANN analyses reveal how a fixed set of physical sensory inputs with relevant but hidden environmental/contextual statistics that the agent can only experience and learn over time are represented in coding schemes to maximize fitness. Moreover, studying ANNs with a VIB is useful because it provides a reasonably realistic model of how encoding schemes are adapted to optimize a given objective function when the resources to process information are limited.

## Generalizing to other ecologically valid reward functions
The solutions to the two decision-making objectives studied here belong to the same family of power-law efficient codes with a single parameter that determines the solution of the decision objective. However, we acknowledge that when the system must deal with more complex sensory–reward mappings, the same analytical solutions might not generally apply. This would have to be tested case by case. Nevertheless, it is possible to go beyond the analytical solutions and employ the same framework to find general strategies of resource allocation for arbitrary stimulus–reward mapping functions. We address this possibility next.

**Non-monotonic stimulus–reward functions.** As has been emphasized in previous work, non-monotonic payoff functions are common[10,12,50]. For instance, suppose that a physical attribute is related to the degree of salinity of food. Too little or too much salt can have deleterious consequences on an organism and its fitness. In such scenarios, it has been suggested that perception should not be tuned directly to the stimulus–reward associations, as the organisms will be able to know only how good or bad the payoff is[50]. However, knowing that the payoff is bad provides no information about why it is bad and hence no clue to the adaptive course of action for an organism[50].

How should the neural resources be allocated in such cases? What strategy should the agent follow? We consider the following three scenarios. Scenario 1 corresponds to the accuracy maximization task ($K_{acc}$). Scenario 2 corresponds to a reward-maximizing task where rewards are linearly (monotonically) mapped to the physical stimulus values ($K_{rew}$, Fig. 7a). Scenario 3 corresponds to a non-monotonic mapping where stimuli in the middle of the sensory space deliver the highest reward values (Fig. 7b). In all three scenarios, we assume a right-skewed distribution of sensory stimuli over the physical value space (Fig. 7a,b) for comparison with the orientation experiments we conducted in humans. For scenario 3, there is no known closed-form solution to find the optimal allocation of resources, but please note that the minimization objective remains the same as for scenario 2: minimize the reward given up for every erroneous decision.

Here we emphasize that a key assumption of our framework is that, on the basis of experience or understanding of task instructions, the agent clearly understands which stimuli deliver more reward. For scenarios 1 and 2, higher levels of the stimuli are preferred. In scenario 3, however, stimuli in the middle are preferred. We can examine the downstream decoding process to understand how resources at encoding should be allocated in each scenario. Recall that the decoding rule

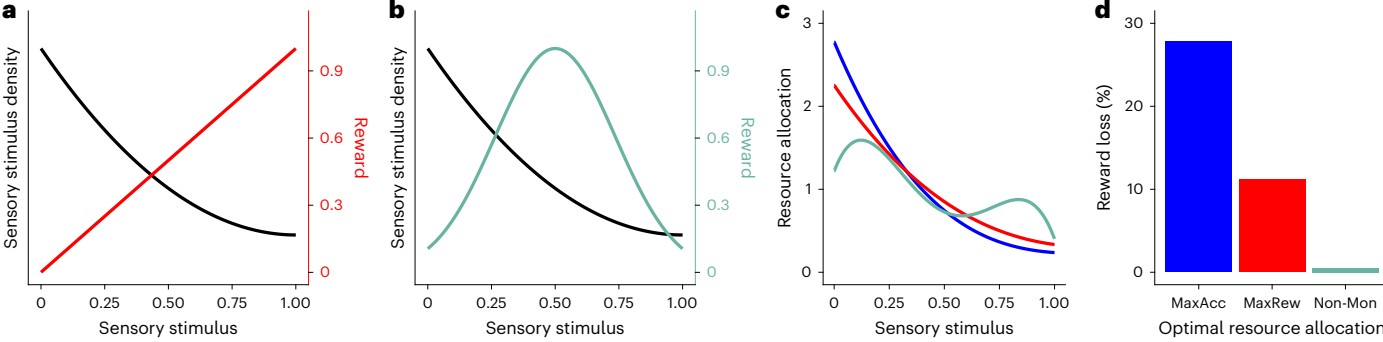

**Fig. 7 | Studying efficient allocation of neural resources with non-monotonic stimulus–reward mappings. a,b,** The prior distribution of sensory stimuli in the environment monotonically decreases with sensory stimuli (black) and is the same in all scenarios. The stimulus–reward mapping function in scenario 2 monotonically increases following a linear relationship (red, **a**) and in scenario 3 is non-monotonic with the highest reward delivered at $s = 0.5$ (green, **b**). Scenario 1 corresponds to the accuracy maximization context—that is, any correct decision yields the same amount of reward. **c,** Optimal solutions of the resource allocation problem for scenario 1 (blue), scenario 2 (red) and scenario 3 (green). **d,** Percentage reward lost in scenario 3 assuming that the agent uses the optimal resource allocations from $K_{acc}$ (MaxAcc, blue), $K_{rew}$ (MaxRew, red) or environments relative to the optimal solution in this non-monotonic stimulus–reward mapping environment (Non-Mon, green).

in our model is the same in all cases: the Bayesian mean squared error. What is a possible strategy for the cases where the stimulus–reward mappings are monotonic or non-monotonic unimodal? A relatively simple strategy that preserves the 'veridicality' of sensory information is one in which the agent employs a categorization threshold $\tau$ over the space of physical stimuli and decodes the values $\hat{s}$ relative to that threshold. A simple implementation is one where the agent computes a relative decoded value $\tilde{s} = -|\tau - \hat{s}(s_0)|$, an operation that could be flexibly implemented in downstream circuits. The choice rule is then choose $s_1$ if $\tilde{s}_1 > \tilde{s}_2$; otherwise, choose $s_2$. Thus, in addition to finding the optimal resource allocation function, the threshold $\tau$ is another latent variable to solve the reward maximization problem.

Before solving the optimization problem numerically, we note that the predictions for $\tau$ are relatively intuitive. In scenarios 1 and 2, $\tau$ should be set to the maximum stimulus value in the physical space, and the optimal resource allocation solutions remain the same as derived in our manuscript. In scenario 3, with reward values peaking in the middle of the stimulus distribution, the threshold will probably be located at $\tau \approx 0.5$ in our example in the encoding low-noise limit (not precisely at 0.5 due to biases and variance of $\hat{s}$ and the related influence of the prior distribution of physical stimuli).

The numerical solutions of resource allocation for scenarios 1 and 2 resemble, as expected, the analytical solutions in which more resources are allocated to regions of the physical space with the highest physical prior density. The amount of information is larger for lower sensory values in scenario 1 but larger for higher sensory values in scenario 2. For scenario 3, the solution for the categorization threshold is $\tau \approx 0.5$, and the resource allocation solution may be surprising and perhaps at first counterintuitive (Fig. 7c). Taking a closer look at the problem, we see that the solution indeed makes sense. First, we observe that the allocation of resources has a general trend to decrease as the sensory stimulus gets larger, thus following the expected result given the shape of the prior distribution of sensory stimuli. Second, the resource allocation solution has a dip at around $s = 0.5 \approx \tau$. This may appear initially counterintuitive given that these are the regions where the reward is the highest. However, note that (1) randomly drawing choice sets from this non-monotonic prior distribution is more likely to generate choice sets that are close in value than in the monotonic reward or accuracy scenarios, and (2) choice sets $s_{1,2}$ with values close to $s = 0.5$ are more likely to generate 'mistakes' given the resource allocation in Fig. 7c, but there is often little reward loss because the value function is relatively flat and symmetric (for example, $s_1 = 0.52$ and $s_2 = 0.48$ deliver the same reward). It is thus not worth investing too many resources near $s = 0.5$

even if the reward promised at those locations is high, because the potential for reward loss is low (Fig. 7d). We emphasize that this example is just one alternative strategy, but one that generates interesting predictions that could be tested in future experiments.

**Efficient resource allocation under reaction time costs.** We used simulations to study the scenario in which agents are rewarded/penalized for short/long reaction times (RTs) in both the $K_{acc}$ and $K_{rew}$ contexts. The goal was to study whether and how resource allocation changed relative to the accuracy maximization task without RT costs. Examining this scenario requires assumptions about a process model that jointly generates decisions and RTs. For simplicity and illustration purposes, we assumed that decisions and RTs were generated by a simple drift-diffusion model (DDM) with a constant decision bound $b$, decision evidence $z$ and diffusion noise $\sigma$ that was independent of the choice set inputs, which can be thought of as a downstream decision noise (Methods).

In this scenario, the loss function for the $K_{acc}$ context is given by

$$\iint_S f(s_1, s_2)(P(\text{error}|s_1, s_2) + \eta E[\text{RT}|s_1, s_2]), \quad (5)$$

and the loss function for the $K_{rew}$ context is given by

$$\iint_S f(s_1, s_2)\left(P(\text{error}|s_1, s_2)|s_1 - s_2| + \eta E[\text{RT}|s_1, s_2]\right), \quad (6)$$

where $\eta$ is the cost per RT unit (for example, in seconds). Note that as $\eta \to 0$, the optimal decision bound would be $z \to \infty$. Thus, the goal was to find the optimal balance between resource allocation and bound $z$ that minimizes the loss in equations (5) and (6) for a given RT cost $\eta$ and prior distribution of sensory stimuli in the environment $f(s)$.

The numerical solutions revealed that the resource allocation solutions in context $K_{acc}$ differed from the RT-cost-free scenario and depended on RT costs (Fig. 8a). While the RT-cost solutions were similar to the RT-cost-free solution for relatively high values of $\eta$, the smaller the RT costs, the more the allocation of resources tended to flatten. As expected, we found that higher $\eta$ resulted in lower $b$ (Fig. 8b). Second, in context $K_{rew}$, the RT-cost solutions were remarkably similar to the RT-cost-free solution. However, contrary to the $K_{acc}$ environment, the RT-cost solutions in $K_{rew}$ appeared to get steeper as the RT cost decreased (at least in the range of RT costs studied here; Fig. 8c). Once again, in context $K_{rew}$, higher $\eta$ resulted in lower $b$ (Fig. 8d).

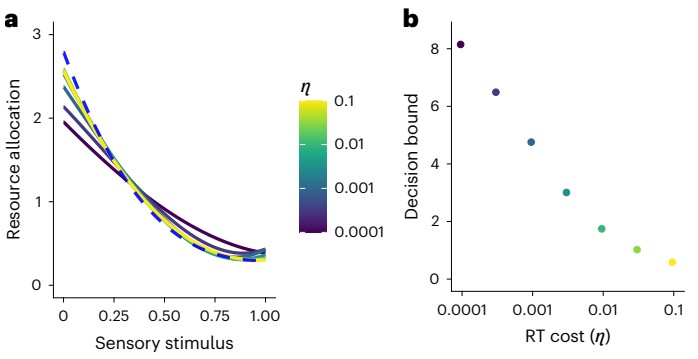

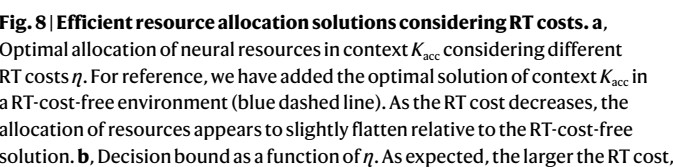

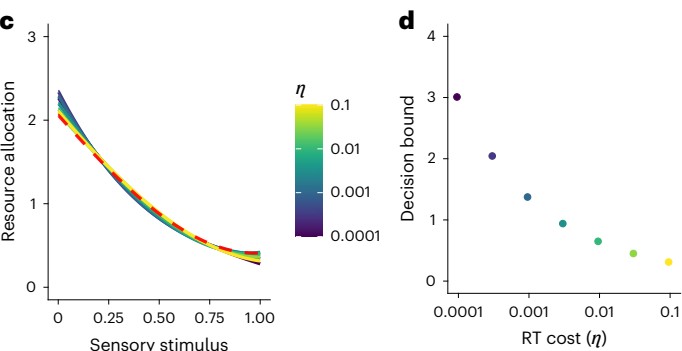

**Fig. 8 | Efficient resource allocation solutions considering RT costs. a**, Optimal allocation of neural resources in context $K_{acc}$ considering different RT costs $\eta$. For reference, we have added the optimal solution of context $K_{acc}$ in a RT-cost-free environment (blue dashed line). As the RT cost decreases, the allocation of resources appears to slightly flatten relative to the RT-cost-free solution. **b**, Decision bound as a function of $\eta$. As expected, the larger the RT cost,

the smaller the decision bound. **c,d**, Same as **a,b**, but this time in context $K_{rew}$. For reference, we have added the optimal solution of context $K_{rew}$ in an RT-cost-free environment (red dashed line). While the RT-cost solutions are similar to the RT-cost-free solution, contrary to the $K_{acc}$ environment, the RT-cost solutions appear to get steeper as the RT cost decreases.

We emphasize that the results presented here are based on a simple DDM with constant bounds. The resource allocation solutions may slightly differ for DDMs where the bounds are allowed to collapse or drift/diffusion parameters dynamically change over time[51–53]. This will be an interesting aspect to investigate in future research. Irrespective of these considerations, we show how the general framework developed here generates a rich set of testable predictions that allow for falsification and further refinement of the theory.

## Discussion

Our theoretical and empirical results provide evidence that early stages of sensory processing encode environmental stimuli to maximize fitness and not necessarily to maximize perceptual accuracy. We have shown this to be the case in humans and artificial agents with sensory processing bottlenecks. Our findings indicate that downstream circuits do not need to continuously compute reward distributions on the basis of stimulus–outcome associations because this information should be efficiently embedded in the neural codes of sensory perception. This notion is supported by recent studies showing that functional remapping of stimulus–reward contingencies in early sensory areas causally depends on top-down control signals from prefrontal structures[17,18,54]. We argue that this gives the organism the advantages of preventing information loss and rapidly transmitting behaviourally relevant information encoded by early sensory systems to downstream circuits specialized in action, learning or decision-making.

Efficient sensory adaptation to behavioural goals can arise without long-lasting synaptic changes or rewiring. Specific fitness-maximizing codes may have a structural basis if the environment and behavioural goals are stable over long periods, as may be the case for retinal contrast coding in the blowfly. However, efficient filtering of sensory information can rapidly occur via mechanisms of top-down contextual modulation of sensory processing, which can be achieved via mechanisms such as top-down attentional normalization[55]. In fact, it has been shown that adaptation to behaviourally relevant sensory statistics (such as edge orientations) can occur in the course of one hour in human participants[56]. Our key argument is that irrespective of whether efficient coding occurs via structural, synaptic or online top-down contextual modulations, it must occur at early stages if it is to be relevant for goal-directed behaviour. Information theory predicts that inefficient coding in regard to behavioural goals will cause a loss of relevant information that cannot be recovered in noisy transmission channels such as the brain. Our experiments with ANNs provide direct empirical evidence for this prediction by showing that restricting the initial encoding scheme to one that maximizes information causes

suboptimal performance in specific contexts. Overall, a key contribution of our work is that we provide a formal justification of why and how neural recoding should occur across contexts in capacity-constrained and noisy transmission systems to maximize reward and fitness.

We found additional supporting evidence for this hypothesis when re-analysing data from a recent human functional MRI (fMRI) study[57]. Specifically, we investigated whether novel goal-directed actions that promote people's 'survival' in hypothetical scenarios they had never before encountered triggered an efficient reorganization of perceptual codes in the human brain (Supplementary Note 2). Our analyses revealed that switching back and forth between survival goals that required participants to use the same items in very different ways led the brain to efficiently represent sensory information in a goal-specific manner. More specifically, novel behavioural goals that relied on object recognition caused changes in stimulus representations at early stages of sensory processing. Regions showing changes in stimulus representation codes included V1–V3 as well as downstream object detection areas such as the lateral occipital cortex (LOC) (Supplementary Note 2). We note that these results do not explicitly support the quantitative theory developed here but instead provide support for the general idea that a system should employ resources in its early sensory areas to represent abstract behavioural goals. In addition, these results do not imply that V1–V3 and LOC are discarding veridical feature information and instead represent only goal-oriented values. Although veridicality might be compromised (resources are finite), strategies might be implemented to ensure that it is not entirely suppressed (for example, disentangling via orthogonalization[19,20]).

Our study has some limitations and also generates interesting predictions that should be addressed in future research. First, the analytical solutions are restricted to accuracy maximization in discrimination tasks and reward maximization in the standard and most studied economic problem where properties of a good or action scale linearly with value. We acknowledge that when the system must deal with more complex sensory–reward mappings, the analytical solution to the resource allocation problem may not exist in a tractable form. Nevertheless, we provided some hints as to how the system can adapt to non-monotonic solutions with the use of categorization thresholds. Second, we acknowledge that our theory does not explain the dynamics of adaptation but generates predictions once the system has adapted after learning from experience. It thus remains unclear what the normative algorithms of efficient adaptation might be and how these could be connected with a biologically plausible algorithm that applies to arbitrary stimulus–reward association contexts such that reward expectation is maximized. Third, for the problems of accuracy and

reward maximization with linear sensory–reward mappings, our model predicts that in edge cases where the prior distribution is approximately flat, the optimal solutions are indistinguishable and the agent should allocate resources equally across the whole sensory space in both cases. Fourth, with regard to the previous point, an additional prediction appears worthy of future testing: if the prior density is low for low sensory values as well as for high sensory values, and there is a linear stimulus–reward mapping across the whole sensory space, our model predicts that, relative to the standard accuracy maximization task, sensitivity should also increase for low sensory values. Our model predicts that this effect should become more pronounced during a reward maximization task than during a standard discrimination task (a hint of this prediction can be found in Supplementary Fig. 1).

Beyond the obvious relevance for biological organisms, our results may have important implications in ongoing developments in artificial intelligence as well. Recent deep generative models show a remarkable ability to encode high-dimensional signals into latent factors under the objective of accurately predicting the local environment with specific encoding constraints. However, on the basis of our results, such an optimization objective will not necessarily match those present in biological organisms. Interestingly, a recent successful artificial intelligence model[58] proposed instead that representation formation should be driven by the need to predict the motivational value of experiences accurately. Our results validate this notion and imply that the development of artificial intelligence algorithms that aim to resemble neurobehavioural functions should go beyond the objective of maximizing only the accurate transmission of information and account for the motivational aspects of the environment that enable the organism (or the artificial agent) to maximize fitness.

Finally, although drawn from a different domain of behaviour, our results lend substantial support to economic theories positing that context-dependent utility functions should maximize expected reward rather than the expected accuracy of decisions guided by reward[3,39,59]. The corroborating evidence presented in our work grounded on the principles of neural coding and decision behaviour should help advance the development and refinement of these theories within economics and related disciplines of evolutionary biology and social sciences[12,60,61].

## Methods

### Participants

The participants were recruited by the Center for Neuroeconomics at the University of Zurich, Switzerland. The participants were instructed about all aspects of the experiment and gave written informed consent. None of the participants suffered from any neurological or psychological disorder or took medication that interfered with participation in our study. The participants received fixed monetary compensation for their participation in the experiment, in addition to a variable monetary payoff that depended on task performance (see below). The experiments conformed to the Declaration of Helsinki, and the experimental protocol was approved by the Ethics Committee of the Canton of Zurich.

Participants who failed to follow the eye fixation instructions on more than 25% of trials were excluded from the data analysis ($n = 12$). We measured the performance of the participants in the training tasks and excluded participants who were unable to perform the task at the easiest difficulty level ($n = 11$). Additionally, we had to exclude three participants due to technical problems with the data collection. The final sample thus comprised $n = 86$ participants ($n = 25$ in experiment 1 and $n = 61$ in experiment 2 (30 in $K_{rew}$)).

### Experimental design and stimuli

The stimuli were generated with MATLAB (version 9.7)[62], using the Psychtoolbox and displayed on a screen that was one metre away from the participants. The angle of the head was kept stable with a chin rest. The height of the chin rest was adjusted to position the centre of the

screen at the height of the eyes. As stimuli, we used oriented Gabor patches, presented on a grey background. Each patch was composed of a high-contrast three-cycles-per-degree sinusoidal grating convoluted with a circular Gaussian with width 0.41° and subtended 2.98° vertically and 2.98° horizontally. In experiment 1, all Gabor patches were presented so that the centres fell 5.7° to the left or right of the centre of the monitor and on the horizontal midline. In experiment 2, the Gabor centres fell 4.7° to the left and right of the vertical midline and 4.7° above or below the horizontal midline.

**Eye tracking.** Eye-tracking data were acquired using an ST Research Eyelink 1000 eye-tracking system. Gaze position was sampled at 500 Hz. Eye movements away from fixation were computed for the window corresponding to the stimulus presentation. For every saved position, the absolute distance to the fixation cross was computed. If the absolute distance exceeded 4° of visual angle, the trial was marked to include an eye movement. For most participants, the average number of trials with eye movements was less than 5%. Participants ($n = 12$) who made eye movements that exceeded 4° of visual angle on more than 25% of trials were excluded from all analyses.

**Experiment 1.** The participants performed the experiment in multiple sessions to allow for training within the two contexts on different days. The order of the accuracy ($K_{acc}$) and reward ($K_{rew}$) context training was counterbalanced across participants. In total, every participant completed 240 trials in the estimation task and 400 trials in the decision task.

**Experiment 2.** In experiment 2, each participant trained in only one stimulus–reward association context (either $K_{acc}$ or $K_{rew}$). Training in the binary judgement decision task was performed either in the two upper locations or in the two lower locations. The participants were randomly allocated to one of the two training locations. In the estimation tasks before and after the training task, the trial locations were evenly distributed between all four possible locations. In total, every participant completed 400 trials in the estimation task and 360 trials in the decision task.

**Orientation estimation task.** Before the start of every trial, the participants had to fixate on a cross in the middle of the screen. At the beginning of the trial, an arrow appeared for 0.5 seconds to indicate on which side the stimulus would be shown. Afterwards, the stimulus appeared on the indicated side for 0.6 seconds. The orientation of the stimulus was determined randomly within (0–179°). During stimulus presentation, the participant had to continue fixating on the cross. After the stimulus disappeared, a Gabor patch appeared in the middle of the screen. By pressing and holding the left mouse button, the participant then rotated the new Gabor patch until its perceived orientation matched the orientation of the previously observed target stimulus. The participant could end the trial by pressing the space key. After five seconds, the trial ended automatically. The trials were separated by a random intertrial interval of 1.5–2 seconds. The estimation task took place before and after the decision task (see below and Fig. 2). To avoid the possibility that participants developed contextual strategies, they were not informed in advance that a second estimation task was going to take place after the decision task.

**Decision task.** The fixation cross turned black to indicate the start of a trial. After 0.5 seconds, two Gabor patch stimuli appeared. The orientation of one of the stimuli was drawn from the approximate distribution of edges in the real world[42]. The orientation of the second stimulus was adjusted by a participant-specific difficulty score to keep performance at approximately 75% accuracy for all participants. The median accuracy across participants in $K_{rew}$ was 77 ± 2.9% and in $K_{acc}$ was 77 ± 2.8%. Additionally, on the basis of (1) calibration to 75% accuracy, (2) the linear mapping between the degree of diagonality

and reward (that is, from 1 Swiss franc (CHF) for 0° to CHF 46 for 45° in the diagonality space), and (3) pilot data, we adjusted the payoff of correct trials in $K_{acc}$ to match the expected payoff in $K_{rew}$. We calculated that setting the payoff for each correct response in $K_{acc}$ to 15 CHF would fulfil these conditions. Our experimental data were in line with these calculations: the median payoff in $K_{acc}$ was 15.00 ± 0 CHF, and in $K_{rew}$ it was 14.70 ± 0.62 CHF.

On average, the stimulus orientation followed a prior distribution $f(s)$ described by equation (7) and shown in Fig. 3a:

$$f(s) = \frac{1}{1.85 - \cos(4s)}. \tag{7}$$

The stimuli were displayed for 0.6 seconds. During stimulus presentation, the participants had to fixate on the cross in the middle of the screen. When the stimuli disappeared, the participants had 2.5 seconds to decide which stimulus was more oblique. Independent of the RT, the full 2.5 seconds had to be waited out. Afterwards, the two stimuli were shown again in their positions, and the result of the choice and the orientations of the stimuli were displayed for 3 seconds until the trial ended. The trials were separated by a 1.5-to-2-second intertrial interval.

### Blowfly retinal LMC experiment

Here we provide a brief description of the data collected in Laughlin's seminal work[36], which we re-analyse in this work. To derive the prior for the sensory stimulus of interest $f(s)$, the researcher measured the distribution of contrasts that occur in woodland settings of the blowfly environment. In brief, photographs were taken in the natural habitat of the blowfly such as sclerophyll woodland and lakeside lands. Relative intensities were measured across these scenes using a detector that scanned horizontally, like the ommatidium of a turning fly. The scans were digitized at intervals of 0.07° and convolved with a Gaussian point spread function of half-width 1.4°, corresponding to the angular sensitivity of a fly photoreceptor. Contrast values were obtained by dividing each scan into intervals of 10, 25 or 50°. Within each interval, the mean intensity ($\bar{I}$) was found and subtracted from every data point to give the fluctuation about the mean ($\Delta I$). This difference value was divided by the mean to give the contrast ($\Delta I / \bar{I}$).

These data were used to construct a histogram, which was later transformed to a CDF (Fig. 1a and Supplementary Fig. 1). Here we used this CDF to reconstruct the probability density function $f(s)$ (Supplementary Fig. 1). Once the prior distribution was obtained, the fly was placed in front of a screen with a light-emitting diode (LED). At the beginning of each trial, the LED luminance was set to the screen luminance and then changed to a new luminance drawn from the prior distribution $f(s)$ for 100 ms. The stimulus $s$ was defined as the proportional change of the difference between the background and LED luminances. We emphasize that the CDF of the contrast statistic comes directly from the contrast measurement methodology described in the preceding paragraph and reported by Laughlin. We thus did not make the original calculations for the prior $f(s)$, nor is it influenced by the fitness-maximizing sensory coding theory.

### Fitness-maximizing neural codes

In this section, we provide a detailed description of the connection between the $L_p$ reconstruction error, the efficient code that maximizes reward expectation and the power-law efficient codes briefly described in the main text.

Suppose that the stimulus distribution is given by $s \sim f(s)$. The function that transforms the input $s$ to neural responses $r$ is given by $r = h(s)$. While the mapping $h(s)$ is deterministic, here we assume that errors in the neural response $r$ follow a distribution $P[r|h(s)]$. We apply a general approach that considers optimality criteria accounting for how well stimulus $s$ can be reconstructed ($\hat{s}$) from the neural representations $r$. Wang and colleagues introduced a general formulation of the efficient coding problem in terms of minimizing the error in such reconstructions $\hat{s}(r)$ according to the $L_p$ norm as a function of the norm parameter $p$ (ref. 63). In brief, the reconstruction is assumed to be based on the maximum likelihood estimate of the decoder in the low-noise regime, where $P[r|h(s)]$ is assumed to be Gaussian distributed.

The goal is to find the optimal mapping function $h^*(s)$ to achieve a minimal $L_p$ reconstruction error for any given prior stimulus distribution $f(s)$. More formally, the problem is defined as: find $h^*(s)$ such that

$$\min \left\langle |\hat{s}(r) - s|^p \right\rangle_{s,r} \quad \text{s.t.} \ 0 \le h(s) \le 1, \tag{8}$$

where, without loss of generality, we assume that the operation range of the neuron is bounded between 0 and 1. It is possible to show that the optimal mapping $h^*(s)$ is given by equation (9)[63]:

$$h^*(s) = \frac{\int_{-\infty}^{s} f(\tilde{s})^{1/(1+p)} \, d\tilde{s}}{\int_{-\infty}^{\infty} f(\tilde{s})^{1/(1+p)} \, d\tilde{s}}. \tag{9}$$

If we define

$$\gamma \equiv 1/(1+p), \tag{10}$$

we observe that the normalized power function of the stimulus distribution $f$ in equation (9) is the escort distribution with parameter $\gamma$ (ref. 64). Note that under this framework, infomax coding is given by the norm parameter $p \to 0$, and therefore $\gamma = 1$, thus leading to the result that $h(s)$ is the CDF of the prior distribution.

**Efficient $L_p$ error-minimizing codes and behavioural goals.** Economics has a long tradition of studying the following problem: for a given distribution $f(s)$ in the environment, what is the optimal shape of the internal representation (that is, $h(s)$, which in economics is known as the utility function) if such function can only take a large but limited set of $n$ discrete subjective values (that is, the internal readings, $r$) that code for any given stimulus $s$ (refs. 3,39)? The utility function is thus restricted to a set of step functions with $n$ jumps, each corresponding to a utility increment of size $1/n$. In this case, discrimination errors originate from the fact that the organism cannot distinguish two alternatives located at the same step of the utility function. Under this formulation, the following variant of the problem was studied: find the optimal utility function ($h^*$) under two evolutionary optimization criteria, (1) the probability of mistakes minimization criterion and (2) the expected reward loss minimization criterion.

To solve this problem, we assume that the organism repeatedly makes choices between two alternatives drawn from the stimulus distribution $f(s)$, where we may suppose that stimuli are linearly mapped to a reward value. The organism is endowed with a utility function that assigns a level of reward to each possible stimulus $s$ from $f(s)$. The alternative that promises more utility to the organism is chosen[39].

If the goal of the organism is to minimize the number of erroneous responses (that is, maximize discrimination accuracy), the optimal utility function $h^*_{accuracy}$ is given by

$$h^*_{accuracy}(s) = \int_{-\infty}^{s} f(\tilde{s}) \, d\tilde{s}. \tag{11}$$

According to this solution, the power parameter of the escort distribution in equation (9) is given by $\gamma = 1$, which corresponds to the infomax strategy.

However, if the goal of the organism is to minimize the expected reward loss (that is, maximize the amount of reward received after many decisions) and stimuli are linearly mapped to reward value, the optimal utility function $h^*_{reward}$ is given by

$$h^*_{\text{reward}}(s) = \frac{\int_{-\infty}^{s} f(\tilde{s})^{2/3} \, d\tilde{s}}{\int_{-\infty}^{\infty} f(\tilde{s})^{2/3} \, d\tilde{s}}. \tag{12}$$

According to this solution, the power parameter of the escort distribution in equation (9) is given by $\gamma = 2/3$, which corresponds to optimizing the $L_p$ minimization problem with parameter $p$ given by

$$\gamma = 2/3 = \frac{1}{1+p} \quad \Rightarrow \quad p = 0.5. \tag{13}$$

We found that this normative fitness-maximizing solution is the error penalty that best describes the LMC data[40] (these results are reported in the main text and Fig. 1). Additionally, please note that the solutions provided in equations (11) and (12) are derived on the basis of maximizing the accurate choices and reward expectation, respectively, without any assumptions about maximizing information efficiency as a goal in itself.

**Connection to power-law efficient codes.** We employed a general method for defining efficient codes by investigating optimal allocation of Fisher information $J$ given (1) a bound of the organism's capacity $c$ to process information, (2) the frequency of occurrence $f(s)$ and (3) the organism's goal (for example, maximize perceptual accuracy or expected reward) according to

$$\underset{J(s)}{\arg\max} - \int ds \, f(s) J(s)^{-\alpha} \tag{14}$$

subject to a capacity bound

$$C(s) = \int ds \, J(s)^{\beta} \le c, \tag{15}$$

with parameters $\alpha$ defining the coding objective and $\beta > 0$ specifying the capacity constraint[43]. The solution of this optimization problem reveals that Fisher information should be proportional to the prior distribution $f(s)$ raised to a power $q$, which is therefore referred to as the power-law efficient code

$$J_{\text{opt}}(s) = c^{1/\beta} \left( \frac{f(s)^{\gamma}}{\int ds f(s)^{\gamma}} \right)^{1/\beta} \triangleq k f(s)^{q}, \tag{16}$$

where $q = 1/(\beta + \alpha)$ and $\gamma = \beta/(\beta + \alpha)$. Note that power-law parameter $q$ is multiply determined, and to make progress in identifying it, we need to make some further assumptions. Here we opted for setting $\beta = 0.5$, as previously proposed in the standard infomax framework[41]; however, our conclusions are not affected by the specific value of $\beta$. This means that $\alpha$ determines how Fisher information is allocated relative to the prior, influencing the values of both $q$ and $\gamma$. It can be shown that the infomax coding rule implies $\gamma = 1$ and therefore an efficient power-law code $q = 2$, and the reward expectation rule implies $\gamma = 2/3$ and therefore an efficient power-law code $q = 4/3$ (Supplementary Note 1). The power-law efficient codes thus allow us to establish a connection between behavioural goals in the contexts studied in this work ($K_{\text{acc}}$ and $K_{\text{rew}}$) and parameter $\gamma$, which incorporates the goals of the organism under the resource-constrained framework that we study here.

**Optimal inference.** When specifying an inference problem using such an encoding–decoding framework, a key aspect for generating predictions of decision behaviour is to obtain expressions of the expected value and variance of the noisy estimations $\hat{s}$ for a given value input $s_0$. However, we first need to specify the encoding and decoding rules. We adopted an encoding function $P(r|s)$ associated with the power-law efficient code that is parameterized as Gaussian[43]

$$P(r|s) = \mathcal{N}\left(s, \frac{1}{kf(s)^q}\right)$$
$$= \sqrt{\frac{kf(s)^q}{2\pi}} \exp\left(-\frac{kf(s)^q}{2}(r-s)^2\right), \tag{17}$$

and therefore Fisher information is allocated using an $s$-dependent variance $\sigma^2 = 1/kf(s)^q$. While we are aware that in our study the stimulus space is circular, given that discriminability thresholds are relatively low for orientation discrimination tasks in humans, it is safe to assume that the likelihood function can be locally approximated as a Gaussian distribution.

At the decoding stage, the observer computes the posterior using Bayes's rule:

$$P(s|r) = \frac{P(r|s)f(s)}{P(r)}. \tag{18}$$

Theoretical and empirical evidence suggests that for orientation estimation tasks, estimates are typically biased away from the prior. This suggests that humans employ an expected value estimator of the posterior, at least for the infomax case[41].

The expected value of the estimator can be defined as the input stimulus $s_0$ plus some average bias $b(s_0)$. Using analytical approximations under the high-signal-to-noise regime, it is possible to show that the bias for the posterior expected value estimator can be approximated by[65]

$$b(s_0) \approx \left(1 - \frac{1}{q}\right) \frac{1}{k} \left(\frac{1}{f(s)^2}\right)'_{s_0}. \tag{19}$$

In a previous study, using model simulations and exploring parsimonious functional forms, it was shown that the proportionality constant of the bias term can be approximated by[43]

$$\frac{\log(q)}{k\sqrt{q}}. \tag{20}$$

The analytical solution and the simulation-based solution of the proportionality constant are approximately equivalent for a range of $q$ values relevant to our work (for example, $q \in [0.5, 2]$); that is

$$\frac{\log(q)}{k\sqrt{q}} \cong \left(1 - \frac{1}{q}\right) \frac{1}{k}, \tag{21}$$

thus validating the results derived in the analytical approximations that we used in the current work. However, using either function does not affect the qualitative or quantitative results in our study.

Using this result, the expected value of the estimators is given by

$$E[\hat{s}|s_0] \approx s_0 + \left(1 - \frac{1}{q}\right) \frac{1}{k} \left(\frac{1}{f(s)^q}\right)'_{s_0}. \tag{22}$$

As already defined in the description of the behavioural task, in this study, we used a parametric form of the prior that closely resembles the shape of the natural distribution of orientations in the environment[42]

$$f(s) = \omega \times \frac{1}{a - \cos(4s)}, \tag{23}$$

with $a > 1$ determining the elevation (steepness) of the prior, and $\omega$ a normalizing constant. Using this parameterization of the prior, we can obtain an explicit analytical approximation of the bias:

$$b(s_0) \approx \left(1 - \frac{1}{q}\right) \frac{1}{k} \frac{\partial}{\partial s} \left(\left(\frac{\omega}{a - \cos(4s)}\right)^{-q}\right)_{s_0}$$

$$\approx \left(1 - \frac{1}{q}\right) \frac{1}{k} \left(\frac{4q \sin(4s)\left(\frac{\omega}{a - \cos(4s)}\right)^{1-q}}{\omega}\right)_{s_0}. \tag{24}$$

We can also obtain an analytical approximation of the variance under the high-signal-to-noise regime using the Cramer–Rao bound formulation:

$$\mathrm{Var}\,[\hat{s}|s_0] \propto \left(\frac{1}{J(s)}\right)_{s_0}$$

$$\approx \frac{1}{k}\left(\frac{1}{f(s)^q}\right)_{s_0} \tag{25}$$

$$\approx \frac{1}{k}\left(\frac{(a - \cos(4s))}{\omega}\right)^q_{s_0}.$$

We can thus use equations (24) and (25) to derive the predictions presented in Fig. 3.

Finally, assuming that the estimators are normally distributed using the expected value and variance derived above, the probability that an agent chooses an alternative with orientation value $s_1$ over a second alternative with orientation value $s_2$ (recall that in our experiment the decision rule (objective) of the participants is to choose the orientation perceived as closer to the diagonal orientation) is given by

$$P(\hat{s}_1 > \hat{s}_2|s_1, s_2) = \Phi\left(\frac{\mathrm{E}\,[\hat{s}_1|s_1] - \mathrm{E}\,[\hat{s}_2|s_2]}{\sqrt{\mathrm{Var}\,[\hat{s}_1|s_1] + \mathrm{Var}\,[\hat{s}_2|s_2]}}\right), \tag{26}$$

where $\Phi()$ is the CDF of the normal distribution. When fitting the choice data to the model, we accounted for potential side (left/right) biases $\beta_0$ and lapse rates $\lambda$ in the decision task using

$$P(\hat{s}_1 > \hat{s}_2|s_1, s_2) = \frac{\lambda}{2} + \Phi\left(\frac{\mathrm{E}\,[\hat{s}_1|s_1] - \mathrm{E}\,[\hat{s}_2|s_2]}{\sqrt{\mathrm{Var}\,[\hat{s}_1|s_1] + \mathrm{Var}\,[\hat{s}_2|s_2]}} + \beta_0\right)(1 - \lambda). \tag{27}$$

**Fitting the power-law efficient model to human data.** To fit the power-law efficient coding model to the choice data from the decision task, we used a hierarchical Bayesian model. We fit the early (1–200) and late (>200) training trials in each reward context separately. Posterior inference of the parameters in the hierarchical models was performed via the Gibbs sampler using the Markov chain Monte Carlo technique implemented in JAGS[66], assuming flat priors for both the mean and the noise of the estimates. For each model, we drew a total of 20,000 burn-in samples and subsequently took 5,000 new samples from three independent chains. We applied a thinning of 5 to this final sample, thus resulting in a final set of 3,000 samples for each parameter. We conducted Gelman–Rubin tests for each parameter to assess convergence of the chains. All latent variables in our Bayesian models had $\hat{R} < 1.05$, which suggests that all three chains converged to a target posterior distribution. We checked convergence of the group-level parameter estimates via visual inspection.

**Behavioural and statistical analyses**
In the estimation task, the observers' behavioural error on a given trial was computed as the difference between the reported orientation and the presented orientation. The direction of the error was defined as positive if the reported orientation was more oblique than the presented orientation, or negative if vice versa. If the error on any given trial was bigger than 25% of the maximum possible error (90 degrees), we discarded that trial. To make full use of the data, we pooled all participants from both experiments for the analysis of the impact of the reward training context. Comparisons between trained and untrained locations used only the data from the location-specific training in experiment 2.

We computed the average bias and variance in five bins of 9° before and after the training phases. Next, we computed the average change in the variance in each bin for each participant. We used the changes in variance within the most cardinal and most oblique bins to test for the predicted interactions between diagonality and training type ($K_{acc}$ or $K_{rew}$) or location (trained or untrained) using Bayesian hierarchical linear regressions implemented with the brms package (version 2.13.5)[67] in the statistical computing software R (version 3.6.3)[68]. For each model, we used four chains with 2,000 samples per chain after burn-in. The $P_{MCMC}$ values reported for these regressions represent one minus the probability of the reported effect being greater (less) than zero given the posterior distributions of the fitted model parameters.

We also compared the performance of participants in the binary judgement decision task using Bayesian hierarchical regressions implemented with brms in R. In this task, the participants had to decide which of two stimuli were more diagonal (closer to 45 degrees). We compared the accuracy of these decisions as a function of diagonality, training phase (early or late) and training type ($K_{acc}$ or $K_{rew}$). We used four chains with 1,000 samples per chain after burn-in for a total of 4,000 posterior samples for each regression parameter. The $P_{MCMC}$ values were computed in the same fashion as described above for the estimation task.

**ANNs**
Suppose that we have a dataset of $\mathbf{x}$ samples from a distribution of images represented by the retina where each image indicates an angular orientation $s$ with an angular prior distribution $p(s)$. Note that a key feature of our analyses is that knowledge about this angular prior is not explicitly given to the neural network; this prior is embedded in the statistics of image occurrences over space and time. Also note that there might be different images $\mathbf{x}_s$ that can be mapped to the same angle $s_0$ (for example, a Gabor patch with identical angle but different angular phases). Each stimulus is encoded by a set of latent codes (or a latent neural distributional code) $\mathbf{z}$ with a prior distribution $p(\mathbf{z})$. This prior distribution results in a posterior distribution $p(\mathbf{z}|\mathbf{x})$ after observing image $\mathbf{x}$. The neural coding system should thus learn a good representation of the environment (the distribution of physical sensory inputs) that might also need to be optimized for a particular downstream task (for example, maximize the reward consumption resulting from decision $y$). More specifically, we propose a VIB-like objective function (equation (4) in main text). In our ANN, the VIB-like objective trades (an approximation of) the amount of 'visual' information $I$ that the encoder can process with the expected reward loss, via the regularization parameter $\beta$. Higher values of $\beta$ thus introduce extra pressures in the network to encode information about the input image that can yield the most significant improvement in the downstream objective function. The neural network received two retinal images corresponding to screen locations where the two Gabor patches were presented in our task. We note that when training the ANN, the parameters of the encoder $\phi$ are shared for both retinal locations where the stimuli $\mathbf{x}_{1,2}$ are presented. The decision rule that the neural network has to learn is to indicate which of the two input stimuli (left or right) is more diagonal, while maximizing the reward received across many trials. Also like in the human experiments, we trained networks in two contexts: $K_{acc}$ and $K_{rew}$. For all VIB-like objectives studied here, we define the regularized 'information transmission' $I$ as

$$I \equiv \mathbb{E}_\mathbf{X}\,[D_{KL}\,(p_\phi(\mathbf{z}_1|\mathbf{x}_1)\,\|\,p(\mathbf{z}_1))$$
$$+ D_{KL}\,(p_\phi(\mathbf{z}_2|\mathbf{x}_2)\,\|\,p(\mathbf{z}_2))], \tag{28}$$

where $D_{KL}$ is the Kullback–Leibler divergence. In context $K_{acc}$, the reward loss in the VIB-like objective is defined as

$$E[\,\mathrm{reward\ loss}\,] \equiv \mathbb{E}_{p_\phi(\mathbf{z}|\mathbf{x})}\,[y(\mathbf{x}_1, \mathbf{x}_2)(1 - p_\theta(y = 1|\mathbf{z}_1, \mathbf{z}_2))$$
$$+ (1 - y(\mathbf{x}_1, \mathbf{x}_2))p_\theta(y = 1|\mathbf{z}_1, \mathbf{z}_2)], \tag{29}$$

with $y = 1$ when the correct response is given by stimulus input $\mathbf{x}_1$, and $y = 0$ otherwise. $p(y = 1|\mathbf{z}_1, \mathbf{z}_2)$ is the probability that the network chooses $\mathbf{x}_1$ given the encoding vectors $\mathbf{z}_{1,2}$.

In context $K_{rew}$, the reward loss in the VIB-like objective is defined as

$$
\begin{aligned}
E[\text{ reward loss }] \equiv \mathbb{E}_{p_\phi(\mathbf{z}|\mathbf{x})} & [|s(\mathbf{x}_1) - s(\mathbf{x}_2)| \\
& \times \{y(\mathbf{x}_1, \mathbf{x}_2)(1 - p_\theta(y = 1|\mathbf{z}_1, \mathbf{z}_2)) \\
& + (1 - y(\mathbf{x}_1, \mathbf{x}_2))p_\theta(y = 1|\mathbf{z}_1, \mathbf{z}_2)\}],
\end{aligned}
\tag{30}
$$

which is identical to the reward loss in the $K_{acc}$ VIB-like objective function, except that the probability of an erroneous ANN decision is weighted by the absolute value of the difference in the cardinality values $s(\mathbf{x}_1)$ and $s(\mathbf{x}_2)$. The ANNs trained with VIB-like objective functions thus penalize reward loss following the $K_{acc}$ and $K_{rew}$ objectives employed in the analytical solutions (see equations (1) and (2) in the main text).

All networks tested here used layers that are standard in the machine learning literature. Each retinal input network consisted of convolutional 4 × 4 kernels, with a stride of two. In the results presented in this work, we used four filters, but we found that our results are largely insensitive to the number of filters used. We also investigated a fully connected input layer with different sizes (50–200 neurons), which led to nearly identical results and conclusions. The stochastic encoder has the form

$$
p(z|x) = \mathcal{N}\left(z|g_e^\mu(x), g_e^\Sigma(x)\right),
\tag{31}
$$

where $g_e$ is a fully connected layer that receives as input the output of the retinal layer, where $g_e$ outputs the $K$-dimensional mean vectors $\mu$ of $z$ as well as the $K \times K$ covariance matrix $\Sigma$. In the results presented here, we use $K = 4$, but our results are similar for a range of $K$ values from 2 to 16. We used the reparameterization trick to write $p(z|x)dz = p(\epsilon)$ $d\epsilon$, where $z = g(x, \epsilon)$ is a deterministic function of $x$ and the Gaussian random variable $\epsilon$. The noise is thus independent of the parameters of the network, and it is possible to take gradients that optimize the objective function in equation (4). The downstream integration network consisted of a fully connected network that receives as input the values of the noisy encoder $z$ for each retinal input. The size of this layer for the results presented here is 20, but the main conclusions of our analyses are insensitive to the size of this layer. Finally, the decision module was a single sigmoidal unit indicating the selection of the left or right stimulus. All hidden units used rectified-linear activations. The networks were trained with Adam optimization with a learning rate of 0.0001.

To compute the Fisher information of the encoder, we first generated 500 inputs for each orientation stimulus $s$ in the cardinality space from 0° to 45° in steps of 0.5°. We computed the empirical expected value vector

$$
\bar{\mathbf{z}}(s) = \mathbb{E}_{\mathbf{z} \sim p(\mathbf{z}|s)}[\mathbf{z}] = \mathbb{E}[\mathbf{z}|s].
\tag{32}
$$

By rescaling the responses $z_i(s)$ such that the noise has unit variance, without loss of generality, the Fisher information $J$ can be expressed as

$$
J(s) = \sum_{i=1}^n \bar{z}_i'(s)^2 = \left\|\frac{\partial \bar{\mathbf{z}}(s)}{\partial s}\right\|_2^2.
\tag{33}
$$

## Resource allocation with RT costs

We used simulations to study the scenario in which agents are rewarded for short RTs in both the $K_{acc}$ and $K_{rew}$ contexts. Examining this scenario requires assumptions about a process model that jointly generates decisions and RTs. We assumed that decisions and RTs $T$ are generated by a simple DDM with a constant decision bound $b$, decision evidence $z$ and diffusion noise $\sigma$ that is independent of the choice set inputs,

which can be thought of as a downstream decision noise. In the DDM, the data generation process does not change if we set, for instance, $\sigma$ to a constant. Here we set $\sigma = 1$. Following the notation in our work, we define the decision evidence $z(s_1, s_2)$ for the choice set $s_{1,2}$

$$
\begin{aligned}
z(s_1, s_2) &= \frac{|E[\hat{s}_1|s_1] - E[\hat{s}_2|s_2]|}{\sqrt{\text{Var}[\hat{s}_1|s_1] + \text{Var}[\hat{s}_2|s_2]}} \\
&= \frac{|E[\hat{s}_1|s_1] - E[\hat{s}_2|s_2]|}{\sqrt{\frac{1}{J(s_1)} + \frac{1}{J(s_2)}}},
\end{aligned}
\tag{34}
$$

where $J(s)$ is Fisher information, which determines resource allocation. To find the optimal resource allocation, we define

$$
J(s) \equiv k \times \tilde{f}(s),
\tag{35}
$$

with the property

$$
\int \tilde{f}(s)ds = 1, \quad \frac{d\tilde{F}}{ds} > 0,
\tag{36}
$$

where $\tilde{F}$ is defined as the CDF of $\tilde{f}$. Here we set $k$ sufficiently high such that the low-noise limit property holds, and we numerically find $\tilde{f}$ (ref. 69).

In the standard DDM, the probability of an erroneous response is given by (for simplicity, we approximate the normal CDF of equation (26) with the logit function corresponding to the analytical solution of the DDM; this approximation does not change the qualitative conclusions of our results)

$$
P(\text{ error }|s_1, s_2) = \frac{1}{1 + e^{2b \times z(s_1, s_2)}},
\tag{37}
$$

and the expected RT is given by[70]

$$
E[RT|s_1, s_2] = \frac{b}{z(s_1, s_2)} \tanh(b \times z(s_1, s_2)).
\tag{38}
$$

In this scenario, the loss function for the $K_{acc}$ context is given by equation (5) in the main text, and the loss function for the $K_{rew}$ context is given by equation (6) in the main text. Note that as $\eta \to 0$, the optimal decision bound would be $z \to \infty$. The goal is thus to find the optimal balance between resource allocation $J(s)$ and optimal bound $z$ that minimizes the loss functions for a given RT cost $\eta$ and for the prior distribution of sensory stimuli in the environment.

## Representational similarity analyses of human fMRI data

We conducted additional conjunction analyses on the whole-brain maps of representational similarity for identity and usefulness that were originally computed by Castegnetti and colleagues[57]. We obtained the thresholded (FWE $P < 0.05$) whole-brain maps from Castegnetti and colleagues and computed conjunctions between the identity and usefulness contrasts, as well as usefulness and independently defined masks of the LOC and primary visual areas V1–V3 to create the figure in Supplementary Note 2. The LOC mask was obtained from the fMRI meta-analysis tool Neurosynth (neurosynth.org) with the keyword 'Lateral Occipital Cortex' and thresholded at the Neurosynth default of $P < 0.01$ (FDR-corrected). The V1–V3 masks were extracted from the Julich-Brain Cytoarchitectonic Atlas and thresholded at 50% probability. The LOC and V1–V3 masks were then conjoined with the cluster-corrected statistical map of usefulness representations. For the full details about the fMRI data analyses, see Supplementary Note 2.

## Reporting summary

Further information on research design is available in the Nature Portfolio Reporting Summary linked to this article.

## Data availability

The behavioural data are available at https://osf.io/an274/. The LOC mask was obtained from the fMRI meta-analysis tool Neurosynth (neurosynth.org). The V1–V3 masks were extracted from the Julich-Brain Cytoarchitectonic Atlas (julich-brain-atlas.de)

## Code availability

The analysis code is available at https://osf.io/an274/.

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

## Acknowledgements

We thank N. Netzer, M. Woodford and A. Stocker for providing helpful feedback on the manuscript text. We thank M. Zurita and B. De Martino for sharing their human fMRI data for re-analysis. J.S. and S.D.B. acknowledge support from Marlene-Porsche Foundation scholarships for their PhD studies. This work was supported by a European Research Council (ERC) starting grant (ENTRAINER) to R.P. This project has received funding from the ERC under the European Union's Horizon 2020 research and innovation programme (grant agreement no. 758604). T.A.H. received support from the Swiss National Science Foundation (SNSF) (grant no. 32003B_166566). P.N.T. received support from the SNSF (grant nos 100019_176016 and 10001C_188878). The funders had no role in study design, data collection and analysis, decision to publish or preparation of the manuscript.

## Author contributions

P.N.T., T.A.H. and R.P. designed the study. J.S. and S.D.B. collected the data. J.S., S.D.B., T.A.H. and R.P. analysed the data. All authors interpreted the results and wrote the manuscript. P.N.T., T.A.H. and R.P. acquired the funding.

## Funding

## Competing interests

The authors declare no competing interests.

## Additional information

**Correspondence and requests for materials** should be addressed to Todd A. Hare or Rafael Polania.

# Reporting Summary

Nature Research wishes to improve the reproducibility of the work that we publish. This form provides structure for consistency and transparency in reporting. For further information on Nature Research policies, see our Editorial Policies and the Editorial Policy Checklist.

## Statistics

For all statistical analyses, confirm that the following items are present in the figure legend, table legend, main text, or Methods section.

| n/a | Confirmed | |
|---|---|---|
| ☐ | ☒ | The exact sample size ($n$) for each experimental group/condition, given as a discrete number and unit of measurement |
| ☐ | ☒ | A statement on whether measurements were taken from distinct samples or whether the same sample was measured repeatedly |
| ☐ | ☒ | The statistical test(s) used AND whether they are one- or two-sided<br>*Only common tests should be described solely by name; describe more complex techniques in the Methods section.* |
| ☐ | ☒ | A description of all covariates tested |
| ☒ | ☐ | A description of any assumptions or corrections, such as tests of normality and adjustment for multiple comparisons |
| ☐ | ☒ | A full description of the statistical parameters including central tendency (e.g. means) or other basic estimates (e.g. regression coefficient) AND variation (e.g. standard deviation) or associated estimates of uncertainty (e.g. confidence intervals) |
| ☒ | ☐ | For null hypothesis testing, the test statistic (e.g. $F$, $t$, $r$) with confidence intervals, effect sizes, degrees of freedom and $P$ value noted<br>*Give P values as exact values whenever suitable.* |
| ☐ | ☒ | For Bayesian analysis, information on the choice of priors and Markov chain Monte Carlo settings |
| ☐ | ☒ | For hierarchical and complex designs, identification of the appropriate level for tests and full reporting of outcomes |
| ☒ | ☐ | Estimates of effect sizes (e.g. Cohen's $d$, Pearson's $r$), indicating how they were calculated |

*Our web collection on statistics for biologists contains articles on many of the points above.*

## Software and code

Policy information about availability of computer code

| | |
|---|---|
| Data collection | Matlab 2019; Psychtoolbox-3 |
| Data analysis | R: A language and environment for statistical computing (version 3.6.3); R package brms (version 2.13.5); JAGS 4.3 |

For manuscripts utilizing custom algorithms or software that are central to the research but not yet described in published literature, software must be made available to editors and reviewers. We strongly encourage code deposition in a community repository (e.g. GitHub). See the Nature Research guidelines for submitting code & software for further information.

## Data

Policy information about availability of data

All manuscripts must include a data availability statement. This statement should provide the following information, where applicable:

- Accession codes, unique identifiers, or web links for publicly available datasets
- A list of figures that have associated raw data
- A description of any restrictions on data availability

Data availability statement is provided in the submitted manuscript.
The Lateral Occipital Cortex (LOC) mask was obtained from the fMRI meta-analysis tool Neurosynth (neurosynth.org)
V1-V3 masks were extracted from the Julich-Brain Cytoarchitectonic Atlas (julich-brain-atlas.de)

# Field-specific reporting

Please select the one below that is the best fit for your research. If you are not sure, read the appropriate sections before making your selection.

☐ Life sciences ☒ Behavioural & social sciences ☐ Ecological, evolutionary & environmental sciences

For a reference copy of the document with all sections, see nature.com/documents/nr-reporting-summary-flat.pdf

# Life sciences study design

All studies must disclose on these points even when the disclosure is negative.

| | |
|---|---|
| Sample size | *Describe how sample size was determined, detailing any statistical methods used to predetermine sample size OR if no sample-size calculation was performed, describe how sample sizes were chosen and provide a rationale for why these sample sizes are sufficient.* |
| Data exclusions | *Describe any data exclusions. If no data were excluded from the analyses, state so OR if data were excluded, describe the exclusions and the rationale behind them, indicating whether exclusion criteria were pre-established.* |
| Replication | *Describe the measures taken to verify the reproducibility of the experimental findings. If all attempts at replication were successful, confirm this OR if there are any findings that were not replicated or cannot be reproduced, note this and describe why.* |
| Randomization | *Describe how samples/organisms/participants were allocated into experimental groups. If allocation was not random, describe how covariates were controlled OR if this is not relevant to your study, explain why.* |
| Blinding | *Describe whether the investigators were blinded to group allocation during data collection and/or analysis. If blinding was not possible, describe why OR explain why blinding was not relevant to your study.* |

# Behavioural & social sciences study design

All studies must disclose on these points even when the disclosure is negative.

| | |
|---|---|
| Study description | We designed an experiment to test if human early visual areas adapt in a manner predicted by the theory of fitness maximization. The data is quantitative experimental, allowing to test direct qualitative predictions of the theory (see Figure 3). |
| Research sample | The sample consisted of young healthy volunteers (age range: 18-40) who were mostly University students/employees. Given that we wanted to test the hypothesis that early visual areas adapt to fitness-maximizing codes in the healthy brain, none of the participants suffered from any neurological or psychological disorder or took medication that interfered with participation in our study. |
| Sampling strategy | The sampling strategy was random.<br>The sample size in Experiment 1 was determined such that the final sample size after exclusion criteria (see below) consisted of at least 25 participants each with 200-400 successful trials per experiment. This sample size typically allows testing well-defined theories of behaviour in standard laboratory settings (e.g., Polania et al., 2019 Nature Neuroscience). The replication/extension sample size in Experiment 2 was chosen for the same reasons. |
| Data collection | Stimuli were generated with Matlab, using the Psychtoolbox and displayed on a screen using the following experimental protocol. Participants were sat one meter from the screen. The angle of the head was kept stable with a chin rest. The height of the chin rest was adjusted to position the center of the screen at the height of the eyes. As orientation stimuli, we used oriented Gabor patches, presented on a grey background.<br><br>Eye-tracking data was acquired using an ST Research Eyelink 1000 eye-tracking system. Gaze position was sampled at 500 Hz.<br><br>no one else was present besides the participants(s) and the researcher. The researcher was not blinded to experimental condition and/or the study hypothesis. |
| Timing | Data collected from the end of 2018 until March 2022 |
| Data exclusions | Participants who failed to follow the eye fixation instructions on more than 25\% of trials were excluded from the data analysis. We measured the performance of participants in the training tasks and excluded eleven that were unable to perform the task at the easiest difficulty level. Additionally, we had to exclude three participants due to technical problems with the data collection. Thus, the final sample comprised n=86 participants (n=25 in Experiment 1, n=61 in Experiment 2). |
| Non-participation | Participants were recruited according to the following inclusion cireterion:<br>None of the participants suffered from any neurological or psychological disorder or took medication that interfered with participation in our study.<br>No participant dropped out/declined participation |

| Randomization | Participants were randomly allocated to each of the experiments. Each experiment consisted of independent samples |
|---|---|

# Ecological, evolutionary & environmental sciences study design

All studies must disclose on these points even when the disclosure is negative.

| Study description | *Briefly describe the study. For quantitative data include treatment factors and interactions, design structure (e.g. factorial, nested, hierarchical), nature and number of experimental units and replicates.* |
|---|---|
| Research sample | *Describe the research sample (e.g. a group of tagged Passer domesticus, all Stenocereus thurberi within Organ Pipe Cactus National Monument), and provide a rationale for the sample choice. When relevant, describe the organism taxa, source, sex, age range and any manipulations. State what population the sample is meant to represent when applicable. For studies involving existing datasets, describe the data and its source.* |
| Sampling strategy | *Note the sampling procedure. Describe the statistical methods that were used to predetermine sample size OR if no sample-size calculation was performed, describe how sample sizes were chosen and provide a rationale for why these sample sizes are sufficient.* |
| Data collection | *Describe the data collection procedure, including who recorded the data and how.* |
| Timing and spatial scale | *Indicate the start and stop dates of data collection, noting the frequency and periodicity of sampling and providing a rationale for these choices. If there is a gap between collection periods, state the dates for each sample cohort. Specify the spatial scale from which the data are taken* |
| Data exclusions | *If no data were excluded from the analyses, state so OR if data were excluded, describe the exclusions and the rationale behind them, indicating whether exclusion criteria were pre-established.* |
| Reproducibility | *Describe the measures taken to verify the reproducibility of experimental findings. For each experiment, note whether any attempts to repeat the experiment failed OR state that all attempts to repeat the experiment were successful.* |
| Randomization | *Describe how samples/organisms/participants were allocated into groups. If allocation was not random, describe how covariates were controlled. If this is not relevant to your study, explain why.* |
| Blinding | *Describe the extent of blinding used during data acquisition and analysis. If blinding was not possible, describe why OR explain why blinding was not relevant to your study.* |

Did the study involve field work? ☐ Yes ☒ No

## Field work, collection and transport

| Field conditions | *Describe the study conditions for field work, providing relevant parameters (e.g. temperature, rainfall).* |
|---|---|
| Location | *State the location of the sampling or experiment, providing relevant parameters (e.g. latitude and longitude, elevation, water depth).* |
| Access & import/export | *Describe the efforts you have made to access habitats and to collect and import/export your samples in a responsible manner and in compliance with local, national and international laws, noting any permits that were obtained (give the name of the issuing authority, the date of issue, and any identifying information).* |
| Disturbance | *Describe any disturbance caused by the study and how it was minimized.* |

# Reporting for specific materials, systems and methods

We require information from authors about some types of materials, experimental systems and methods used in many studies. Here, indicate whether each material, system or method listed is relevant to your study. If you are not sure if a list item applies to your research, read the appropriate section before selecting a response.

## Materials & experimental systems

| n/a | Involved in the study |
|---|---|
| ☒ | Antibodies |
| ☒ | Eukaryotic cell lines |
| ☒ | Palaeontology and archaeology |
| ☒ | Animals and other organisms |
| ☐ ☒ | Human research participants |
| ☒ | Clinical data |
| ☒ | Dual use research of concern |

## Methods

| n/a | Involved in the study |
|---|---|
| ☒ | ChIP-seq |
| ☒ | Flow cytometry |
| ☒ | MRI-based neuroimaging |

# Antibodies

| | |
|---|---|
| Antibodies used | *Describe all antibodies used in the study; as applicable, provide supplier name, catalog number, clone name, and lot number.* |
| Validation | *Describe the validation of each primary antibody for the species and application, noting any validation statements on the manufacturer's website, relevant citations, antibody profiles in online databases, or data provided in the manuscript.* |

# Eukaryotic cell lines

Policy information about cell lines

| | |
|---|---|
| Cell line source(s) | *State the source of each cell line used.* |
| Authentication | *Describe the authentication procedures for each cell line used OR declare that none of the cell lines used were authenticated.* |
| Mycoplasma contamination | *Confirm that all cell lines tested negative for mycoplasma contamination OR describe the results of the testing for mycoplasma contamination OR declare that the cell lines were not tested for mycoplasma contamination.* |
| Commonly misidentified lines (See ICLAC register) | *Name any commonly misidentified cell lines used in the study and provide a rationale for their use.* |

# Palaeontology and Archaeology

| | |
|---|---|
| Specimen provenance | *Provide provenance information for specimens and describe permits that were obtained for the work (including the name of the issuing authority, the date of issue, and any identifying information).* |
| Specimen deposition | *Indicate where the specimens have been deposited to permit free access by other researchers.* |
| Dating methods | *If new dates are provided, describe how they were obtained (e.g. collection, storage, sample pretreatment and measurement), where they were obtained (i.e. lab name), the calibration program and the protocol for quality assurance OR state that no new dates are provided.* |

☐ Tick this box to confirm that the raw and calibrated dates are available in the paper or in Supplementary Information.

| | |
|---|---|
| Ethics oversight | *Identify the organization(s) that approved or provided guidance on the study protocol, OR state that no ethical approval or guidance was required and explain why not.* |

Note that full information on the approval of the study protocol must also be provided in the manuscript.

# Animals and other organisms

Policy information about studies involving animals; ARRIVE guidelines recommended for reporting animal research

| | |
|---|---|
| Laboratory animals | *For laboratory animals, report species, strain, sex and age OR state that the study did not involve laboratory animals.* |
| Wild animals | *Provide details on animals observed in or captured in the field; report species, sex and age where possible. Describe how animals were caught and transported and what happened to captive animals after the study (if killed, explain why and describe method; if released, say where and when) OR state that the study did not involve wild animals.* |
| Field-collected samples | *For laboratory work with field-collected samples, describe all relevant parameters such as housing, maintenance, temperature, photoperiod and end-of-experiment protocol OR state that the study did not involve samples collected from the field.* |
| Ethics oversight | *Identify the organization(s) that approved or provided guidance on the study protocol, OR state that no ethical approval or guidance was required and explain why not.* |

Note that full information on the approval of the study protocol must also be provided in the manuscript.

# Human research participants

Policy information about studies involving human research participants

| | |
|---|---|
| Population characteristics | See above |
| Recruitment | Participants were recruited via the online system recruitment platform of the University of Zurich.<br>There might be slight selection bias in our sample given that participants were recruited online and required access to a computer or mobile phone. However, given that we here we study basic sensory system mechanisms, we believe that this selection had no impact in the interpretation and conclusions derived in our work |
| Ethics oversight | The experiments conformed to the Declaration of Helsinki and the experimental protocol was approved by the Ethics Committee of the Canton of Zurich. |

Note that full information on the approval of the study protocol must also be provided in the manuscript.

# Clinical data

Policy information about clinical studies
All manuscripts should comply with the ICMJE guidelines for publication of clinical research and a completed CONSORT checklist must be included with all submissions.

| | |
|---|---|
| Clinical trial registration | *Provide the trial registration number from ClinicalTrials.gov or an equivalent agency.* |
| Study protocol | *Note where the full trial protocol can be accessed OR if not available, explain why.* |
| Data collection | *Describe the settings and locales of data collection, noting the time periods of recruitment and data collection.* |
| Outcomes | *Describe how you pre-defined primary and secondary outcome measures and how you assessed these measures.* |

# Dual use research of concern

Policy information about dual use research of concern

## Hazards

Could the accidental, deliberate or reckless misuse of agents or technologies generated in the work, or the application of information presented in the manuscript, pose a threat to:

No   Yes
- ☐ ☐ Public health
- ☐ ☐ National security
- ☐ ☐ Crops and/or livestock
- ☐ ☐ Ecosystems
- ☐ ☐ Any other significant area

## Experiments of concern

Does the work involve any of these experiments of concern:

No   Yes
- ☐ ☐ Demonstrate how to render a vaccine ineffective
- ☐ ☐ Confer resistance to therapeutically useful antibiotics or antiviral agents
- ☐ ☐ Enhance the virulence of a pathogen or render a nonpathogen virulent
- ☐ ☐ Increase transmissibility of a pathogen
- ☐ ☐ Alter the host range of a pathogen
- ☐ ☐ Enable evasion of diagnostic/detection modalities
- ☐ ☐ Enable the weaponization of a biological agent or toxin
- ☐ ☐ Any other potentially harmful combination of experiments and agents

# ChIP-seq

## Data deposition

- ☐ Confirm that both raw and final processed data have been deposited in a public database such as GEO.

- ☐ Confirm that you have deposited or provided access to graph files (e.g. BED files) for the called peaks.

| | |
|---|---|
| Data access links<br>*May remain private before publication.* | *For "Initial submission" or "Revised version" documents, provide reviewer access links. For your "Final submission" document, provide a link to the deposited data.* |
| Files in database submission | *Provide a list of all files available in the database submission.* |
| Genome browser session<br>(e.g. UCSC) | *Provide a link to an anonymized genome browser session for "Initial submission" and "Revised version" documents only, to enable peer review. Write "no longer applicable" for "Final submission" documents.* |

## Methodology

| | |
|---|---|
| Replicates | *Describe the experimental replicates, specifying number, type and replicate agreement.* |
| Sequencing depth | *Describe the sequencing depth for each experiment, providing the total number of reads, uniquely mapped reads, length of reads and* |

| Sequencing depth | *whether they were paired- or single-end.* |
| Antibodies | *Describe the antibodies used for the ChIP-seq experiments; as applicable, provide supplier name, catalog number, clone name, and lot number.* |
| Peak calling parameters | *Specify the command line program and parameters used for read mapping and peak calling, including the ChIP, control and index files used.* |
| Data quality | *Describe the methods used to ensure data quality in full detail, including how many peaks are at FDR 5% and above 5-fold enrichment.* |
| Software | *Describe the software used to collect and analyze the ChIP-seq data. For custom code that has been deposited into a community repository, provide accession details.* |

# Flow Cytometry

## Plots

Confirm that:

☐ The axis labels state the marker and fluorochrome used (e.g. CD4-FITC).

☐ The axis scales are clearly visible. Include numbers along axes only for bottom left plot of group (a 'group' is an analysis of identical markers).

☐ All plots are contour plots with outliers or pseudocolor plots.

☐ A numerical value for number of cells or percentage (with statistics) is provided.

## Methodology

| Sample preparation | *Describe the sample preparation, detailing the biological source of the cells and any tissue processing steps used.* |
| Instrument | *Identify the instrument used for data collection, specifying make and model number.* |
| Software | *Describe the software used to collect and analyze the flow cytometry data. For custom code that has been deposited into a community repository, provide accession details.* |
| Cell population abundance | *Describe the abundance of the relevant cell populations within post-sort fractions, providing details on the purity of the samples and how it was determined.* |
| Gating strategy | *Describe the gating strategy used for all relevant experiments, specifying the preliminary FSC/SSC gates of the starting cell population, indicating where boundaries between "positive" and "negative" staining cell populations are defined.* |

☐ Tick this box to confirm that a figure exemplifying the gating strategy is provided in the Supplementary Information.

# Magnetic resonance imaging

## Experimental design

| Design type | *Indicate task or resting state; event-related or block design.* |
| Design specifications | *Specify the number of blocks, trials or experimental units per session and/or subject, and specify the length of each trial or block (if trials are blocked) and interval between trials.* |
| Behavioral performance measures | *State number and/or type of variables recorded (e.g. correct button press, response time) and what statistics were used to establish that the subjects were performing the task as expected (e.g. mean, range, and/or standard deviation across subjects).* |

## Acquisition

| Imaging type(s) | *Specify: functional, structural, diffusion, perfusion.* |
| Field strength | *Specify in Tesla* |
| Sequence & imaging parameters | *Specify the pulse sequence type (gradient echo, spin echo, etc.), imaging type (EPI, spiral, etc.), field of view, matrix size, slice thickness, orientation and TE/TR/flip angle.* |
| Area of acquisition | *State whether a whole brain scan was used OR define the area of acquisition, describing how the region was determined.* |

Diffusion MRI ☐ Used ☐ Not used

## Preprocessing

**Preprocessing software**
*Provide detail on software version and revision number and on specific parameters (model/functions, brain extraction, segmentation, smoothing kernel size, etc.).*

**Normalization**
*If data were normalized/standardized, describe the approach(es): specify linear or non-linear and define image types used for transformation OR indicate that data were not normalized and explain rationale for lack of normalization.*

**Normalization template**
*Describe the template used for normalization/transformation, specifying subject space or group standardized space (e.g. original Talairach, MNI305, ICBM152) OR indicate that the data were not normalized.*

**Noise and artifact removal**
*Describe your procedure(s) for artifact and structured noise removal, specifying motion parameters, tissue signals and physiological signals (heart rate, respiration).*

**Volume censoring**
*Define your software and/or method and criteria for volume censoring, and state the extent of such censoring.*

## Statistical modeling & inference

**Model type and settings**
*Specify type (mass univariate, multivariate, RSA, predictive, etc.) and describe essential details of the model at the first and second levels (e.g. fixed, random or mixed effects; drift or auto-correlation).*

**Effect(s) tested**
*Define precise effect in terms of the task or stimulus conditions instead of psychological concepts and indicate whether ANOVA or factorial designs were used.*

**Specify type of analysis:** ☐ Whole brain  ☐ ROI-based  ☐ Both

**Statistic type for inference**
(See Eklund et al. 2016)
*Specify voxel-wise or cluster-wise and report all relevant parameters for cluster-wise methods.*

**Correction**
*Describe the type of correction and how it is obtained for multiple comparisons (e.g. FWE, FDR, permutation or Monte Carlo).*

## Models & analysis

| n/a | Involved in the study |
|-----|-----------------------|
| ☐ | ☐ Functional and/or effective connectivity |
| ☐ | ☐ Graph analysis |
| ☐ | ☐ Multivariate modeling or predictive analysis |

**Functional and/or effective connectivity**
*Report the measures of dependence used and the model details (e.g. Pearson correlation, partial correlation, mutual information).*

**Graph analysis**
*Report the dependent variable and connectivity measure, specifying weighted graph or binarized graph, subject- or group-level, and the global and/or node summaries used (e.g. clustering coefficient, efficiency, etc.).*

**Multivariate modeling and predictive analysis**
*Specify independent variables, features extraction and dimension reduction, model, training and evaluation metrics.*

