## [Peer Review File · Nature Human Behaviour]

Peer Review Information

Journal: Nature Human Behaviour

Manuscript Title: Sensory perception relies on fitness-maximizing codes

Corresponding author name(s): Todd A. Hare and Rafael Polania

Reviewer Comments & Decisions:

Decision Letter, initial version:

20th January 2022

Dear Dr Hare,

Thank you once again for your manuscript, entitled "Sensory perception relies on fitness-maximizing codes", and for your patience during the peer review process.

Your Article has now been evaluated by 3 referees. You will see from their comments copied below that, although they find your work of potential interest, they have raised quite substantial concerns. In light of these comments, we cannot accept the manuscript for publication, but would be interested in considering a revised version if you are willing and able to fully address reviewer and editorial concerns.

We hope you will find the referees' comments useful as you decide how to proceed. If you wish to submit a substantially revised manuscript, please bear in mind that we will be reluctant to approach the referees again in the absence of major revisions. We are committed to providing a fair and constructive peer-review process. Do not hesitate to contact us if there are specific requests from the reviewers that you believe are technically impossible or unlikely to yield a meaningful outcome.

To guide the scope of the revisions, the editors discuss the referee reports in detail within the team, including with the chief editor, with a view to (1) identifying key priorities that should be addressed in revision and (2) overruling referee requests that are deemed beyond the scope of the current study. We hope that you will find the prioritised set of referee points to be useful when revising your study. Please do not hesitate to get in touch if you would like to discuss these issues further.

1. All reviewers question the strength of evidence in support of your claims. In revision, we expect that your manuscript will provide additional human behavioural data following the specific recommendations by our reviewers. We also expect that your computational modelling would be substantially strengthened, following the recommendations by Reviewers 2 and 1 (and discussion of limitations, as requested by Reviewer 3). On the other hand, given the journal's focus on human behaviour, we would leave it up to you whether to provide more compelling fly electrophysiology data, which Reviewers 1 and 2 point out would be needed to support your claims. Additional fly data would certainly strengthen your argument, but are not a condition on considering a revised manuscript (you may want to reconsider, however, whether the inclusion of the existing blowfly data in your manuscript is warranted, given our reviewers' concerns about their strength).

2. All reviewers agree that more details regarding the model training, assumptions and limitations are needed. Please carefully revise your manuscript to provide detailed information on the underlying assumptions, the training and potential limitations of your models. In doing so, please ensure that the information reported is consistent across the whole manuscript.

3. Reviewer 1 and 3 raise novelty concerns. In response to their comments, we ask you to more clearly emphasize what the novel contribution of your work is. Specifically, we ask you to follow Reviewer 3's recommendation to engage with the existing literature and place your work in the context of this

knowledge.

4. In addition, Reviewer 1 and Reviewer 2 voice hesitations about the validity of the fitness-maximization approach. We ask that you revise your manuscript to explicitly describe how the fitness-maximization approach is different from the infomax approach and to which extent this approach can be expected to generalize to more ecologically valid reward functions.

5. Moreover, we ask that you follow Reviewer 2's suggestions to increase the strength of the evidence. We request that you show that the effects in trained area K_{acc} and K_{rew} for contexts follow the distinct predictions for fitness-maximization in each of these contexts. In addition, please include an analysis of reaction times or provide other data to show that low performance in the K_{acc} for oblique trials is not simply a result of participants choosing to ignore the most difficult trials in a context that rewards all trials equally.

6. Furthermore, to validate your findings and support your claims, please provide additional evidence from analyses in which the stage of the ANN was manipulated.

7. Please also carefully evaluate all other reviewer comments and revise your manuscript accordingly.

If you wish to submit a suitably revised manuscript we would hope to receive it within 6 months. We understand that the COVID-19 pandemic is causing significant disruptions which may prevent you from carrying out the additional work required for resubmission of your manuscript within this timeframe. If you are unable to submit your revised manuscript within 6 months, please let us know. We will be happy to extend the submission date to enable you to complete your work on the revision.

- Include a "Response to the editors and reviewers" document detailing, point-by-point, how you addressed each editor and referee comment. If no action was taken to address a point, you must provide a compelling argument. This response will be used by the editors to evaluate your revision and sent back to the reviewers along with the revised manuscript.
- Highlight all changes made to your manuscript or provide us with a version that tracks changes.

[REDACTED]

Thank you for the opportunity to review your work. Please do not hesitate to contact me if you have any questions or would like to discuss the required revisions further.

Sincerely,

Samantha Antusch

Samantha Antusch, PhD
Editor
Nature Human Behaviour

Reviewer expertise:

Reviewer #1: optimal coding ; Bayesian efficient coding ; DNN

Reviewer #2: optimal coding ; Bayesian efficient coding

Reviewer #3: DNN ; vision

REVIEWER COMMENTS:

Reviewer #1:

Remarks to the Author:

This paper suggests that sensory information is encoded in the brain by maximizing reward (or fitness), in contrast to the well-known infomax hypothesis. In support of this claim, the paper presents: (1) an analysis of previously published neural data from the blowfly's retina; (2) new behavioral data from psychophysical experiments in humans; and (3) simulations of ANNs trained to solve one of the tasks from the human experiment. Additionally, the appendix includes an analyses of human fMRI data from an object perception task.

I found this paper very interesting to read. It is well written (up to some minor issues detailed below), and the breadth of the analyses and results is impressive. The human behavioral experiments are clever, revealing that human sensory representations may adapt to external reward functions. I also greatly value the fact that this study aims to apply the same mathematical model to different types of data, which provides converging evidence. Therefore, I believe the topic and scope of this paper are appropriate for publication in Nature Human Behavior and could potentially make a great contribution. However, I also have several substantial concerns, as explained below, but I believe they could be addressed to a large extent with some revisions.

1. The analyses of the blowfly's data seems somewhat unconvincing, for two main reasons: First, it relies on a very specific (hand-picked) reward function, which has not been empirically validated. It is also unclear to what extent the results are robust to errors in that reward function. Second, if I'm not mistaken, the data from the blowfly's retina was collected without giving the fly any reward for its responses. Therefore, it is unclear why it makes sense to model these data as maximizing reward, and more generally, it is unclear what the paper's approach predicts in this case (i.e., what is the meaning of fitness-maximizing codes without some reward/fitness function?).

2. The paper proposes fitness-maximization as an alternative to infomax. In practice, however, the difference between infomax and fitness-maximizing codes boils down to varying one continuous variable, q . Therefore, the fitness-maximization approach seems more like a softer version of infomax rather than a qualitatively different approach. In addition, I may be wrong here, but it seems that the analytical derivation of the fitness-maximizing codes depends on strong assumptions about the reward function. If so, it is unclear to what extent this approach can generalize to more ecologically valid reward functions, that may be complex functions of both the agent's internal state and the external environment.

3. Related to my comments above, the behavioral experiment considers only two simple reward conditions in the decision task (K_{acc} and K_{rew}), and it's unclear to what extent the results would generalize to other reward functions. For example, can the same approach generalize to conditions in which the participants are also rewarded for short reaction times (e.g., by considering some weighted average between K_{acc} or K_{rew} and a reward for speed)?

In fact, consider the following thought experiment: suppose that participants in the decision task would only be rewarded for speed, regardless of their decisions. Intuitively, an adaptive fitness-maximizing code should support minimal reaction times in this case, which is achieved by making a decision without representing the Gabor patches at all. However, it is highly unlikely that this is what will happen in practice, suggesting that sensory neural representation may not be fully governed by fitness maximization. How does this fit with the overall premise of the paper?

4. There seems to be some confusion in the paper about deep variational information bottleneck (VIB). First, VIB is not a neural network architecture, as implied in the paper (e.g., p6, Fig 5), but rather a

combination of an objective function (bound on the information bottleneck objective) and a method for training deep nets to optimize it (similar to VAEs).

Second, there is a discrepancy in the paper about how the ANNs were trained. The main text (p6) states that they were trained with respect to K_{acc} or K_{rew} , but in the methods section there's a VIB-like objective function (eq. 23). In addition, eq. 23 seems ill-formed (or at least unclear) — where is the expectation over $p(y|z)$? Isn't the reward a function of y and x rather than a constant? How was β selected?

More generally, the VIB objective (or eq. 23) suggests an optimality criterion different from the fitness-maximization criterion (i.e., the optimization principle that yields eq. 7). That is, VIB trades (an approximation of) $I(X;Z)$ with the predictability of Y given Z (or reward, in eq. 23), whereas the fitness-maximization criterion does not take into account $I(X;Z)$. The paper, however, argues that sensory neural codes are governed by the fitness-maximization criterion, and does not mention VIB until the ANN section. Why, then, didn't the authors train the ANNs with respect to the same objective function they're arguing for? How would the fitness-maximization results compare with those obtained by VIB?

5. Is each ANN trained only on one context (K_{acc} or K_{rew})? If so, this seems like a major difference compared to the behavioral experiment, where the same participants adapt to different contexts. A more relevant simulation would be to train the same ANN on the two contexts and see if/how it adapts.

6. I view this study as part of a growing body of work on information—fitness tradeoffs in neuroscience, and not as an entirely new theory as seems to be implied in the paper. In particular, the framing (especially in the introduction) does not distinguish between traditional infomax efficient coding approaches and more recent approaches that consider efficient coding in the sense of rate-distortion theory or related principles like the information bottleneck (e.g., refs 6, 10, 13, and 14 in the paper) which can be seen as tradeoffs between minimizing coding errors and maximizing fitness / reward / performance in some task. This is quite jarring because some of the references in the introduction consider the information bottleneck principle, which the current paper also employs. It's also worth noting in this context that the information bottleneck has previously been applied to neural representations in the retina:

Palmer et al., Predictive information in neurons. PNAS, 2015.

Wang et al., Efficient encoding of motion is mediated by gap junctions in the fly visual system. PLOS Computational Biology, 2017.

In addition, the paper doesn't address two related lines of work that have also argued that sensory neural representations are governed by tradeoffs between information and reward:

Tkacik and Bialek, Information Processing in Living Systems. Annu. Rev. Condens. Matter Phys. 2016.

Friston, The free-energy principle: a rough guide to the brain? TiCS, 2009.

To clarify, this comment is not about adding more references to the paper. It is about explicitly drawing the connection between the current study and the broader literature on information—fitness tradeoffs in neuroscience and cognitive science. The paper already includes several key references in that area, however it does so without the proper context. In my view, giving the right context and drawing the connection mentioned above would increase the significance of paper and its appeal to a broader audience.

Minor comments:

7. It would be helpful if the authors could define precisely the reward functions that are used in each part of the main text. This detail is a very central for assessing the paper's claims.

8. Figure 3 is a big hard to digest, as it contains a lot of information and the caption is extremely long. Also, in 3b the color bar is reversed (starting at 4.75 and climbing up to 3.5), which is confusing.

9. Figure 5a is confusing: a neural architecture for the information bottleneck can be any network that implements an encoder $p(z|x)$ and a decoder $p(y|z)$. In other words, "comparison and decision" component is part of the information bottleneck. In addition, the text below the arrow, $p(z|x)$, seems misplaced.

10. p8: "Crucially, we show that no matter how complex, flexible, or unconstrained downstream decision circuits are, they cannot maximize fitness if the encoding step maximizes accuracy rather than reward consumption."

This statement seems way too strong, as the paper only investigates a few very specific settings.

11. On p11, after eq 5, I think alpha is a typo (shouldn't it be gamma?).

12. Also on p11, $h(s)$ is presented as a utility function, however earlier in the paper it is presented as the neural response.

Reviewer #2:

Remarks to the Author:

In this paper, the authors investigate the extent to which a fitness maximizing coding scheme in the earliest layers of sensory processing provides better predictions for behavior than classic information maximizing codes. As evidence they first present parameter free, fitness-maximizing model fits to LMC blowfly cells data which were motivated by normative models from economic theory. They find a code which is fitness-maximizing, here operationalized as a reward maximizing, is a better match to neural data than a purely information maximizing code. Next, they turn to humans and design a set of orientation discrimination experiments with reward feedback. Two distinct reward contingencies, K_{rew} and K_{acc} , allow the authors to differentiate between an information maximizing code which predicts constant behavior across reward contexts and a fitness maximizing code which predicts behavior adapts according to the effects of different reward contingencies. They find performance is modulated according to reward context in line with a code that optimally allocates neural resources in order to maximize rewards. Further, performance in a straightforward sensory estimation task and in a version of the task with inputs to specific retinotopic locations both show biases that indicate the early stages of sensory processing follow this coding scheme. Finally, the authors seek to validate these results in an ANN that solves the same task with an information bottleneck. They find that information in this bottleneck matches what one would expect based on humans' behavioral performance in each context. Further, they find that a sensory encoder trained to maximize decision accuracy fails to produce good performance on K_{rew} even when re-training the decision-making decoder. From this the authors conclude that reward maximizing adaptation of coding 'must occur at early stages if it is to be relevant for goal directed behavior.'

The authors present well designed experiments and indeed show that signatures of performance shift in ways that are predicted by a code which maximizes reward under information constraints. The claims that these changes must adhere at the earliest stages of sensory processing in order for the organism to benefit from such a coding scheme warrant further investigation.

In particular, the power of this argument from the ANN analysis would be significantly strengthened if the authors manipulated the stage of processing at which the information bottleneck occurs. In the paper, the authors place the bottleneck after 4 convolutional filters and before a fully connected decision layer. This bottleneck models information constraints at early sensory processing stages. However, in the presented architecture the information bottleneck could go even earlier, for example after the very first convolutional filter. This would provide a strong demonstration that systems with information bottlenecks at the very earliest phases of sensory processing, like e.g. blowfly LMCs, exhibit reward maximizing coding. Another architecture the authors should check in order to validate their results is one where the information bottleneck is moved to the decision-making layers in the network. In one potential architectural set up, the retinal layers would embed each image and subsequently a fully connected layer would process the concatenation of these embeddings and then pass through the information bottleneck. With the addition of this model we could determine more precisely whether the results in this section are

due to an information bottleneck at the earliest phases of sensory processing, or rather result from any information limited, reward maximizing code.

Another important argument from this section comes from the retraining analysis, where an encoder trained on decision-accuracy displays a large decrease in performance when the decision weights are retrained in the K_{rew} context. Presumably, this is because under information constraints the encoder chooses not to represent information about the least likely orientations i.e. the oblique orientations, and these are exactly the orientations which provide the highest reward in the K_{rew} . With all that in mind, it would be illuminating to show how the strictness of the information bottleneck, controlled by the β parameter in the loss function, affects the results in this section. How much reward can you recover by easing the bottleneck (i.e. reducing β) How do the curves of information content shown in 5c. shift as you let more and more information pass to the decision making layers? Finally, a few more details regarding model training would be useful. What performance criteria is achieved by each network trained in the K_{acc} and K_{rew} contexts?

The behavioral studies are well designed overall. For the version of the behavioral task where participants train in retinotopic specific areas, the authors show that estimation effects of training on are only apparent in the retinotopic regions where training occurred. This indeed suggests that training on reward affects early phases of sensory processing, but without a comparison of reward contexts it is difficult to definitely conclude that these changes reflect a reward maximizing code. In particular, the predicted reduction in the estimation bias shown in Fig. 4b. is also consistent overall increased precision in rewarded location. The data for the estimation variance is in the correct direction, but a key prediction of the fitness maximizing code - that there should be a significant reduction in variance for oblique directions after K_{rew} training - isn't immediately evident. In fact, fig 4c shows no decrease in variance after training compared to without training. To establish that early sensory effects here are truly fitness maximizing, and not simply a result of increased information coding in rewarded retinotopic areas, the authors would have to show that the effects in trained area K_{acc} and K_{rew} for contexts follow the distinct predictions for fitness-maximization in each context. As far as I can tell, this is not the case.

It would also be a good check to show that low performance in the K_{acc} for oblique trials (Fig. 3h.) aren't simply a result of participants choosing to ignore the most difficult trials in a context that rewards all trials equally. Some analysis of reaction times or other data that shows participants are equally engaged on these trials would be a nice addition.

Minor

Type: 'the the' below question 23

the y-axis in Fig 3. b-j was difficult to decipher at first. Consider simply labeling with degrees? Is it representing the orientation of the correct stimulus?

The b) label in Fig. 3 caption is missing.

Reviewer #3:

Remarks to the Author:

The paper marshals a wide array of evidence to address a fundamental conceptual question in neuroscience and perception: are neurons best thought of as encoding properties of the world, or the potential rewards that those properties might bring the observer? This topic has been hotly debated over the last decade, and the present paper may contribute a valuable method for deriving and comparing quantitative predictions made by the two hypotheses. It is well written and data are well presented and rigorously analysed. However, I have three main areas of concern: the lack of engagement with previous debate in this area; the assumptions required by the modelling framework; and the strength of the evidence for the claims. It is possible that some or all of these could be cleared up with further explanation from the authors.

MAIN COMMENTS

1. Relation to Hoffman et al.'s "Interface Theory"?

The current paper seems to be arguing for a similar interpretation of neural coding as Donald Hoffman and colleagues have put forward in several papers over the last decade under the term "interface theory". E.g. to compare a couple of quotes:

From the abstract of the current paper:

"We found that neural codes that maximize reward expectation -- and not accurate sensory representations -- account for retinal LMC activity."

From the abstract of Hoffman, Singh & Prakash (2015):

"We find that veridical perceptions -- strategies tuned to the true structure of the world -- are routinely dominated by nonveridical strategies tuned to fitness."

The above paper from Hoffman et al is cited, but only in passing in the Discussion, so it is hard to tell how this work relates to and deviates from Hoffman et al's work. Given that the main contribution of the current paper is an explicit computational framework for deriving what a "fitness maximising" neural code should look like in any given domain, it seems critical to explain how this differs from the computational frameworks developed by Hoffman et al. E.g. in the following papers:

- Prakash, C., Stephens, K. D., Hoffman, D. D., Singh, M., & Fields, C. (2021). Fitness beats truth in the evolution of perception. *_Acta Biotheoretica_, _69_(3)*, 319-341.
- Prakash, C., Fields, C., Hoffman, D. D., Prentner, R., & Singh, M. (2020). Fact, fiction, and fitness. *_Entropy_, _22_(5)*, 514.
- Hoffman, D. D., Singh, M., & Mark, J. (2013). Does evolution favor true perceptions?. In *_Human Vision and Electronic Imaging XVIII_* (Vol. 8651, p. 865104). International Society for Optics and Photonics.

The 2015 Psychonomic Bulletin & Review paper received a large number of commentaries, along with counter-responses from the authors. These extensive discussions on the topic of whether we perceive "fitness" or "reality" are very pertinent to the current paper. To pick just one point of contention, Interface Theory posits that perception departs from reality most clearly when fitness functions are non-monotonic (e.g. to achieve homeostasis, too much salt or sugar is a bad thing, but so is too little salt or sugar). Does the formalism in the current paper allow for non-monotonic fitness functions? My understanding is no; fitness is assumed to be monotonically or even linearly related to some underlying physical property? If the formalism can't encompass non-monotonic fitness functions, this seems to be a major shortcoming. And if it can, it should then address potential problems with that, e.g. the critique raised by Bart Anderson that directly perceiving fitness, in the case of a non-monotonic fitness function, means that the observer has discarded information about the direction in which it should adjust its behaviour:

- Anderson, B. L. (2015). Where does fitness fit in theories of perception?. *_Psychonomic bulletin & review_, _22_(6)*, 1507-1511.

Generally, the idea that we perceive fitness, rather than veridical properties of the world, has been extensively debated over the past decade, and it is important to situate this work within that debate. What is the novel contribution (e.g. perhaps explicit predictions about the shapes of neural response curves, which might be considered complementary to Interface Theory's game-theoretic simulations about population fitness)? How does the current proposal differ from that of Interface Theory, and how does it meet the critiques that have been previously raised about IT?

2. Assumptions and limitations of the modelling framework?

The main contribution of the paper is to present a quantitative way to derive and compare predicted neural response or psychometric functions, given the distribution of stimuli in the environment, under the assumptions that the perceptual system is either trying to maximise (a) ability to recover true stimulus value (e.g. orientation, contrast level etc), or (b) expected reward (assuming different values of the stimulus are associated with different reward levels). The solution is adapted from a 2009 economics paper on mental representation to support optimal choice-making, given limited representational resources. The application to neuroscience seems novel and is potentially valuable.

However, I would like to see a clearer presentation of the assumptions and limitations of the modelling framework. For example, it seems to assume a monotonic (p2) or perhaps even strictly linear (p11) mapping between stimulus values and reward values. If so, this seems like a major limitation. The need for homeostasis means that for most animals most physical properties (brightness, heat, salination, etc) likely have "sweet spots" at moderate levels, with very high or very low levels being detrimental. In the empirical example of flies seeking flowers by contrast levels, it doesn't seem at all obvious that the highest contrast levels are the most rewarding. The highest contrast levels in a natural environment are likely to be created by behaviourally-neutral things like any object silhouetted against the sky. Whereas the contrast differences between different flower species or other food sources are likely to occupy some range of moderate contrast levels. Can the framework accommodate such a possibility, and can it make testable predictions about which fitness functions best account for the neural code?

I may well be missing something crucial in the explanation, but it seemed that the transformation from the distribution of stimulus values to the predicted fitness-maximising neural/perceptual function happens *without specifying* any particular fitness function? If this is the case, does the framework make any predictions about how neural coding or perception should change when the fitness function in the environment changes? What happens at edge cases; for example, if a linear mapping from stimulus value to reward is sufficiently shallow, the difference between "veridical encoding" and "reward encoding" should presumably collapse?

3. Level of empirical evidence provided for the theoretical claims.

The paper makes a major claim: that neural encoding and perception are attuned not to the statistics of our environments, but rather to the distribution of rewards in our environments. They support it via empirical evidence from fly electrophysiology, human psychophysics, human fMRI, and deep neural network simulations. This array of methods and target systems is impressive and ambitious. However, the quality of evidence provided by each seems underwhelming.

The fly electrophysiology evidence consists of reanalysing a set of recordings from blowfly retinal cells from a (seminal) 1981 paper, and relating these to contrast measurements from natural blowfly habitats reported in the same paper, with the help of some simplifying assumptions (that higher-contrast stimuli have monotonically/linearly(?) higher reward value to blowflies). The fitness-maximising neural response curve derived does indeed better align with the cell recordings than does the classical interpretation in terms of maximising discrimination ability given the statistics of the environment. This is suggestive but not very strong evidence. Given that the differences between the two predictions are a matter of changing slope in a sigmoidal function, could there be other explanations (e.g. that there is a small mismatch between the contrast statistics of the woodlands and lakesides from which the measurements are taken, and the actual experienced contrasts of the experimental laboratory flies)? How plausible is the assumption that higher contrast levels have higher reward values for blowflies? A really compelling test of the hypothesis would involve manipulating both the environmental statistics and the reward mappings for new sets of flies, and showing that retinal responses reliably followed the reward mapping rather than the environmental statistics.

The human fMRI evidence is the weakest, and is raised only briefly in the Discussion. It shows only that task-relevant information (e.g. attention and task reward value) can modulate representations in early visual areas over the slow timescales of fMRI, which is consistent with conventional views on visual encoding (e.g. that specific object features can be amplified in V1 or even earlier via top-down attention). The paper seems to be arguing for a more radical interpretation than this - i.e. that V1-3 are discarding veridical feature information and instead representing only reward value? It would be good to spell out exactly how radical the argument is, and how it departs from widely held views of attention-as-gain-control that can flexibly amplify reward-relevant features.

The bulk of the empirical contribution of the paper describes two psychophysics experiments. They show that after receiving training in which oblique orientations are more rewarded than cardinal ones (vs a comparison group who receive equal rewards for all orientations), participants' discrimination accuracy improves for oblique orientations, and there is a reduction of the usual "oblique effect" (higher precision for cardinal than oblique orientations) in that their response variance in an adjustment task decreases for oblique and increases for cardinal orientations, and their cardinal bias decreases.

This all seems consistent with observers deploying greater attention and/or more stringent response

criteria towards the more task-rewarded orientations, and thereby improving discrimination and matching performance at those orientations at the expense of others. It has previously been shown that attention to specific orientations has both neural and perceptual consequences, e.g.:

- Liu, T., Larsson, J., & Carrasco, M. (2007). Feature-based attention modulates orientation-selective responses in human visual cortex. *Neuron*, *55*(2), 313-323.

It has also previously been reported that both attention and task-based rewards can modulate orientation adaptation, raising the possibility that different orientations were differently adapted during the repeated exposure of the discrimination task (which would also be consistent with the effect being retinotopically-localised):

- Festman, Y., & Ahissar, M. (2004). Attentional states and the degree of visual adaptation to gratings. *Neural Networks*, *17*(5-6), 849-860.

- Pascucci, D., & Turatto, M. (2013). Immediate effect of internal reward on visual adaptation. *Psychological Science*, *24*(7), 1317-1322.

The authors say in the Discussion that they are agnostic about the particular mechanisms by which neural codes come to maximise fitness, and that adaptation and attention are two plausible short-time-scale mechanisms. But this leaves me wondering what the new contribution of this paper is, since most would agree that rewarded stimuli attract attention, with concomitant benefits to perceptual and behavioural precision.

Is the novel contribution the normative framework for predicting exactly what form that should take? (If so, I would like to see more tests and discussions of how the framework can handle various and arbitrary reward mappings)

Or is the argument a more radical one, along the lines of Interface Theory? (If so, I would like to see more explanation of why the presented evidence isn't consistent with more standard views of neural codes that encode physical stimulus dimensions but whose sensitivity can be modulated by reward and relevance).

Author Rebuttal to Initial comments

Reviewer 1

This paper suggests that sensory information is encoded in the brain by maximizing reward (or fitness), in contrast to the well-known infomax hypothesis. In support of this claim, the paper presents: (1) an analysis of previously published neural data from the blowfly's retina; (2) new behavioral data from psychophysical experiments in humans; and (3) simulations of ANNs trained to solve one of the tasks from the human experiment. Additionally, the appendix includes an analyses of human fMRI data from an object perception task.

I found this paper very interesting to read. It is well written (up to some minor issues detailed below), and the breadth of the analyses and results is impressive. The human behavioral experiments are clever, revealing that human sensory representations may adapt to external reward functions. I also greatly value the fact that this study aims to apply the same mathematical model to different types of data, which provides converging evidence. Therefore, I believe the topic and scope of this paper are appropriate for publication in Nature Human Behavior and could potentially make a great contribution. However, I also have several substantial concerns, as explained below, but I believe they could be addressed to a large extent with some revisions.

Resp: Thank you for your positive feedback and constructive criticisms of our work which improved the presentation of our results, their interpretation, and implications for future research.

- 1. The analyses of the blowfly's data seems somewhat unconvincing, for two main reasons: First, it relies on a very specific (hand-picked) reward function, which has not been empirically validated. It is also unclear to what extent the results are robust to errors in that reward function. Second, if I'm not mistaken, the data from the blowfly's retina was collected without giving the fly any reward for its responses. Therefore, it is unclear why it makes sense to model these data as maximizing reward, and more generally, it is unclear what the paper's*

approach predicts in this case (i.e., what is the meaning of fitness-maximizing codes without some reward/fitness function?).

Resp: We agree with the reviewer that the qualitative analyses of the blowfly data should not be seen as definitive evidence supporting the fitness-maximizing hypothesis, as the evidence is mostly anecdotal. We partially acknowledged this in our original submission where we wrote (p. 2): *"While the predictions of reward-maximizing sensory codes and the data from blowfly retinal neurons (i.e., the earliest level of encoding) show a striking similarity (Fig. 1), this finding does not directly support all aspects of our hypothesis that neural codes in early sensory areas adapt to the organism's behavioral needs. This is because we don't know the specific function linking contrast to fitness for the blowfly, and we can't show that the code used in their retinas adapts between contexts because we only have data from one context"*. The only aspect that is conclusive is that the LMC code is not infomax but something else. However, we would like to emphasize that the reward function was not "hand-picked" but followed the simple assumption that stimulus magnitude and reward are linearly related as is commonly assumed in economic models. Our rationale for making use of the blowfly data was to use it as motivation to introduce our framework by adopting the simple assumption that there is a linear sensory-reward mapping. In discussing whether or not to include these data with several colleagues, we found that they all agreed that using the blowfly data is a good way to introduce the theoretical framework, as long as we make it clear to the reader that the evidence is merely illustrative. We agree with our colleagues and reviewer that conclusive statements about the results from this analysis should be toned down. We revised the text accordingly in the new version of the manuscript (abstract, introduction, results, and discussion), and the illustrative nature of the evidence is made clear to the reader at the moment of introducing the example. If the editor and reviewer feel strongly that it is better to delete this section from the manuscript, we will agree to do so. However, we hope you agree that introducing the framework with the blowfly data is a useful way to illustrate and introduce the topic and the theoretical framework. Additionally, we now make sure in the revised version of the manuscript that all the conclusions of our work rest on the human data and artificial neural network simulations.

2. *The paper proposes fitness-maximization as an alternative to infomax. In practice, however, the difference between infomax and fitness-maximizing codes boils down to varying one continuous variable, q . Therefore, the fitness-maximization approach seems more like a softer version of infomax rather than a qualitatively different approach. In addition, I may be wrong here, but it seems that the analytical derivation of the fitness-maximizing codes depends on strong assumptions about the reward function. If so, it is unclear to what extent this approach can generalize to more ecologically valid reward functions, that may be complex functions of both the agent's internal state and the external environment.*

Resp: Our main contribution is that we formally demonstrate how a (neural) system should allocate information processing resources for two of the most common problems studied in decision-making: Accuracy maximization in perceptual discrimination tasks, and reward maximization in the standard and most studied class of economic problems, i.e., choices in which a particular attribute is monotonically related to a given currency value. Therefore, our theory leads to predictions of how neural systems could generate "utility-like" functions (in addition to predicting sensory estimation and discrimination behavior) for these two important decision-making problems. As the reviewer points out, we found that the solution to this problem looks remarkably similar for both objectives. The solutions to both problems belong to the same family of power-law efficient codes with a single parameter that determines the solution of the decision objective. We believe that this simple and elegant link between the functions is part of what makes our result intriguing, and may partially explain why for some applications the infomax rule has been historically successful.

However, we acknowledge that when the system must deal with more complex sensory-reward mappings, the same analytical solutions might not generally apply. This would have to be tested on a case-by-case basis. It is possible to go beyond the analytical solutions and employ the same framework to find general strategies of resource allocation for arbitrary stimulus reward mapping functions, for instance, when the mapping is non-monotonic.

How should the neural resources be allocated for the case in which there is a non-monotonic mapping between physical stimulus values and rewards? What strategy should the agent follow? We consider the following three scenarios. Scenario 1 corresponds to the accuracy maximization task (K_{Acc} in our manuscript). Scenario 2 corresponds to a reward-maximizing task where rewards are linearly (monotonically) mapped to the physical stimulus values (K_{Rev} in our manuscript, Figure R1a). Scenario 3 corresponds to a non-monotonic mapping where stimuli in the middle of the sensory space deliver the highest reward values (Figure R1b). In all three scenarios, we assume a right-skewed distribution of sensory stimuli over the physical value space (Figure R1a,b) for comparison with the orientation experiments we conducted in humans. For scenario 3 there is no known

closed-form solution to find the optimal allocation of resources, but please note that the minimization objective remains the same as for Scenario 2

$$E[\text{loss}] = \int \int f(s_1, s_2) P(\text{error} | s_1, s_2) |R(s_1) - R(s_2)| ds_1 ds_2. \quad (1)$$

This expression does not depend on explicit payoffs of each choice alternative $s_{1,2}$ in absolute terms, but depends on the relative difference of the reward $R(s)$ delivered by two options in the choice set (independent of the range they are drawn from). Thus, irrespective of reward mapping our expression is always equivalent to minimizing the reward that is given up with every wrong decision.

Here, we emphasize that a key assumption of our framework is that, based on experience or understanding of task instructions, the agent has a clear understanding of which stimuli deliver more reward. For scenarios 1 and 2, sensory stimuli with higher values are preferred. In scenario 3, however, stimuli in the middle are preferred. We can examine the downstream decoding process to understand how the allocation of resources at encoding should be allocated in each scenario. Recall that the decoding rule in our model is the same in all cases: the Bayesian mean squared error (BMSE). What is the strategy for the cases where the stimulus-reward mappings are monotonic or non-monotonic unimodal? A relatively simple strategy that preserves the "veridicality" of sensory information is one in which the agent employs a "categorization" threshold τ over the space of physical stimuli and decodes the values s^\wedge relative to that threshold. A simple implementation is one where the agent computes a relative decoded value $s^\sim = -|\tau - s^\wedge(s_0)|$, an operation that could be flexibly implemented in downstream circuits. The choice rule is then choose s_1 if $s_1^\sim > s_2^\sim$, otherwise choose s_2 . Thus, in addition to finding the optimal resource allocation function, the threshold τ is an additional latent variable to solve the reward maximization problem.

Before solving the optimization problem numerically, we note that the predictions for τ are relatively intuitive. In scenarios 1 and 2, τ should be set to the maximum stimulus value in the physical space, and the optimal resource allocation solutions remain the same as derived in our manuscript. In scenario 3, with reward values peaking in the middle of the stimulus distribution, the threshold will be likely located at $\tau \approx 0.5$ in our example in the encoding low-noise limit (not precisely at 0.5 due to biases and variance of s^\wedge and the related influence of the prior distribution of physical stimuli).

The numerical solutions of resource allocation for scenarios 1 and 2 resemble, as expected, the analytical solutions in which more resources are allocated to regions of the physical space with the highest physical prior density, but the amount of information is larger for lower sensory values in scenario 1 and larger for higher sensory values in scenario 2. For scenario 3, the solution for the "categorization" threshold is $\tau \approx 0.5$, and the resource allocation solution is in principle surprising and perhaps at first counter-intuitive (Figure R1c). Taking a closer look at the problem, we will see that the solution indeed makes sense. First, we observe that the allocation of resources has a general trend to decrease as the sensory stimulus gets larger, thus following the expected result given the shape of the prior distribution of sensory stimuli. Second, the resource allocation solution has a dip at around $s = 0.5 \approx \tau$. This may appear initially counter-intuitive given that these are the regions where the reward is highest. However, note that (i) randomly drawing choice sets from this non-monotonic prior distribution is more likely to generate choice sets that are close in value compared to the monotonic reward or accuracy scenarios, and (ii) choice sets $s_{1,2}$ with values close to $s = 0.5$ are more likely to generate "mistakes" given the resource allocation in Figure R1c, but there is often little reward loss because the value function is relatively flat and symmetric (e.g., $s_1 = 0.52$ and $s_2 = 0.48$ deliver the same reward). Thus, it is not worth investing too many resources near $s = 0.5$ even if the reward promised at those locations is high, because the potential for reward loss is low.

Figure R1. Studying efficient allocation of neural resources with non-monotonic stimulus-reward mappings. a,b) The prior distribution of sensory stimuli in the environment monotonically decreases with sensory stimuli (black) and is the same in all scenarios. The stimulus-reward mapping function in Scenario 2 monotonically increases following a linear relationship (red, panel a), and in Scenario 3 is non-monotonic with the highest reward delivered at $s = 0.5$ (green, panel b). Note that in Scenario 1 any correct decision yields the same amount of reward. c) Optimal solutions of the resource allocation problem for Scenario 1 (blue), Scenario 2 (red), and Scenario 3 (green). d) Percentage reward lost in Scenario 3 assuming that the agent makes use of the optimal resource allocations from the accuracy maximization (K_{Acc} , blue), reward maximization (K_{Rew} , red) or environments relative to the optimal solution in this non-monotonic stimulus-reward mapping environment (green).

We emphasize that this example is just one alternative strategy, but one that generates interesting novel predictions that could be tested in future experiments. In the revised version of the manuscript, we not only discuss the advantages of our approach but also predictions that may confirm or falsify the theory, as well as potential limitations of our theory in the discussion section (p. 11). Moreover, we have also added the analyses and results presented above in the Results section (pp. 8-10), preserving most of the structure of our response in this letter, which we hope readers will find useful in guiding future research.

3. *Related to my comments above, the behavioral experiment considers only two simple reward conditions in the decision task (K_{Acc} and K_{Rew}), and it's unclear to what extent the results would generalize to other reward functions. For example, can the same approach generalize to conditions in which the participants are also rewarded for short reaction times (e.g., by considering some weighted average between K_{Acc} or K_{Rew} and a reward for speed)?*

In fact, consider the following thought experiment: suppose that participants in the decision task would only be rewarded for speed, regardless of their decisions. Intuitively, an adaptive fitness-maximizing code should support minimal reaction times in this case, which is achieved by making a decision without representing the Gabor patches at all. However, it is highly unlikely that this is what will happen in practice, suggesting that sensory neural representation may not be fully governed by fitness maximization. How does this fit with the overall premise of the paper?

Resp. We partially answered this question in our response to point 2, where we mentioned that the analytical solutions derived in our work hold for the cases of accuracy maximization and reward maximization in the common economic problem where value monotonically scales with a given currency value (e.g., money or calories). Regarding the extreme thought experiment proposed by the reviewer, if, evolutionarily speaking, organisms would not need to represent detailed visual information at all but simply react to, say, any visual flash and be rewarded in proportion to their response times, then we agree that decisions can be achieved without representing the Gabor patches at all and there is no need to maximize any coding objective. The premise of our paper rests on the idea that when the contents of sensory information are relevant to guide decisions, as is very often the case, then if the system has the possibility to reallocate its processing resources, it will be beneficial to do so given the fact that noisy communication channels such as the brain always lose information during transmission.

Using our framework, we studied the scenario proposed by the reviewer in which the participants are rewarded for short reaction times (RT) in both the K_{Acc} and K_{Rew} contexts. The goal is to study whether and how resources are allocated relative to the accuracy maximization task without RT costs. Examining this scenario requires assumptions about a process model that jointly generates decisions and response times. For simplicity and illustration purposes, we assume that decisions and response times T are generated by a simple drift-diffusion model (DDM) with a constant decision bound b , decision evidence z , and diffusion noise σ that is independent of the choice set inputs which can be thought of as a downstream decision noise. In the DDM the data generation process does not change if we set, for instance, c to a constant. Here we set $c = 1$. Following the notation in our

work, we define the decision evidence $z(s_1, s_2)$ for the choice set $s_{1,2}$

$$\begin{aligned} z(s_1, s_2) &= \frac{|E[\hat{s}_1 | s_1] - E[\hat{s}_2 | s_2]|}{\sqrt{\text{Var}[\hat{s}_1 | s_1] + \text{Var}[\hat{s}_2 | s_2]}} \\ &= \frac{|E[\hat{s}_1 | s_1] - E[\hat{s}_2 | s_2]|}{\sqrt{\frac{1}{J(s_1)} + \frac{1}{J(s_2)}}}, \end{aligned} \quad (2)$$

where, recall that, $J(s)$ is Fisher's information which determines resource allocation. In the standard DDM, the probability of an erroneous response is given by (note: for simplicity, we approximate the Normal CDF of Eq. 21 in the methods section of the original submission with the logit function corresponding to the analytical solution of the DDM. This approximation does not change the qualitative conclusions of our results)

$$P(\text{error} | s_1, s_2) = \frac{1}{1 + e^{2b \cdot z(s_1, s_2)}}, \quad (3)$$

and the expected RT is given by

$$E[RT | s_1, s_2] = \frac{b}{z(s_1, s_2)} \tanh(b \cdot z(s_1, s_2)). \quad (4)$$

In this scenario, the loss function for the K_{Acc} context is given by

$$E[\text{loss}] = \int \int f(s_1, s_2) (P(\text{error} | s_1, s_2) + \eta E[RT | s_1, s_2]) ds_1 ds_2, \quad (5)$$

and the loss function for the K_{Rew} context is given by

$$E[\text{loss}] = \int \int f(s_1, s_2) (P(\text{error} | s_1, s_2) |s_1 - s_2| + \eta E[RT | s_1, s_2]) ds_1 ds_2, \quad (6)$$

where η is the cost per reaction time unit (e.g., in seconds). Note that as $\eta \rightarrow 0$, the optimal decision bound would be $z \rightarrow \infty$. Thus, the goal is to find the optimal balance between resource allocation and optimal bound z that minimizes the loss in Eq. 6 for a given RT cost η and the prior distribution of sensory stimuli in the environment.

First, the numerical solutions reveal that the resource allocation solutions in context K_{Acc} differ from the RT cost-free scenario and depend on RT costs (Figure R2a). While the RT-cost solutions are similar to the RT-cost-free solution for relatively high values of η , the smaller the RT costs the more the allocation of resources tend to flatten. As expected, we found that higher RT costs η result in lower decision bounds b (Figure R2b). Second, in context K_{Rew} , the RT-cost solutions are remarkably similar to the RT-cost-free solution. However, contrary to the K_{Acc} environment, the RT-cost solutions in K_{Rew} appear to get steeper as the RT cost decreases (at least in the range of RT costs studied here). Once again, in context K_{Rew} , higher RT costs η result in lower decision bounds b (Figure R2d)

Figure R2.
Efficient
resource
allocation
solutions
considering

RT costs. **a)** Optimal allocation of neural resources in context K_{Acc} , considering different RT costs η . For reference, we have added the optimal solution of context K_{Acc} in a RT-cost-free environment (blue dashed line). As the RT cost decreases, the allocation of resources appear to slightly flatten relative to the RT-cost-free solution. **b)** Decision bound as a function of RT costs η . As expected, the larger the RT cost the smaller the decision bound. **c,d)** Same as panels a, b, but this time in context K_{Rew} . For reference, we have added the optimal solution of context K_{Rew} in an RT-cost-free environment (red dashed line). While the RT-cost solutions are similar to the RT-cost-free solution, contrary to the K_{Acc} environment, the RT-cost solutions appear to get steeper as the RT cost decreases.

We emphasize that the results presented here are based on a simple DDM with constant bounds. It is well possible that the resource allocation solutions may slightly differ for DDMs where the bounds are allowed to collapse, and this will be an interesting aspect to investigate in future research. Irrespective of these considerations, we show how the general framework developed here leads to a rich set of testable predictions that may lead to the falsification, confirmation, or further refinement of the theory. We have included this set of analyses and discussions in the revised version of the manuscript in the results (p. 10) and methods section (p. 18).

4. There seems to be some confusion in the paper about deep variational information bottleneck (VIB). First, VIB is not a neural network architecture, as implied in the paper (e.g., p6, Fig 5), but rather a combination of an objective function (bound on the information bottleneck objective) and a method for training deep nets to optimize it (similar to VAEs).

Second, there is a discrepancy in the paper about how the ANNs were trained. The main text (p6) states that they were trained with respect to K_{Acc} or K_{Rew} , but in the methods section there's a VIB-like objective function (eq. 23). In addition, eq. 23 seems ill-formed (or at least unclear) — where is the expectation over $p(y|z)$? Isn't the reward a function of y and x rather than a constant? How was beta selected?

More generally, the VIB objective (or eq. 23) suggests an optimality criterion different from the fitness-maximization criterion (i.e., the optimization principle that yields eq. 7). That is, VIB trades (an approximation of) $I(X;Z)$ with the predictability of Y given Z (or reward, in eq. 23), whereas the fitness-maximization criterion does not take into account $I(X;Z)$. The paper, however, argues that sensory neural codes are governed by the fitness-maximization criterion, and does not mention VIB until the ANN section. Why, then, didn't the authors train the ANNs with respect to the same objective function they're arguing for? How would the fitness-maximization results compare with those obtained by VIB?

Resp: First, we apologize for the confusion generated due to the description of the ANNs in the original submission. In the revised version of the manuscript, we now make sure that VIB is not defined as an architecture, but as a method for training the information bottleneck objective.

Second, we agree with the reviewer that the presentation of the ANN objective function in the original submission (Eq. 23) was not clear and did not explicitly reflect that the network was indeed trained with respect to K_{Acc} and K_{Rew} . In the revised version of the manuscript, the VIB-like objective function for K_{Acc} now reads as follows

$$\begin{aligned} & \mathbf{x}_2))p_{\theta}(y = 1 | \mathbf{z}_1, \mathbf{z}_2)) + \\ & \kappa_L (p_{\phi}(\mathbf{z}_2 | \mathbf{x}_2) || p(\mathbf{z}_2)), \end{aligned} \quad (7)$$

with $y = 1$ when the correct response is given by stimulus input \mathbf{x}_1 , and $y = 0$ otherwise. The parameters of the encoder are given by ϕ and the parameters of the downstream decision network are given by θ . We note that when training the ANN, the parameters of the encoder ϕ are shared for both retinal locations where the stimuli $\mathbf{x}_{1,2}$ are presented. $p(y = 1 | \mathbf{z}_1, \mathbf{z}_2)$ is the probability that the network chooses \mathbf{x}_1 given the encoding vectors $\mathbf{z}_{1,2}$. The VIB-like objective function for K_{Rew} reads as follows

$$\min_{\phi, \theta} \mathbb{E}_{p_{\phi}(\mathbf{x}|\mathbf{x})} \{ |s(\mathbf{x}_1) - s(\mathbf{x}_2)| \{ y(\mathbf{x}_1, \mathbf{x}_2)(1 - p_{\theta}(y = 1 | \mathbf{z}_1, \mathbf{z}_2)) + (1 - y(\mathbf{x}_1, \mathbf{x}_2))p_{\theta}(y = 1 | \mathbf{z}_1, \mathbf{z}_2) \} + \beta \mathbb{E}_{\mathbf{X}} [D_{KL}(p_{\phi}(\mathbf{z}_1 | \mathbf{x}_1) \| p(\mathbf{z}_1)) + D_{KL}(p_{\phi}(\mathbf{z}_2 | \mathbf{x}_2) \| p(\mathbf{z}_2))] \} \quad (8)$$

which is identical to the K_{Acc} VIB-like objective function, except that the probability of an erroneous ANN decision is weighted by the absolute value of the difference in the cardinality values $s(\mathbf{x}_1)$ and $s(\mathbf{x}_2)$. Thus, the ANNs trained with VIB-like objective functions penalize errors following the K_{Acc} and K_{Rew} objectives.

Nevertheless, the reviewer is right in pointing out that the VIB-like objective trades (an approximation of) $I(X; Z)$ with the expected reward loss, whereas the analytical solutions developed in our work "drop" costs on $I(X; Z)$ by assuming that the noise in the encoder is small compared with the dynamic range of the signal (i.e., the small-noise approximation, which is commonly adopted in early sensory systems to study neural coding efficiency, often leading to satisfactory predictions (Tkacik and Bialek, 2016)). But we acknowledge that there is certainly room for further theoretical developments using biophysically informed noise models, and we now make the reader aware of this important matter in the revised manuscript (p. 6, right column). We note that we used this form of the objective function in the ANNs as the VIB-like objective provides a parsimonious way to induce pressures in the encoder to disentangle information up to a certain bound in a systematic manner. We controlled the values of β such that for the given ANN architecture performance was relatively challenging for the network, as we found that values of $\beta \rightarrow 0$ resulted in an almost perfect performance. Moreover, this approach allowed us to study and match the ANN at different levels of performance across the K_{Acc} and K_{Rew} contexts. We have added a supplementary figure in the revised manuscript (Supplementary Figure 3; for convenience added in this letter as Figure R3, below) where we demonstrate how different values of β affected the ANN's performance and the allocation of resources in the bottleneck layer. Crucially, we found that the FI pattern predicted by our analytical solutions (i.e., FI is lower for more cardinal angles but higher for more diagonal angles when the network is trained in context K_{Rew}) holds across different levels of ANN accuracy performance (and therefore across different values of β).

Figure R3. Detailed analyses of the VIB-like objective in the ANNs. a) Performance of the ANN trained in context K_{Acc} in terms of proportion of correct responses (left panel) and expected reward loss (middle panel) as a function of different β values. The relation between the expected reward loss and the expected proportion of correct responses is shown in the right panel for several ANNs trained at different β values (each network is an open circle). b) Same as panel a but for the ANN trained in context K_{Rew} . The relation between expected reward loss and the proportion of correct responses can be thought of as a rate-distortion curve. We studied the relation of the form:

$A \cdot (1 - \exp\{b \cdot \log(2E[\text{correct}] - 1)\})^c$, where A is the expected reward loss when the expected proportion of correct responses is at chance level (i.e., $E[\text{correct}] = 0.5$). Thus, A is determined by the reward contingencies of the environment (i.e., is not a free parameter), and therefore the free parameters that determine the shape of the curve are b and c . Note that this is a monotonically decreasing function where the extreme points are interpretable: $E[\text{reward loss}; E[\text{correct}] = 0.5] = A$ and $E[\text{reward loss}; E[\text{correct}] = 1] = 0$ as desired. The fits of this curve to the behavior of the ANNs are the solid lines shown in the right plots of panels a and b, which we found to capture the behavior of the ANNs very well. **c)** We took advantage of the excellent qualitative fits of the relation between the expected reward loss and the expected proportion of correct responses to estimate the proportion of reward loss of the ANN network trained in context K_{Acc} vs the same metric for the ANN trained in context K_{Rew} as a function of the expected proportion of incorrect responses (solid black line). We found that for the range of β values studied here, the ANN trained in context K_{Acc} always led to a reward loss relative to the ANN trained in context K_{Rew} . **d)** Fisher's Information (FI) of the second retinotopic layer of the ANN trained K_{Acc} (blue) and K_{Rew} (red) contexts at different matched levels of accuracy performance across the ANNs (grey dashed arrows in panel c). In line with the analytical predictions, FI is lower for more cardinal angles but higher for more diagonal angles when the network is trained in context K_{Rew} across different levels of ANN accuracy performance (and therefore across different values of β).

5. *Is each ANN trained only on one context (K_{acc} or K_{rew})? If so, this seems like a major difference compared to the behavioral experiment, where the same participants adapt to different contexts. A more relevant simulation would be to train the same ANN on the two contexts and see if/how it adapts.*

Resp: We did both, each ANN was trained in each context, and also we trained the network following the behavioral experiments, namely, the ANN is first trained using the K_{Acc} objective, and after convergence the same network was assigned the K_{Rew} loss objective (these are the results presented in Figure 5d). In both cases the encoder of the ANN displays virtually the same Fisher's Information in the corresponding context. We now make sure that this point is made clear in the revised version of the manuscript (p. 8, right column).

6. *I view this study as part of a growing body of work on information—fitness tradeoffs in neuroscience, and not as an entirely new theory as seems to be implied in the paper. In particular, the framing (especially in the introduction) does not distinguish between traditional infomax efficient coding approaches and more recent approaches that consider efficient coding in the sense of rate-distortion theory or related principles like the information bottleneck (e.g., refs 6, 10, 13, and 14 in the paper) which can be seen as tradeoffs between minimizing coding errors and maximizing fitness / reward / performance in some task. This is quite jarring because some of the references in the introduction consider the information bottleneck principle, which the current paper also employs. It's also worth noting in this context that the information bottleneck has previously been applied to neural representations in the retina:*

Palmer et al., Predictive information in neurons. PNAS, 2015.

Wang et al., Efficient encoding of motion is mediated by gap junctions in the fly visual system. PLOS Computational Biology, 2017.

In addition, the paper doesn't address two related lines of work that have also argued that sensory neural representations are governed by tradeoffs between information and reward:

Tkacik and Bialek, Information Processing in Living Systems. Annu. Rev. Condens. Matter Phys. 2016.

Friston, The free-energy principle: a rough guide to the brain? TiCS, 2009.

To clarify, this comment is not about adding more references to the paper. It is about explicitly drawing the connection between the current study and the broader literature on information—fitness tradeoffs in neuroscience and cognitive science. The paper already includes several key references in that area, however it does so without the proper context. In my view, giving the right context and drawing the connection mentioned above would increase the significance of paper and its appeal to a broader audience.

Resp: We agree with the reviewer's point that it is essential to explicitly draw connections between the current study and the broader literature on information processing constraints in neuroscience and cognitive science, and make the contribution of our work clear. In the revised manuscript, we have updated the introduction section accordingly (p. 1).

Minor Comments

7. *It would be helpful if the authors could define precisely the reward functions that are used in each part of the main text. This detail is a very central for assessing the paper's claims.*

Resp: We now provide a more precise definition of the reward functions in the corresponding sections of the text.

8. *Figure 3 is a big hard to digest, as it contains a lot of information and the caption is extremely long. Also, in 3b the color bar is reversed (starting at 4.75 and climbing up to 3.5), which is confusing.*

Resp: Thank you for this suggestion. To reduce the size of the figure, we now remove "panel a" which uses a large amount of text in the caption, and we now include it in Figure 2 to illustrate the inference process that we assume in our model. We also reduced the length of the captions. We reversed the color bar as suggested by the reviewer.

9. *Figure 5a is confusing: a neural architecture for the information bottleneck can be any network that implements an encoder $p(z|x)$ and a decoder $p(y|z)$. In other words, "comparison and decision" component is part of the information bottleneck. In addition, the text below the arrow, $p(z|x)$, seems misplaced.*

Resp: Thank you for this observation. The arrow and $p(z|x)$ were indeed misplaced, and we now corrected this. Also, we replaced "Information Bottleneck" with "Sensory Encoding", which we believe is more appropriate.

10. *p8: "Crucially, we show that no matter how complex, flexible, or unconstrained downstream decision circuits are, they cannot maximize fitness if the encoding step maximizes accuracy rather than reward consumption."*

This statement seems way too strong, as the paper only investigates a few very specific settings.

Resp: We rephrased and toned down this statement in the corresponding part of the results section in the revised manuscript, which now reads as follows (p. 8, right column):

The findings from our artificial neural network analysis crucially clarify our human behaviour results. Our ANN analyses reveal how a fixed set of physical sensory inputs that have relevant but hidden environmental/contextual statistics that the agent can only experience and learn over time are represented in coding schemes to maximize fitness. Moreover, studying ANNs with a variational information bottleneck is useful because it provides a reasonably realistic model of how encoding schemes are adapted to optimize a given objective function when the resources to process information are limited.

11. *On p11, after eq 5, I think alpha is a typo (shouldn't it be gamma?).*

Resp: Thank you, we have corrected the typo.

12. *Also on p11, $h(s)$ is presented as a utility function, however, earlier in the paper it is presented as the neural response.*

Resp: The reason why we presented $h(s)$ as "utility function" in that part of the text is that we intended to make an analogy with the findings and language in the Economics literature. But, considering the reviewer's observation, we now make it clear that $h(s)$ corresponds to the internal representation, which in Economics is known as the "utility function", whereas, in a neural circuit it corresponds to the neural response.

Reviewer 2

1. *The authors present well designed experiments and indeed show that signatures of performance shift in ways that are predicted by a code which maximizes reward under information constraints. The claims that these changes must adhere at the earliest stages of sensory processing in order for the organism to benefit from such a coding scheme warrant further investigation.*

In particular, the power of this argument from the ANN analysis would be significantly strengthened if the authors manipulated the stage of processing at which the information bottleneck occurs. In the paper, the authors place the bottleneck after 4 convolutional filters and before a fully connected decision layer. This bottleneck models information constraints at early sensory processing stages. However, in the presented architecture the information bottleneck could go even earlier, for example after the very first convolutional filter. This would provide a strong demonstration that systems with information bottlenecks at the very earliest phases of sensory processing, like e.g. blowfly LMCs, exhibit reward maximizing coding. Another architecture the authors should check in order to validate their results is one where the information bottleneck is moved to the decision-making layers in the network. In one potential architectural set up, the retinal layers would embed each image and subsequently a fully connected layer would process the concatenation of these embeddings and then pass through the information bottleneck. With the addition of this model we could determine more precisely whether the results in this section are due to an information bottleneck at the earliest phases of sensory processing, or rather result from any information limited, reward maximizing code.

2. *Another important argument from this section comes from the retraining analysis, where an encoder trained on decision-accuracy displays a large decrease in performance when the decision weights are retrained in the K_{Rew} context. Presumably, this is because under information constraints the encoder chooses not to represent information about the least likely orientations i.e. the oblique orientations, and these are exactly the orientations which provide the highest reward in the K_{rew} . With all that in mind, it would be illuminating to show how the strictness of the information bottleneck, controlled by the β parameter in the loss function, affects the results in this section. How much reward can you recover by easing the bottleneck (i.e. reducing β) How do the curves of information content shown in 5c. shift as you let more and more information pass to the decision making layers? Finally, a few more details regarding model training would be useful. What performance criteria is achieved by each network trained in the K_{Acc} and K_{Rew} contexts?*

Resp: Here we respond to the reviewer's comments 1 and 2. Following the reviewer's suggestions, **(i)** we perform a systematic study of the ANN behavior under different levels of information processing costs induced by the parameter β . **(ii)** We further investigate whether Fisher information (FI) signatures predicted by our analytical solutions are also present at the earliest encoding stage (i.e., the first retinotopic layer in our ANN architecture), even if we do not impose the information processing costs at this earliest layer. **(iii)** We study the ANN behavior when the information processing cost is imposed at the decision-making layer.

- (i)** We have added a supplementary figure in the revised manuscript (Supplementary Figure 3; for convenience added in this letter as Figure R4, below) where we demonstrate how different values of β (when the informational bottleneck is applied in the second retinotopic layer) affected the ANN performance and the allocation of resources in the encoder. We found that the FI pattern predicted by our analytical solutions (i.e., FI is lower for more cardinal angles but higher more diagonal angles when the network is trained in context K_{Rew}) holds across different levels of ANN accuracy performance (and therefore across different values of β).

Figure R4. Detailed analyses of the VIB-like objective in the ANNs. **a)** Performance of the ANN trained in context K_{Acc} in terms of proportion of correct responses (left panel) and expected reward loss (middle panel) as a function of different β values. The relation between the expected reward loss and the expected proportion of correct responses is shown in the right panel for several ANNs trained at different β values (each network is an open circle). **b)** Same as panel a but for the ANN trained in context K_{Rew} . The relation between expected reward loss and the proportion of correct responses can be thought of as a rate-distortion curve. We studied the relation of the form:

$A \cdot (1 - \exp\{b \cdot \log(2E[\text{correct}] - 1)\})^c$, where A is the expected reward loss when the expected proportion of correct responses is at chance level (i.e., $E[\text{correct}] = 0.5$). Thus, A is determined by the reward contingencies of the environment (i.e., is not a free parameter), and therefore the free parameters that determine the shape of the curve are b and c . Note that this is a monotonically decreasing function where the extreme points are interpretable: $E[\text{reward loss}; E[\text{correct}] = 0.5] = A$ and $E[\text{reward loss}; E[\text{correct}] = 1] = 0$ as desired. The fits of this curve to the behavior of the ANNs are the solid lines shown in the right plots of panels a and b, which we found to capture the

behavior of the ANNs very well. **c)** We took advantage of the excellent qualitative fits of the relation between the expected reward loss and the expected proportion of correct responses to estimate the proportion of reward loss of the ANN network trained in context K_{Acc} vs the same metric for the ANN trained in context K_{Rew} as a function of the expected proportion of incorrect responses (solid black line). We found that for the range of β values studied here, the ANN trained in context K_{Acc} always led to a reward loss relative to the ANN trained in context K_{Rew} . **d)** Fisher's Information (FI) of the second retinotopic layer of the ANN trained K_{Acc} (blue) and K_{Rew} (red) contexts at different matched levels of accuracy performance across the ANNs (grey dashed arrows in panel c). In line with the analytical predictions, FI is lower for more cardinal angles but higher for more diagonal angles when the network is trained in context K_{Rew} across different levels of ANN accuracy performance (and therefore across different values of β).

- (ii)** We demonstrate that at different values of β imposed in the second retinotopic layer, the general FI pattern in the first layer predicted by the fitness-maximizing strategy (i.e., FI is lower for more cardinal angles but higher for more diagonal angles when the network is trained in context K_{Rew}) holds across different levels of ANN accuracy performance. However, we noted that this was less pronounced relative to the effect found in the second retinotopic layer (see the results for the second retinotopic layer, above). We argue that this general pattern is also found at this early stage as the resources at this processing stage are also finite. These results are presented in Supplementary Figures 3-6 of the revised manuscript.

Figure R5. FI at the first retinotopic layer when informational bottleneck pressures are applied in the second layer. a) This panel is the same as the one shown in Figure R4c, but reproduced here for convenience. We took advantage of the excellent qualitative fits of the relation between the expected reward loss and the expected proportion of correct responses to estimate the proportion of reward loss of the ANN network trained in context K_{Acc} vs the same metric for the ANN trained in context K_{Rew} as a function of the expected proportion of incorrect responses. We found that for all the space of β values studied here, the ANN trained in context K_{Acc} always led to a reward loss relative to the ANN trained in context K_{Rew} . **b)** Fisher's Information (FI) of the first retinotopic layer of the ANN trained K_{Acc} (blue) and K_{Rew} (red) contexts at different matched levels of accuracy performance across the ANNs (grey dashed arrows in panel a). FI is in general lower for more cardinal angles but higher for more diagonal angles when the network is trained in context K_{Rew} across different levels of ANN accuracy performance (and therefore across different values of β). This pattern is less pronounced relative to the one found in the second retinotopic layer (see Figure R4), but it was consistent across different levels of ANN performance as it is evident from the FI ratio between the networks trained in contexts K_{Acc} and K_{Rew} (see panel c).

(iii) We studied how the ANN allocates information processing resources in the first and second retinotopic layers when informational bottleneck pressures are imposed at the decision-making layer. Validating the VIB approach, we found that the performance of the ANN was also modulated by the level of β when informational processing pressures are applied at the decision stage (Figure R6a-c, below).

On the one hand, we found that the FI pattern predicted by our analytical solutions (i.e., FI is lower for more cardinal angles but higher for more diagonal angles when the network is trained in context K_{Rew}) was also present in this scenario in the second retinotopic layer (Figure R6d). However, this effect was less pronounced relative to the scenario when the informational bottleneck pressures were applied directly in the second layer (compare the corresponding results presented in Figure R4d). On the other hand, we found that the effects in layer 1 were marginally present for high levels of network performance (i.e., generally low β in the decision layer) or rather absent for low levels of performance (i.e., high values of β in the decision layer), see Figure R7. Based on this set of results, we conclude that even when information processing pressures are very small at early sensory stages (and relatively large in downstream decision layers), neural networks still try to develop fitness-maximizing codes at the early sensory stages to compensate for reward loss due to processing limitations in downstream circuits. However, these effects are more pronounced if information processing constraints are present at early stages. The results presented here are now incorporated and discussed in the Results and Discussion section of the revised manuscript and the corresponding figures are added as Supplementary Figures 3-6.

Figure R6. Detailed analyses of the VIB-like objective in the ANNs when informational bottleneck pressures are applied at the decision layer. The approach of this figure is the same as that of Figure R4, with the difference that here the informational bottleneck pressure indicated by β was induced at the decision layer. The Fisher's information (FI) analyses presented in panel d are conducted at the second retinotopic layer.

Figure R7. FI at the first retinotopic layer when informational bottleneck pressures are applied in the decision layer. The approach of this figure is the same as that of Figure R5, with the difference that here the informational bottleneck pressure indicated by β was induced at the decision layer. The Fisher's information (FI) analyses presented in panels b and c are conducted at the first retinotopic layer.

3. The behavioral studies are well designed overall. For the version of the behavioral task where participants train in retinotopic specific areas, the authors show that estimation effects of training on are only apparent in the retinotopic regions where training occurred. This indeed suggests that training on reward affects early phases of sensory processing, but without a comparison of reward contexts it is difficult to definitely conclude that these changes reflect a reward maximizing code. In particular, the predicted reduction in the estimation bias shown in Fig. 4b. is also consistent overall increased precision in rewarded location. The data for the estimation variance is in the correct direction, but a key prediction

of the fitness maximizing code - that there should be a significant reduction in variance for oblique directions after K_{Rew} training - isn't immediately evident. In fact, fig 4c shows no decrease in variance after training compared to without training. To establish that early sensory effects here are truly fitness maximizing, and not simply a result of increased information coding in rewarded retinotopic areas, the authors would have to show that the effects in trained area K_{Acc} and K_{Rew} for contexts follow the distinct predictions for fitness-maximization in each context. As far as I can tell, this is not the case.

Resp: We collected an additional data set in which human participants received location-specific training with the accuracy maximizing payoff rules (K_{Acc}). When examining the changes in variance for oblique relative to cardinal orientations, we do not find significant differences (posterior probability = 0.14) between trained and untrained locations in this new K_{Acc} group. The original group trained with location-specific fitness maximizing payoff rules (K_{Rew}) does show the expected pattern with a greater decrease in variance for oblique relative to cardinal orientations in trained relative to untrained locations (posterior probability = 0.96). When comparing across the groups we find that the interaction effect between orientations and location training is greater for the K_{Rew} group than the K_{Acc} group (posterior probability = 0.96).

4. It would also be a good check to show that low performance in the K_{Acc} for oblique trials (Fig. 3h.) aren't simply a result of participants choosing to ignore the most difficult trials in a context that rewards all trials equally. Some analysis of reaction times or other data that shows participants are equally engaged on these trials would be a nice addition.

Resp: We have followed the reviewer's suggestion to analyze the reaction time data as a function of orientation and condition. Consistent with the prediction that they are more perceptually difficult, responses in oblique trials are slower in both conditions. However, there was no main effect of condition nor any interactions between condition and orientation in either early or late training trials. We now include these results in the supplementary materials and have pasted the table below for convenience.

	Estimate	Est.Error	Pmcmc
Intercept	0.69	0.02	0.0001
s	0.04	0.01	0.0001
Krew	0.01	0.01	0.1465
Later	-0.11	0.02	0.0001
s *Krew	0.01	0.01	0.1548
s *Later	0.00	0.01	0.4173
Krew*Later	0.01	0.02	0.1827
s *Krew*Later	0.01	0.01	0.2155

This table reports the results of a Bayesian hierarchical linear regression on response times in the decision task. The regressor Krew is a dummy variable for training type (1 = reward, 0 = accuracy). The regressor s is the diagonality of the stimuli. The regressor Later is a dummy variable for later trials in the decision task (1 = later trials, 0 = early trials).

Minor Comments

Typo: 'the the' below question 23

Resp: Thank you, we have corrected the typo.

The y-axis in Fig 3. b-j was difficult to decipher at first. Consider simply labeling with degrees? Is it representing the orientation of the correct stimulus?

Resp: Thank you for this observation. In panels b,e,h the x-axis represents cardinality value of the correct stimulus, and in panels c,d,f,g,i,j it corresponds to the actual cardinality value as the y-axis metrics are derived from the estimation task. We now clarify this in the figure caption of the revised manuscript.

The b) label in Fig. 3 caption is missing.

Resp: Thank you, we added the missing label.

Reviewer 3

1. Relation to Hoffman et al.'s "Interface Theory"?

The current paper seems to be arguing for a similar interpretation of neural coding as Donald Hoffman and colleagues have put forward in several papers over the last decade under the term "interface theory". E.g. to compare a couple of quotes:

From the abstract of the current paper: "We found that neural codes that maximize reward expectation – and not accurate sensory representations – account for retinal LMC activity."

From the abstract of Hoffman, Singh & Prakash (2015): "We find that veridical perceptions – strategies tuned to the true structure of the world – are routinely dominated by nonveridical strategies tuned to fitness."

The above paper from Hoffman et al is cited, but only in passing in the Discussion, so it is hard to tell how this work relates to and deviates from Hoffman et al's work. Given that the main contribution of the current paper is an explicit computational framework for deriving what a "fitness maximising" neural code should look like in any given domain, it seems critical to explain how this differs from the computational frameworks developed by Hoffman et al. E.g. in the following papers:

- Prakash, C., Stephens, K. D., Hoffman, D. D., Singh, M., & Fields, C. (2021). Fitness beats truth in the evolution of perception. *Acta Biotheoretica*, 69(3), 319-341.
- Prakash, C., Fields, C., Hoffman, D. D., Prentner, R., & Singh, M. (2020). Fact, fiction, and fitness. *Entropy*, 22(5), 514.
- Hoffman, D. D., Singh, M., & Mark, J. (2013). Does evolution favor true perceptions?. In *Human Vision and Electronic Imaging XVIII* (Vol. 8651, p. 865104). International Society for Optics and Photonics.

The 2015 *Psychonomic Bulletin & Review* paper received a large number of commentaries, along with counter-responses from the authors. These extensive discussions on the topic of whether we perceive "fitness" or "reality" are very pertinent to the current paper. To pick just one point of contention, Interface Theory posits that perception departs from reality most clearly when fitness functions are non-monotonic (e.g. to achieve homeostasis, too much salt or sugar is a bad thing, but so is too little salt or sugar). Does the formalism in the current paper allow for non-monotonic fitness functions? My understanding is no; fitness is assumed to be monotonically or even linearly related to some underlying physical property? If the formalism can't encompass non-monotonic fitness functions, this seems to be a major shortcoming. And if it can, it should then address potential problems with that, e.g. the critique raised by Bart Anderson that directly perceiving fitness, in the case of a non-monotonic fitness function, means that the observer has discarded information about the direction in which it should adjust its behaviour:

- Anderson, B. L. (2015). Where does fitness fit in theories of perception?. *Psychonomic bulletin & review*, 22(6), 1507-1511.

Generally, the idea that we perceive fitness, rather than veridical properties of the world, has been extensively debated over the past decade, and it is important to situate this work within that debate. What is the novel contribution (e.g. perhaps explicit predictions about the shapes of neural response curves, which might be considered complementary to Interface Theory's game-theoretic simulations about population fitness)? How does the current proposal differ from that of Interface Theory, and how does it meet the critiques that have been previously raised about IT?

Resp: We thank the reviewer for bringing up this discussion of the work by Hoffman and colleagues on the Interface Theory (IT). We agree that it is essential to position our framework in relation to previous

related work and clarify our novel contributions. In the revised version we now incorporate this previous literature and theory in our introduction and discussion and clarify our contribution. In this letter, we first consider the fundamental differences between the two approaches and then we address the points brought forward by Anderson (2015):

IT departs from the premise that "natural selection tunes perceptions to payoffs" by assuming that perceptions have been directly shaped by tracking the utility associated with the physical objects of the environment. The evolutionary games are then played using the currency of payoffs. A critical aspect that is omitted in their specification (which does not necessarily make IT incorrect as a theory) is that IT remains silent as to how a capacity-constrained sensory system should allocate resources to efficiently *encode* "true" physical features and how these are *decoded* to serve a behavioral goal. That is, ours is a theory of sensory encoding where we provide a normative foundation of how a limited resource should be allocated to maximize expected reward in a binary choice task (where elements from the environment are drawn from a prior distribution), which does not necessarily imply that "objective" reward values tied to the physical stimuli are directly tracked. Moreover and crucially, we explicitly define the nature of both the encoder and decoder. This is not the case in IT. To make our explanation more concrete, we adopt the formalism that Hoffman et. al. use as the departure point to motivate their theory using Bayesian decision theory. For clarity, we will use the notation adopted in our work.

IT attempts to find a *perceptual strategy* of the distribution of possible perceptual interpretations s^\wedge (named x in Hoffman et. al.) for a given image s_0 (named y_0 in Hoffman et. al.), that is, find $p(s^\wedge | s_0)$ that promotes fitness maximization. Hoffman et. al. posit that this can be solved by applying the Bayes formula $p(s^\wedge | s_0) \propto p(s_0 | s^\wedge)p(s^\wedge)$. Then they proceed by stating that to define a perceptual strategy, the possible perceptual interpretations do not need to be identical to the possible states of the world W , where the perceptual strategy can be defined as a mapping $P : W \rightarrow \hat{S}$. They argue that a *naive realist* requires $W = \hat{S}$ and P an isomorphism, while it should be something else for the *interface* strategy depending on the mapping of reward from physical objects to rewards. At no stage of their specification do they define how s_0 should be encoded and decoded in the first place, but we do, and this is where our approach differs from IT in a fundamental manner.

We start a step back and assume that a *noisy* neural response r given a stimulus s generates a distribution of potential neural responses $p(r | s)$. For a given prior distribution over possible *physical* or *environmental* stimuli $p(s)$, we compute the posterior distribution $p(s | r)$ by applying Bayes' rule with $p(s | r) \propto p(r | s)p(s)$ (please note the difference with the Bayesian IT specification where the prior is over possible perceptual strategies $p(s^\wedge)$). Then we also adopt a specific assumption about the nature of the decoder, which we explicitly assume to be based on the expected value of the posterior distribution, that is $s^\wedge = E[p(s | r)]$. Finally and crucially, we have a specific function to optimize: Fisher's information. Here the goal is to efficiently allocate the precision of possible neural responses r over the space/dimension where s lives such that reward is maximized for the set of possible decisions that the organism may encounter in the environment. Please note that, as in the Bayesian IT, our approach leads ultimately to a distribution over potential decoded precepts $p(s^\wedge | s_0)$ due to the stochasticity in the neural responses r for a given s_0 . But in our approach, this distribution emerges from an inference process based on fundamental properties of the concrete encoding and decoding operations.

Having explained the differences between the two approaches, next we move to the topic of (non)monotonicity in reward mappings raised by Anderson (2015). In the specification of Hoffman et al., a *realist* organism is defined as one that monotonically maps available perceptual responses (e.g., color precepts in their example) to ranges of a resource quantity (e.g., salinity, following the example in Anderson 2015) irrespective of how resource quantity maps to reward. However, in the scenario where too much or too little salinity has deleterious effects on the organism, intermediate levels of salt will have higher payoffs in the salinity space. In this case, the interface strategy suggests that the color precepts are monotonically mapped to payoffs, which is problematic. As indicated by Anderson, this means that "a small amount of salt is experienced in exactly the same way as an excessive abundance of salt", and that this is critical because "if perception is tuned directly to the payoff structure, an animal will only know how good or poor a payoff is" but this provides "no clue to the adaptive course of action for an animal". We actually agree with this issue raised by Anderson and, as we will show below in more detail, this does not necessarily occur under our model specification and the set of assumptions outlined above. There is a second issue that follows from the issue raised by Anderson: Even in the limit of a large amount of

available perceptual resources (e.g., nearly infinite amount of color percepts in the example) organisms will be unable to differentiate high vs low levels of salinity. To see why this does not necessarily occur in our approach, we start by noting that the objective function to be minimized in the case of maximizing reward expectation (in two alternative decision tasks when physical objects are mapped to a payoff level) is equivalent to minimizing (proof given in Supplementary Note 1)

$$E[|\text{loss}|] = \int \int f(s_1, s_2) P(\text{error} | s_1, s_2) |R(s_1) - R(s_2)| ds_1 ds_2. \quad (9)$$

Note that this expression does not depend on explicit payoffs of each choice alternative $s_{1,2}$ individually, but depends on the absolute difference of the reward R delivered by sensory stimuli (independent of the range they are drawn from), thus, irrespective of reward mapping this objective is always equivalent to minimizing the reward that is given up with every wrong decision. Another important observation in light of this discussion is that in the standard accuracy maximizing discrimination task (where the observer receives a constant reward R for each correct decision) the reward mapping to sensory stimuli can be interpreted as "*nonlinear*" if the choice rule is to choose the stimulus with the highest "physical value".

Consider the following example: Assume three stimuli with scalar values $s_A > s_B > s_C$. If the choice set is $s_{A,B}$ then the agent receives reward R if she chooses s_B . However, if the choice set is $s_{B,C}$ and the agent chooses again—but this time erroneously— s_B then the reward received is 0. Thus, the reward received can take one of two values depending on the choice set. Crucially, note that this contrasts with the reward-maximizing case, where the agent receives the reward R_B associated with stimulus s_B any time the agent chooses it irrespective of the choice set it is included in. Why is this important? Like in any decision-making task the agent must have an understanding of what sensory values are larger than others (by relying on decoded values based on learning/experience) and use this knowledge to guide decisions even if physical values are mapped to rewards via some (non)monotonic function. Although this sounds trivial, this part of the discussion will be important to study the non-monotonic reward mapping problem.

Now the question is, how should the neural resources be allocated for the case in which there is a non-monotonic mapping between physical stimulus values and rewards? What strategy should the agent follow? We consider the following three scenarios. Scenario 1 corresponds to the accuracy maximization task (K_{Acc} in our manuscript). Scenario 2 corresponds to a reward-maximizing task where rewards are linearly (monotonically) mapped to the physical stimulus values (K_{Rew} in our manuscript, Figure R1a). Scenario 3 corresponds to a non-monotonic mapping where stimuli in the middle of the physical space have the highest reward values (Figure R1b). For all three scenarios, we consider a right-skewed distribution of sensory stimuli over the physical value space (Figure R1a,b) to aid comparison with experiments in humans. For scenario 3 there is no known closed-form solution to find the optimal allocation of resources (i.e., Fisher's information $J(s)$), but please note that the minimization objective remains the same. Therefore, we employ numerical methods to solve the resource allocation problem for all three scenarios.

Following our previous discussion, a key assumption of our framework is that there is an understanding of what stimuli deliver more reward (through experience, observation, or instructions; this is also the case in the evolutionary games in the work by Hoffman et. al.). For scenarios 1 and 2, the agent must have learned that sensory stimuli with higher values are preferred. In scenario 3, however, stimuli in the middle are preferred. The decoding rule in our model is the same in all cases: the Bayesian mean squared error (BMSE). What is the strategy for the cases where the stimuli-reward mappings are monotonic or non-monotonic unimodal? A relatively simple strategy that preserves the "veridicality" of sensory information is one in which the agent employs a "categorization" threshold τ over the space of physical stimuli and decodes the values s^\wedge relative to that threshold. A simple implementation is one where the agent computes a relative decoded value

$s^\sim = -|\tau - s^\wedge(s_0)|$, an operation that could be flexibly implemented in downstream circuits. The choice rule is then

choose s_1 if $s^\sim_1 > s^\sim_2$, otherwise choose s_2 . Thus, in addition to finding the optimal resource allocation function, the threshold τ is an additional latent variable to solve the reward maximization problem.

Before solving the optimization problem numerically, we note that the predictions for τ are relatively intuitive.

In scenarios 1 and 2, τ should be set to the maximum stimulus value in the physical space, and the optimal resource allocation solutions remain the same as derived in our manuscript. In scenario 3, with reward values peaking in the middle of the stimulus distribution, the threshold will be likely located at $\tau \approx 0.5$ in our example in the encoding low-noise limit (not precisely at 0.5 due to biases and variance of \hat{s} and the related influence of the prior distribution of physical stimuli).

The numerical solutions of resource allocation for scenarios 1 and 2 resemble, as expected, the analytical solutions where more resources are allocated to regions of the physical space with the highest physical prior density, but the amount of information is larger for lower sensory values in scenario 1 but larger for higher sensory values in scenario 2. For scenario 3, the solution for the "categorization" threshold is $\tau \approx 0.5$, and the resource allocation solution is in principle surprising and counter-intuitive (Figure R8c), but taking a closer look at the problem, it actually makes sense. First, we observe that the allocation of resources has a general trend to decrease as the sensory stimulus gets larger, thus following the expected result given the shape of the prior distribution of sensory stimuli. Second, the resource allocation solution has a dip at around $s = 0.5 \approx \tau$. This may appear initially counter-intuitive given that these are the regions where the reward is highest. However, note that (i) randomly drawing choice sets from the prior distribution generates with more likelihood choice sets that are close in value, and (ii) choice sets $s_{1,2}$ with values close to $s = 0.5$ are the problems where more mistakes are made (e.g., $s_1 = 0.52$ and $s_2 = 0.48$ deliver the same reward). Thus, it is not worth investing too many resources near $s = 0.5$ even if the reward promised at those locations is high (Figure R8d).

Figure R8. Studying efficient allocation of neural resources with non-monotonic stimulus-reward mappings. a,b) The prior distribution of sensory stimuli in the environment monotonically decreases with sensory stimuli (black) and is the same in all scenarios. The stimulus-reward mapping function in Scenario 2 monotonically increases following a linear relationship (red, panel a), and in Scenario 2 is non-monotonic with the highest reward delivered at $s = 0.5$ (green, panel b). c) Optimal solutions of the resource allocation problem for Scenario 1 (blue), Scenario 2 (red), and Scenario 3 (green). d) Percentage reward lost assuming that the agent makes use of the optimal resource allocations from the accuracy maximization (K_{Acc} , blue) and reward maximization (K_{Rew} , red) environments in the non-monotonic stimulus-reward mapping environment relative to the corresponding optimal solution.

Taken together, we propose a possible heuristic where our general reward expectation maximization objective alongside a resource-constrained system leads to a solution that promotes fitness maximization while still allowing to decode information about the veridical physical attributes of the environment that are fundamental for coping with adaptive processes. We emphasize that this is just one alternative (there might be better ones), but one that generates novel predictions that could be tested in future experiments. We thank the reviewer for this comment. We now include the structure of this response in the results section of the revised manuscript (pp.

9-10), which we hope readers may find useful to guide future research.

2. Assumptions and limitations of the modelling framework?

The main contribution of the paper is to present a quantitative way to derive and compare predicted neural response or psychometric functions, given the distribution of stimuli in the environment, under the assumptions that the perceptual system is either trying to maximize (a) ability to recover true stimulus value (e.g. orientation, contrast level, etc), or (b) expected reward (assuming different values of the stimulus are associated with different reward levels). The solution is adapted from a 2009 economics paper on mental representation to support optimal choice-making, given limited representational resources. The application to neuroscience seems novel and is potentially valuable.

However, I would like to see a clearer presentation of the assumptions and limitations of the modelling

framework. For example, it seems to assume a monotonic (p2) or perhaps even strictly linear (p11) mapping between stimulus values and reward values. If so, this seems like a major limitation. The need for homeostasis means that for most animals most physical properties (brightness, heat, salination, etc) likely have "sweet spots" at moderate levels, with very high or very low levels being detrimental. In the empirical example of flies seeking flowers by contrast levels, it doesn't seem at all obvious that the highest contrast levels are the most rewarding. The highest contrast levels in a natural environment are likely to be created by behaviourally-neutral things like any object silhouetted against the sky. Whereas the contrast differences between different flower species or other food sources are likely to occupy some range of moderate contrast levels. Can the framework accommodate such a possibility, and can it make testable predictions about which fitness functions best account for the neural code?

*I may well be missing something crucial in the explanation, but it seemed that the transformation from the distribution of stimulus values to the predicted fitness-maximizing neural/perceptual function happens *without specifying* any particular fitness function? If this is the case, does the framework make any predictions about how neural coding or perception should change when the fitness function in the environment changes? What happens at edge cases; for example, if a linear mapping from stimulus value to reward is sufficiently shallow, the difference between "veridical encoding" and "reward encoding" should presumably collapse?*

Resp: We agree with the reviewer that it is important to highlight the reach and limitations of our theoretical quantifications.

Regarding the first part of the reviewer's comment about non-monotonicity, we believe that we have addressed most of it in our response to the previous comment. Nevertheless, it is important to clarify to the reader the reach of our theory. As the reviewer pointed out, our main contribution is that we provide analytical solutions of how a (neural) system should allocate information processing resources for two of the most common problems studied in decision-making: Accuracy maximization in perceptual discrimination tasks, and reward maximization in the standard and most studied economic problem where properties of a good or action scale monotonically with value, and therefore may lead to predictions of how neural systems could generate utility functions. Intriguingly, we found that the solution to this problem looks remarkably similar, belonging to the family of power-law efficient codes, where we formally and empirically demonstrate that (i) the two problems lead to different resource allocation solutions, and (ii) the only difference is the power-law coefficient. However, we acknowledge that when the system must deal with more complex sensory-reward mappings, the analytical solution to the resource allocation problem is unknown (and may not exist). Nevertheless, we provided hints as to how this problem could be treated using the general optimization criteria proposed in our work, and our argument continues to be the same: Given that noisy communication channels such as the brain always lose information during transmission, we argue that it is more efficient for sensory systems to allocate neural resources to adapt to the utility-maximizing rules of a particular environment at the earliest stages of sensory processing.

Regarding the second point: (i) Our analytical solutions are restricted to the problems mentioned above but, with the use of categorization thresholds, we provided some hints as to how the system can adapt to non-monotonic solutions. However, we acknowledge that our theory does not explain the dynamics of adaptation, but generates predictions once the system has adapted after learning from experience. Thus, it remains unclear what the normative algorithms of efficient adaptation might be and how these could be connected with a biologically-plausible algorithm that applies to arbitrary stimulus-reward association contexts such that reward expectation is maximized. This topic deserves attention and should be studied further. (ii) For the problems of accuracy and reward maximization with linear sensory-reward mappings, our model predicts that in edge cases where the prior distribution is approximately flat, the optimal solutions are indistinguishable and the agent should allocate resources equally across the whole sensory space in both cases. (iii) There is an additional prediction that we did not emphasize enough in the original submission: if the prior density is low for low sensory values, as well as for high sensory values, and there is a linear stimulus-reward mapping across the whole sensory space, our model predicts that, relative to the standard accuracy maximization task, sensitivity should also increase for low sensory values. Psychologically speaking this could be thought of as a "saliency effect". Our model predicts that this "saliency effect" should get more pronounced during a reward maximization task vs a standard discrimination task (a hint of this prediction can be found in Supplementary Figure 1).

Following the reviewer's suggestion, we have now added this information to the discussion section of the

3. Level of empirical evidence provided for the theoretical claims.

The paper makes a major claim: that neural encoding and perception are attuned not to the statistics of our environments, but rather to the distribution of rewards in our environments. They support it via empirical evidence from fly electrophysiology, human psychophysics, human fMRI, and deep neural network simulations. This array of methods and target systems is impressive and ambitious. However, the quality of evidence provided by each seems underwhelming.

The fly electrophysiology evidence consists of reanalysing a set of recordings from blowfly retinal cells from a (seminal) 1981 paper, and relating these to contrast measurements from natural blowfly habitats reported in the same paper, with the help of some simplifying assumptions (that higher-contrast stimuli have monotonically/linearly(?) higher reward value to blowflies). The fitness-maximising neural response curve derived does indeed better align with the cell recordings than does the classical interpretation in terms of maximising discrimination ability given the statistics of the environment. This is suggestive but not very strong evidence. Given that the differences between the two predictions are a matter of changing slope in a sigmoidal function, could there be other explanations (e.g. that there is a small mismatch between the contrast statistics of the woodlands and lakesides from which the measurements are taken, and the actual experienced contrasts of the experimental laboratory flies)? How plausible is the assumption that higher contrast levels have higher reward values for blowflies? A really compelling test of the hypothesis would involve manipulating both the environmental statistics and the reward mappings for new sets of flies, and showing that retinal responses reliably followed the reward mapping rather than the environmental statistics.

The human fMRI evidence is the weakest, and is raised only briefly in the Discussion. It shows only that task-relevant information (e.g. attention and task reward value) can modulate representations in early visual areas over the slow timescales of fMRI, which is consistent with conventional views on visual encoding (e.g. that specific object features can be amplified in V1 or even earlier via top-down attention). The paper seems to be arguing for a more radical interpretation than this - i.e. that V1-3 are discarding veridical feature information and instead representing only reward value? It would be good to spell out exactly how radical the argument is, and how it departs from widely held views of attention-as-gain-control that can flexibly amplify reward-relevant features.

The bulk of the empirical contribution of the paper describes two psychophysics experiments. They show that after receiving training in which oblique orientations are more rewarded than cardinal ones (vs a comparison group who receive equal rewards for all orientations), participants' discrimination accuracy improves for oblique orientations, and there is a reduction of the usual "oblique effect" (higher precision for cardinal than oblique orientations) in that their response variance in an adjustment task decreases for oblique and increases for cardinal orientations, and their cardinal bias decreases.

This all seems consistent with observers deploying greater attention and/or more stringent response criteria towards the more task-rewarded orientations, and thereby improving discrimination and matching performance at those orientations at the expense of others. It has previously been shown that attention to specific orientations has both neural and perceptual consequences, e.g.:

*-Liu, T., Larsson, J., Carrasco, M. (2007). Feature-based attention modulates orientation-selective responses in human visual cortex. *Neuron*, 55(2), 313-323.*

It has also previously been reported that both attention and task-based rewards can modulate orientation adaptation, raising the possibility that different orientations were differently adapted during the repeated exposure of the discrimination task (which would also be consistent with the effect being retinotopically-localised):

*-Festman, Y., Ahissar, M. (2004). Attentional states and the degree of visual adaptation to gratings. *Neural Networks*, 17(5-6), 849-860.*

*-Pascucci, D., Turatto, M. (2013). Immediate effect of internal reward on visual adaptation. *Psychological Science*, 24(7), 1317-1322.*

The authors say in the Discussion that they are agnostic about the particular mechanisms by which neural codes come to maximise fitness, and that adaptation and attention are two plausible short-time-scale mechanisms. But this leaves me wondering what the new contribution of this paper is, since most would agree that rewarded stimuli attract attention, with concomitant benefits to perceptual and behavioural precision.

Is the novel contribution the normative framework for predicting exactly what form that should take? (If so, I would like to see more tests and discussions of how the framework can handle various and arbitrary reward mappings)

Or is the argument a more radical one, along the lines of Interface Theory? (If so, I would like to see more explanation of why the presented evidence isn't consistent with more standard views of neural codes that encode physical stimulus dimensions but whose sensitivity can be modulated by reward and relevance).

Resp. In our response to this point, we address each comment made by the reviewer in turn.

Blowfly data: We agree with the reviewer that the qualitative analyses of this data should not be seen as definitive evidence supporting the fitness-maximizing hypothesis, as the evidence is mostly anecdotal. We acknowledged this in our original submission where we wrote (p. 2): *"While the predictions of reward-maximizing sensory codes and the data from blowfly retinal neurons (i.e., the earliest level of encoding) show a striking similarity (Fig. 1), this finding does not directly support all aspects of our hypothesis that neural codes in early sensory areas adapt to the organism's behavioral needs. This is because we don't know the specific function linking contrast to fitness for the blowfly, and we can't show that the code used in their retinas adapts between contexts because we only have data from one context"*. Our rationale for making use of the blowfly data was to use it as motivation to introduce our framework, and after discussing with colleagues, they all agree that using it is a good way to introduce the theoretical framework, as long as we make it clear to the reader that the evidence is not conclusive. We agree with our colleagues and reviewer that conclusive statements about the results from this analysis should be toned down. We revised the text accordingly in the new version of the manuscript (abstract, introduction, results, and discussion), and we hope that the degree of "suggestive" evidence is made clear at the moment of introducing the example. If the editor and reviewer decide that it will be more convenient to take this out from the manuscript, we will agree to do so, however, we hope you agree that introducing the framework with the blowfly data is an instructive way to illustrate and introduce the topic and the theoretical framework. Additionally, we now make sure that all the conclusions that can be derived from the empirical results are supported by the human data.

fMRI data: It was not our intention to convey the radical idea—from the fMRI analyses—that V1-V3 are discarding veridical feature information and instead represent only reward value. We agree that the way in which we wrote this part of the text in the original submission may have led to misinterpretations of our arguments. Our intention with these analyses was to demonstrate that, already at the early stages of sensory processing, visual systems process abstract goal-directed information. In other words, fMRI RSA analyses only allow concluding that early visual systems employ some of their resources to represent abstract goal-directed information, and this does not imply that the representation of veridical information is discarded. Moreover, we also make clear that these results do not explicitly support the quantitative theory developed here (the reason why we included this in the discussion section and not as a main result), but instead the general idea of why a system should employ resources in its early sensory areas to represent abstract behavioral goals if this could also be implemented downstream. Again, the main idea we want to convey is that given that noisy communication channels such as the brain always lose information during transmission, it is arguably more efficient for sensory systems to allocate neural resources that promote fitness maximization at the earliest stages of sensory processing. Although veridicality might be compromised (resources are finite) strategies might be implemented to ensure that it is not completely suppressed (e.g., disentangling via orthogonalization, e.g. see Avitan and Stringer, Neuron 2022).

We have revised this part of the discussion to avoid such misinterpretations occurring in the future.

Psychophysics experiments: We agree with the reviewer, and we actually pointed to some of that literature in our original submission: it has previously been shown that attention to specific sensory features has both neural and perceptual consequences and both attention and task-based rewards can modulate sensory adaptation.

Answering the reviewer's question *"Is the novel contribution the normative framework for predicting exactly what form that should take?"* Our answer is: yes. Beyond the phenomenological explanations

provided in most previous work, there was no normative foundation of why sensory-reward mappings at the neural level should adapt in the first place, and what form they should take for different behavioral goals. Also, thanks to the insightful comments by this reviewer in point 1, and in our response to these comments, we show how our framework can generate a set of novel predictions of how limited-capacity systems may allocate their finite resources, also in the presence of non-monotonic sensory-reward mapping functions, and that strategies can be developed to avoid the "suppression" of veridicality while allowing to promote fitness. We make sure that this idea is conveyed clearly, and we hope the reviewer agrees that our theory opens the door for a breadth of future research at the behavioral and neurobiological level that might help falsify, confirm, or improve the theory and related frameworks of fitness maximization in other disciplines.

Decision Letter, first revision:

6th February 2023

Dear Dr. Hare,

Thank you for submitting your revised manuscript "Sensory perception relies on fitness-maximizing codes" (NATHUMBEHAV-211117187A). It has now been seen by the original referees and their comments are below. As you can see, the reviewers find that the paper has improved in revision. We will therefore be happy in principle to publish it in Nature Human Behaviour, pending minor revisions to satisfy the referees' final requests and to comply with our editorial and formatting guidelines.

We are now performing detailed checks on your paper and will send you a checklist detailing our editorial and formatting requirements within a week. Please do not upload the final materials and make any revisions until you receive this additional information from us.

Sincerely,

Samantha Antusch

Samantha Antusch, PhD
Senior Editor
Nature Human Behaviour

Reviewer #2 (Remarks to the Author):

Thanks to the authors for their thoughtful revisions in light of the concerns raised in the original review. In general, I think the neural network analysis shows much more convincingly that signatures of fitness-maximizing codes are present in the initial stages of sensory processing in the ANNs, and that this effect is modulated in a sensible way by varying the strictness of the information bottleneck. Further, the additional data on location specific training of both Krew and Kacc conditions does confirm that encoding of oblique vs. diagonal directions is only significantly affected in locations exposed to Krew training. This is consistent with a fitness-maximizing code. Overall I think all appropriate revisions have been made in response to the original review.

One typo: page 6 right below equation 4; 'AAN' should presumably be 'ANN'.

Reviewer #3 (Remarks to the Author):

I thank the authors for engaging exceptionally deeply with my comments and those of the other reviewers. The revised paper contains substantial new analyses that provide quantitative answers to questions we raised during review, such as how resources would be allocated under non-monotonic reward functions, or how additional objectives, like minimising reaction times, could be incorporated into the framework. I think these strengthen the work, and point to useful testable hypotheses.

I am still digesting the messages of this paper, but I think it provides a useful formalisation, and will generate valuable discussion in the field.

Final Decision Letter:

Dear Dr Hare,

We are pleased to inform you that your Article "Sensory perception relies on fitness-maximizing codes", has now been accepted for publication in Nature Human Behaviour.

Please note that Nature Human Behaviour is a Transformative Journal (TJ). Authors whose manuscript was submitted on or after January 1st, 2021, may publish their research with us through the traditional subscription access route or make their paper immediately open access through payment of an article-processing charge (APC). Authors will not be required to make a final decision about access to their article until it has been accepted. **IMPORTANT NOTE:** Articles submitted before January 1st, 2021, are not eligible for Open Access publication. Find out more about Transformative Journals

Authors may need to take specific actions to achieve compliance with funder and institutional open access mandates. If your research is supported by a funder that requires immediate open access (e.g. according to Plan S principles) then you should select the gold OA route, and we will direct you to the compliant route where possible. For authors selecting the subscription publication route, the journal's standard licensing terms will need to be accepted, including self-archiving policies. Those licensing terms will supersede any other terms that the author or any third party may assert apply to any version of the manuscript.

With best regards,

Samantha Antusch

Samantha Antusch, PhD
Senior Editor
Nature Human Behaviour